# FIRE: Multi-Fidelity Regression with Distribution-Conditioned In-Context Learning Using Tabular Foundation Models

**Rosen Ting-Ying Yu** [1]   **Nicholas Sung** [2]   **Faez Ahmed** [1,2]

## Abstract

Multi-fidelity (MF) regression often operates in regimes of extreme data imbalance, where the commonly-used Gaussian-process (GP) surrogates struggle with cubic scaling costs and overfit to sparse high-fidelity observations, limiting efficiency and generalization in real-world applications. We introduce FIRE, a training-free MF framework that couples tabular foundation models (TFMs) to perform zero-shot in-context Bayesian inference via a high-fidelity correction model conditioned on the low-fidelity model's posterior predictive distributions. This cross-fidelity information transfer via distributional summaries captures heteroscedastic errors, enabling robust residual learning without model retraining. Across 31 benchmark problems spanning synthetic and real-world tasks (e.g., DrivAerNet, LCBench), FIRE delivers a stronger performance–time trade-off than seven state-of-the-art GP-based or deep learning MF regression methods, ranking highest in accuracy and uncertainty quantification with runtime advantages. Limitations include context window constraints and dependence on the quality of the pre-trained TFMs. Code & data can be found here: https://github.com/rosenyu304/FIRE.

## 1. Introduction

Multi-fidelity (MF) regression is the problem of learning to predict an expensive, high-accuracy target function using a small number of high-fidelity observations together with a larger set of cheaper but less accurate low-fidelity (LF) observations. Multi-fidelity regression typically leverages probability models that couple high-fidelity (HF) measurements with abundant, noisy low-fidelity data to generate accurate predictions and uncertainty quantification, which are critical in various fields. For instance, engineering and scientific modeling increasingly rely on MF regression to navigate the trade-off between expensive HF simulation (e.g., physical experiments or time-consuming computational fluid dynamics CFD simulations) and cheap LF analytical parametric modeling with lower accuracy (Bonfiglio et al., 2018; Feldstein et al., 2020; Romor et al., 2023; Zhang et al., 2024). Another application is hyperparameter optimization (HPO) tasks, where training budgets (epochs) serve as varying fidelities (Eggensperger et al., 2021; Wu et al., 2020; Rakotoarison et al., 2024). However, these practical applications often reside in an extreme data imbalance regime, where HF measurements are so scarce (less than 10% of the total dataset), that common probability regression models struggle to maintain reliable generalization (Xing et al., 2023; Qian et al., 2025; Aretz et al., 2025).

Classical MF regression methods typically rely on Gaussian Process (GP) surrogates using autoregressive (AR) couplings and residual decompositions to transfer knowledge across fidelities (Kennedy & O'Hagan, 2000; Perdikaris et al., 2017; Xing et al., 2021). However, these methods face significant computational and statistical barriers. GP's cubic scaling costs make it less effective as LF datasets grow (Fernández-Godino, 2023; Liu et al., 2020). GP also is prone to overfitting when learning complex kernel hyperparameters from very few HF data (Cutajar et al., 2019). Furthermore, multi-fidelity data are rarely collected on nested input sets in practice. HF measurements are often obtained independently, resulting in disjoint input locations across fidelities (Niu et al., 2024). Most GP-based MF methods rely on nested designs ($X_{\mathrm{HF}} \subseteq X_{\mathrm{LF}}$) to enable stable learning (Kennedy & O'Hagan, 2000; Xing et al., 2023). When this assumption is violated, these models struggle to transfer information reliably across fidelities.

Recent advances in tabular foundation models (TFMs) provide an effective alternative to classification and regression tasks: zero-shot Bayesian inference that uses in-context learning (ICL) to generate posterior predictions in a single forward pass without local parameter updates (Müller et al.,

---

[1]Center for Computational Science and Engineering, Massachusetts Institute of Technology, Cambridge, MA, USA [2]Department of Mechanical Engineering, Massachusetts Institute of Technology, Cambridge, MA, USA. Correspondence to: Rosen Ting-Ying Yu <rosenyu@mit.edu>.

*Proceedings of the $43^{rd}$ International Conference on Machine Learning*, Seoul, South Korea. PMLR 306, 2026. Copyright 2026 by the author(s).

2022). Pre-trained offline on millions of synthetic structured causal priors, TFMs such as TabPFN (Hollmann et al., 2023; 2025) are shown to have an advantage over existing tabular learning models in data-scarce regimes for many regression tasks (Ye et al., 2025) and rank highly on tabular benchmarks (Erickson et al., 2025). These findings motivate our investigation into whether TFMs can address the limitations of existing MF methods and provide state-of-the-art (SOTA) performance under very limited non-nested HF data.

Beyond scalability and model choice, a key shortcoming of many MF methods using multiple surrogates lies in their transfer mechanism. Autoregressive and residual-based formulations typically condition HF corrections on LF mean predictions, assuming that cross-fidelity discrepancies are homoscedastic (Xing et al., 2021; Ravi et al., 2024). However, in realistic settings, LF uncertainty varies strongly across the input space, and residual errors tend to grow in precisely those regions where LF predictions are least certain (Sella et al., 2023; Wilke, 2024). Conditioning only on the mean obscures this structure, motivating the use of uncertainties and distribution-level information, such as variance and quantile predictions, to guide cross-fidelity correction (Zou & Yuan, 2008; Yao et al., 2018; Song et al., 2019).

We introduce FIRE (Fidelity-aware In-context REgression), a training-free framework for MF regression using TFMs with uncertainty-aware autoregressive residual learning. FIRE decomposes multi-fidelity regression into three stages: (i) low-fidelity inference, (ii) residual modeling conditioned on augmented statistical information, and (iii) additive uncertainty propagation, using frozen TFMs for zero-shot regression. To summarize, our contributions include:

1. **Algorithmic:** We introduce distribution-conditioned residual transfer for MF regression: a two-stage inference procedure that conditions the HF correction on LF predictive distributions (mean/variance/quantiles), enabling heteroscedastic and non-Gaussian discrepancy handling without retraining.

2. **Systems:** We show how to turn an off-the-shelf TFM into a training-free MF surrogate via in-context residual learning and bi-level fidelity aggregation that scales across multiple fidelities.

3. **Empirical:** On 31 MF benchmarks, FIRE improves the accuracy–uncertainty–runtime tradeoff and remains robust in extreme HF scarcity (2–5%), where standard GP-based methods struggle. Overall, it achieves the best predictive performance for both non-nested and nested problems.

4. **Datasets:** We collect a comprehensive benchmark of 31 problems, including HPO and engineering tasks, while adding a new CFD-based car simulations dataset

for evaluating MF regression generalization, moving beyond the limited test cases from previous works.

## 2. Background

**Multi-fidelity Regression.** Multi-fidelity regression problem is often defined over an input domain $\mathcal{X} \subset \mathbb{R}^d$, with a hierarchy of $T$ information sources (fidelities), indexed by $t \in \{1, \ldots, T\}$. Each fidelity corresponds to an unknown function $f^{(t)} : \mathcal{X} \to \mathbb{R}$, with increasing accuracy and cost as $t$ increases. For each fidelity $t$, we observe a dataset

$$\mathcal{D}_t = \{(x_i^{(t)}, y_i^{(t)})\}_{i=1}^{N_t}, \quad y_i^{(t)} = f^{(t)}(x_i^{(t)}). \quad (1)$$

In practice, the standard MF imbalance regime is: $N_T \ll N_{T-1} \ll \cdots \ll N_1$, reflecting the high cost of acquiring HF observations (Qian et al., 2025). The objective is to construct a predictor for the HF response $f^{(T)}(x)$ by leveraging information from all available lower-fidelity data.

MF regression algorithms commonly adopt an additive autoregressive formulation (Kennedy & O'Hagan, 2000),

$$f^{(t)}(x) = \rho_t f^{(t-1)}(x) + \delta_t(x), \quad t \in 2, ..., T \quad (2)$$

where $\rho_t \in \mathbb{R}$ is a scaling factor and $\delta_t(x)$ is a residual process representing cross-fidelity discrepancies between fidelity levels $t$. This formulation is commonly adopted by classical GP-based methods to transfer information across fidelities (Perdikaris et al., 2017; Xing et al., 2021). However, these approaches typically assume nested input, and their performance degrades in non-nested regimes where HF data are sparse and misaligned (Wang et al., 2022). Motivated by this challenge, recent work proposes generalized autoregressive formulations that support non-nested data in high-dimensional or infinite-fidelity settings (Xing et al., 2023). Other lines of work explore deep learning models for MF regression to improve scalability via learned representations (Yi et al., 2024; Niu et al., 2024; Taghizadeh et al., 2025). In this work, we uses zero-shot TFM and distribution conditioning to improve upon previous avenue.

**Tabular Foundation Models (TFMs).** TFMs have demonstrated the ability to achieve strong regression performance by leveraging in-context learning (Erickson et al., 2025), in which predictions for query inputs are produced by conditioning on a set of context datasets. While FIRE is TFM agnostic, to demonstrate its efficacy, we leverage one of the current SOTA TFM models on tabular regression benchmarks, Prior-data Fitted Networks, and especially the TabPFN model (Hollmann et al., 2023; 2025; Grinsztajn et al., 2025). TabPFN is a transformer-based model pre-trained on millions of pre-trained on synthetic datasets $D_{\text{train}} = \{(x_i, y_i)\}_{i=1}^M$ from randomly generated structural causal models by maximizing the likelihood $q(y|x, D_{\text{train}})$ (Hollmann et al., 2025). Given an in-context

dataset $D_{\text{context}}$ during inference, TabPFN estimates the posterior predictive distribution of the target $y_*$ at the input location $x_*$ as $p(y_*|x_*) \approx q_\theta(y_* \mid x_*, \mathcal{D}_{\text{context}})$.

Transformer-based architectures are particularly well-suited to structured inputs, including discrete variables. Recent work has exploited this property to encode fidelity as an input token in TFMs for multi-fidelity HPO (Lee et al., 2025; Rakotoarison et al., 2024) and shown promising result. The results motivate the use of fidelity token without architectural modification in MF regression setting beyond HPO. However, as we show later in Appendix A, the token is not sufficient in imbalanced data settings.

**Heteroscedasticity and Distributional Stacking.** Heteroscedasticity refers to input-dependent variability, where predictive uncertainty varies across the domain and cannot be captured by a constant noise assumption (Kersting et al., 2007; Lázaro-Gredilla & Titsias, 2011). In multi-fidelity regression, such effects arise in cross-fidelity discrepancies: regions where LF models are less reliable often exhibit larger and more variable residuals (Sella et al., 2023; Wilke, 2024). Standard autoregressive frameworks often fail to account for this, as they use mean-only couplings that assume LF reliability is uniform across the domain. While weighted architectures and warping have been proposed to model input-dependent noise, they frequently rely on Gaussian assumptions that fail to capture skewed or heavy-tailed distributions common in complex real-world systems (Colombo et al., 2025). A few results on predictive stacking indicate that leveraging full predictive distributions is more robust than aggregating point estimates when models are imperfect (Yao et al., 2018). This motivates our model to incorporate conditioning HF corrections on distributional summaries (mean, variance, and quantiles) rather than on the low-fidelity mean alone (Angrist et al., 2006; Zou & Yuan, 2008).

## 3. Proposed Method: FIRE

FIRE decomposes zero-shot multi-fidelity regression learning into three stages: (1) low-fidelity inference, (2) distribution-conditioned residual learning, and (3) prediction and uncertainty quantification. This architecture has a few similarities to autoregressive MF decompositions, but avoids explicit parametric assumptions and repeated retraining by using ICL. Instead, all stages operate in a training-free, inference-time manner using pre-trained TFMs. An overview of the FIRE algorithm is shown in Figure 1 and Algorithm 1.

### 3.1. Bi-Level Fidelity Representation

Building upon the general $T$-fidelity hierarchy introduced in Section 2, FIRE employs a unified two-level representation

---

**Algorithm 1** FIRE

1: **Input:**
   Training data:
   $\mathcal{D}_{\text{LF}} = \bigcup_{t=1}^{T-1} \{([x_i, t], y_i)\}_{i=1}^{N_t} = \{(\mathbf{x}_{\text{LF}}, \mathbf{y}_{\text{LF}})\}$ ;
   $\mathcal{D}_{\text{HF}} = \{([x_j, T], y_j)\}_{j=1}^{N_{\text{HF}}} = \{(\mathbf{x}_{\text{HF}}, \mathbf{y}_{\text{HF}})\}$;
   Testing data: $\mathbf{x}_* = \{[x_p, T]\}_{p=1}^{N_{\text{test}}}$
   Two TFM models: $f_\theta$ base, $\delta_\phi$ residual.
   Quantile levels: $\mathcal{Q} = \{0.1, 0.2, \dots, 0.9\}$;
2: **In-context Learning Phase:**
3:   Fit LF base model $f_\theta$ on $\mathcal{D}_{\text{LF}}$
4:   $\mu_\theta(\mathbf{x}_{\text{HF}}) \leftarrow \mathbb{E}[f_\theta(\mathbf{x}_{\text{HF}})]$;
     $\sigma_\theta^2(\mathbf{x}_{\text{HF}}) \leftarrow \text{Var}[f_\theta(\mathbf{x}_{\text{HF}})]$;
     $q_\theta^{(\tau)}(\mathbf{x}_{\text{HF}}) \leftarrow \text{Quantile}(f_\theta(\mathbf{x}_{\text{HF}}), \tau)$ for $\tau \in \mathcal{Q}$
5:   Statistical augmentation:
     $z_{\text{aug}} \leftarrow [\mathbf{x}_{\text{HF}}, \mu_\theta(\mathbf{x}_{\text{HF}}), \sigma_\theta^2(\mathbf{x}_{\text{HF}}), \{q_\theta^{(\tau)}(\mathbf{x}_{\text{HF}})\}_{\tau \in \mathcal{Q}}]$
6:   Compute residual: $r \leftarrow \mathbf{y}_{\text{HF}} - \mu_\theta(\mathbf{x}_{\text{HF}})$
7:   $\mathcal{D}_{\text{aug}} \leftarrow \{(z_{\text{aug}}, r)\}$
8:   Fit residual model $\delta_\phi$ conditioned on $\mathcal{D}_{\text{aug}}$
9: **Prediction Phase:**
10:   $\mathbf{z}_{\text{aug}}^* \leftarrow [\mathbf{x}_*, \mu_\theta(\mathbf{x}_*), \sigma_\theta^2(\mathbf{x}_*), \{q_\theta^{(\tau)}(\mathbf{x}_*)\}_{\tau \in \mathcal{Q}}]$
11:   $\mu_\phi(\mathbf{z}_{\text{aug}}^*) \leftarrow \mathbb{E}[\delta_\phi(\mathbf{z}_{\text{aug},*})]$;
      $\sigma_\phi^2(\mathbf{z}_{\text{aug}}^*) \leftarrow \text{Var}[\delta_\phi(\mathbf{z}_{\text{aug}}^*)]$;
12: **Return Output:**
    $\hat{y}(\mathbf{x}_*) \leftarrow \mu_\theta(\mathbf{x}_*) + \mu_\phi(\mathbf{z}_{\text{aug}}^*)$;
    $\hat{\sigma_*^2} = \sigma_\theta^2(\mathbf{x}_*) + \sigma_\phi^2(\mathbf{z}_{\text{aug}}^*)$

---

to facilitate in-context learning. We aggregate all available lower-fidelity sources $t \in \{1, \dots, T-1\}$ into a singular LF dataset,

$$\mathcal{D}_{\text{LF}} = \bigcup_{t=1}^{T-1} \{([x_i^{(t)}, t], y_i^{(t)})\}_{i=1}^{N_t}.$$

Here, the fidelity index $t$ is treated as a integer categorical feature, allowing the TFM to learn a shared representation across diverse noise regimes. Specifically, we treat the highest fidelity $T$ as the target (high-fidelity, HF) level at which predictions are ultimately evaluated:

$$\mathcal{D}_{\text{HF}} = \{([x_j^{(T)}, T], y_j^{(T)})\}_{j=1}^{N_T}.$$

This reduction reflects the practical setting in which multiple auxiliary fidelities are available, but only the highest fidelity is queried at test time, exploiting TFM's capability for discrete data modeling in a single forward pass. Empirical evidence on the advantage of our bi-level model over the standard recursive one surrogate model per fidelity level is detailed in Appendix A.1.

### 3.2. Low-Fidelity Inference

Let $f_\theta$ denote a probabilistic TFM pre-trained offline on a large collection of synthetic regression tasks with frozen weight $\theta$. At inference time, $f_\theta$ performs in-context learning

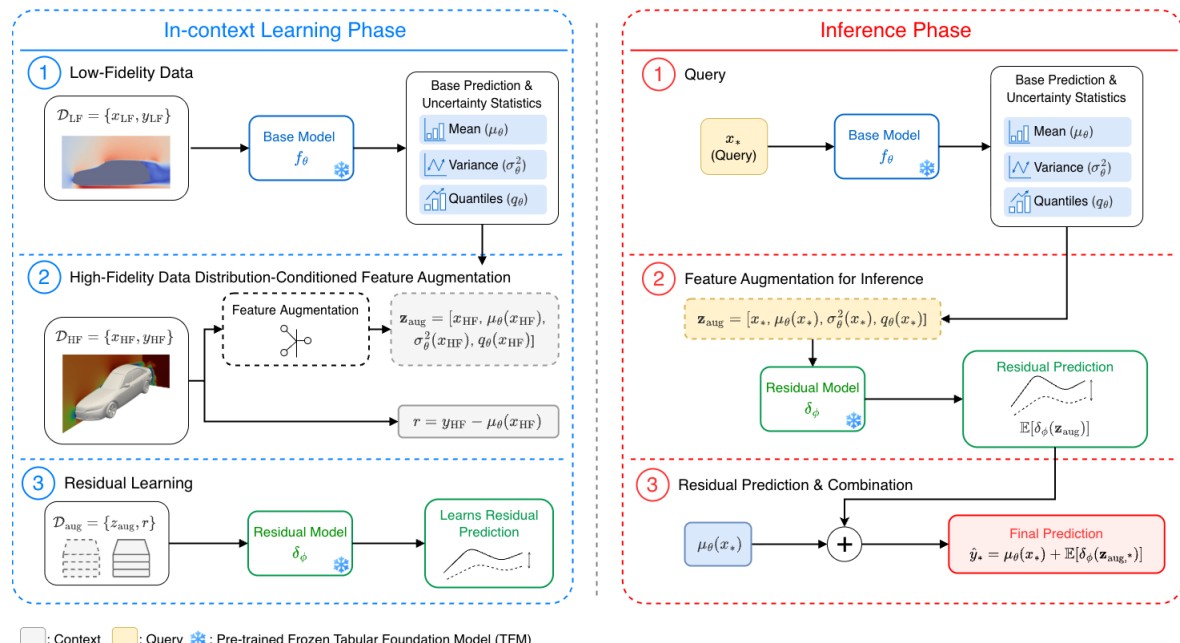

*Figure 1.* **Overview of FIRE.** The method decomposes multi-fidelity regression into a low-fidelity base inference and a residual correction step, conditioned on the base model's statistical and distributional information.

on the aggregated low-fidelity dataset $\mathcal{D}_{\mathrm{LF}}$. Given a high-fidelity input $x_{\mathrm{HF}}$, the model outputs a predictive distribution

$$f_\theta(x_{\mathrm{HF}}) \sim q_\theta(y \mid x_{\mathrm{HF}}, \mathcal{D}_{\mathrm{LF}}),$$

which we summarize using: (1) the predictive mean $\mu_\theta(x)$, (2) the predictive variance $\sigma_\theta^2(x)$, and (3) a finite set of conditional quantiles $\{q_\theta^{(\tau)}(x)\}_{\tau \in \mathcal{Q}}$, where $\mathcal{Q} = \{0.1, 0.2, \ldots, 0.9\}$ (TabPFN's default quantile set). These quantities serve as lightweight summaries of the low-fidelity predictive distribution and are treated as epistemic uncertainty estimates induced by in-context Bayesian inference.

### 3.3. Distribution-Conditioned Residual Learning

FIRE adopts the autoregressive structure in equation implicitly by learning a residual correction at the highest fidelity level. For each high-fidelity training point $x_{\mathrm{HF}}$, we define the residual

$$r = y_{\mathrm{HF}} - \mu_\theta(x_{\mathrm{HF}}).$$

To model this residual, we construct an augmented input vector

$$z_{\mathrm{aug}} = [x_{\mathrm{HF}}, \mu_\theta(x_{\mathrm{HF}}), \sigma_\theta^2(x_{\mathrm{HF}}), \{q_\theta^{(\tau)}(x_{\mathrm{HF}})\}_{\tau \in \mathcal{Q}}].$$

This augmentation encodes: (1) a point prediction of the lower-fidelity signal, (2) uncertainty magnitude through predictive variance, (3) distributional shape through predictive quantiles. The augmented residual dataset is

$$\mathcal{D}_{\mathrm{aug}} = \{(z_{\mathrm{aug}}, r)\}.$$

A second TFM $\delta_\phi$ (with same weight $\phi \equiv \theta$) is then used to perform in-context learning on $\mathcal{D}_{\mathrm{aug}}$, producing a predictive distribution for the residual:

$$\delta_\phi(z_{\mathrm{aug}}) \sim q_\phi(r \mid z_{\mathrm{aug}}, \mathcal{D}_{\mathrm{aug}}).$$

### 3.4. Prediction and Uncertainty Quantification

Given a test input $x_*$ at the highest fidelity, prediction proceeds as follows:

1. Low-fidelity inference: Evaluate the low-fidelity model conditioned on $\mathcal{D}_{\mathrm{LF}}$ to obtain

$$\mu_\theta(x_*), \quad \sigma_\theta^2(x_*), \quad \{q_\theta^{(\tau)}(x_*)\}_{\tau \in \mathcal{Q}}.$$

2. Feature augmentation: Construct the augmented input

$$z_{\mathrm{aug}}^* = [x_*, \mu_\theta(x_*), \sigma_\theta^2(x_*), \{q_\theta^{(\tau)}(x_*)\}_{\tau \in \mathcal{Q}}].$$

3. Residual correction: Predict the residual using $\delta_\phi$ and form the final prediction and uncertainty

$$\hat{y}_* = \mu_\theta(x_*) + \mathbb{E}[\delta_\phi(z_{\mathrm{aug}}^*)], \quad \hat{\sigma}_*^2 = \sigma_\theta^2(x_*) + \sigma_\phi^2(z_{\mathrm{aug}}^*)$$

where $\sigma_\phi^2$ denotes the predictive variance of the residual model. The theoretical justification of this uncertainty formulation is detailed in Appendix C.3.

# 4. Experiment

We compare FIRE against seven SOTA GP-based multi-fidelity baselines on a diverse benchmark suite of 31 multi-fidelity problems that can be categorized into three distinct domains, testing each algorithm's generalizability. Details on the benchmark datasets and algorithm implementation are in Appendices D and E.

## 4.1. Algorithms

**Baseline Algorithms.** We compare against 7 MF regression baselines. The standard GP-based autoregressive methods **AR(1)** (Kennedy & O'Hagan, 2000), **ResGP** (Xing et al., 2021), **NARGP** (Perdikaris et al., 2017), infinite-fidelity methods **ContinuAR** (Xing et al., 2023), and multi-information source **MisoKG** (Poloczek et al., 2017). We also include deep learning baselines **MFBNN** (DNN-LR-BNN) (Yi et al., 2024) and **MFRNP** (Niu et al., 2024).

**FIRE's TFM.** We select **TabPFNv2.5** (Grinsztajn et al., 2025) as the model for FIRE due to its top-tier performance in the tabular benchmark even without tuning. We assess FIRE's generalization with different TFMs Appendix A.4.

## 4.2. Benchmark Datasets Design

**Synthetic Benchmarks (18 tasks).** We include 11 standard two-fidelity MF benchmarks from the MF2 (van Rijn & Schmitt, 2020), two three-fidelity problems from Emukit (Paleyes et al., 2023), and five high-dimensional synthetic tasks (HD suite) with $d \in 10, 20, 30, 40, 50$ input feature dimension (Yi et al., 2024), which are used specifically for our algorithm runtime analysis in Appendix B.

**HPO Datasets (six tasks).** We take six learning curves datasets from LCBench (Zimmer et al., 2021), treating epoch budget as fidelity. We implement them as five-fidelity regression tasks, predicting validation performance at the maximum epoch from partial training curves.

**Physics-based Simulations (seven tasks).** We evaluate on seven physics-based multi-fidelity regression tasks. We take three analytical parametric MF benchmarks from the engineering literature (Yousefpour et al., 2024), and select two Blended Wing Body (BWB) aerodynamic simulated datasets (Sung et al., 2025; 2026). Finally, we create an additional physics-based MF dataset by augmenting the HF CFD data in 3D car aerodynamic prediction using the DrivAerNet dataset (Elrefaie et al., 2025) with a corresponding LF model, and also run the LF modeling for the HF concrete compressive strength benchmark (Yeh, 1998).

**Data Imbalance Protocol.** A critical challenge in multi-fidelity learning is the extreme scarcity of HF data relative to LF data. We systematically evaluate this regime by fixing the LF budget $N_{\text{LF}}$ (set to 200 or 2000 depending on

dimensionality $d$) and varying the HF budget $N_{\text{HF}}$ across six imbalance ratios: $\{2\%, 4\%, 5\%, 10\%, 20\%, 25\%\}$ of $N_{\text{LF}}$. This stress-tests the ability of models to effectively transfer information from abundant low-fidelity sources.

## 4.3. Experiment Protocol & Evaluation Metrics

**Experiment Protocol.** The algorithm evaluation aims to thoroughly compare FIRE to current SOTA MF regression algorithms across problems. For each test problem, our experiment consists of 10 independent trials repeated for a 5-fold outer cross-validation, each trial using a distinct random seed fixed across algorithms. To stress-test model flexibility, we enforce non-nested splits between fidelities. HF points are not co-located with LF points. This reflects realistic scenarios where data sources are independent.

Following Xing et al. (2023)'s protocol of fair benchmarking MF algorithms, all GP-based models are trained with 200 iterations, and all deep learning models with 1000 iterations to reach convergence. All algorithms are implemented in PyTorch and can be GPU-accelerated. All models are used out-of-the-box, without fine-tuning, for a fair comparison of surrogates' generalization across different problems.

Experiments were conducted on a distributed cluster featuring Intel Xeon Platinum 8480+ CPUs and NVIDIA H100 GPUs. To ensure fairness, each run uses identical resources: a single H100 node with 24 CPU cores and 200GB RAM.

**Individual Metrics.** Following existing literature (Cutajar et al., 2019; Ravi et al., 2024; Niu et al., 2024), we use normalized root mean squared error (NRMSE) and negative log likelihood (NLL) for evaluating accuracy and uncertainty quantification ability, respectively, for our main results. In addition, we measure the wall-clock runtime of each algorithm to compare the computational efficiency.

**Aggregated Metrics.** Here we evaluate models using the Elo rating (Elo, 1967) (higher is better), a pairwise comparisons rating that reflects its predicted probability of winning against other models, and average ranking (lower is better). We calibrate 1000 Elo to the performance of MFRNP and perform 100 rounds of bootstrapping to obtain 95% confidence intervals, similar to Erickson et al. (2025)'s approach. Additional metrics evaluations (e.g. normalized score, statitstical ranking) is presented in Appendix B.

# 5. Results

**Overall Performance across Benchmarks.** Figure 2 summarizes performance across all 31 benchmarks. FIRE achieves the highest Elo rating and lowest average rank of both accuracy (NRMSE) and uncertainty (NLL), consistently outperforming both GP-based and deep learning baselines. This performance is driven by our distributional

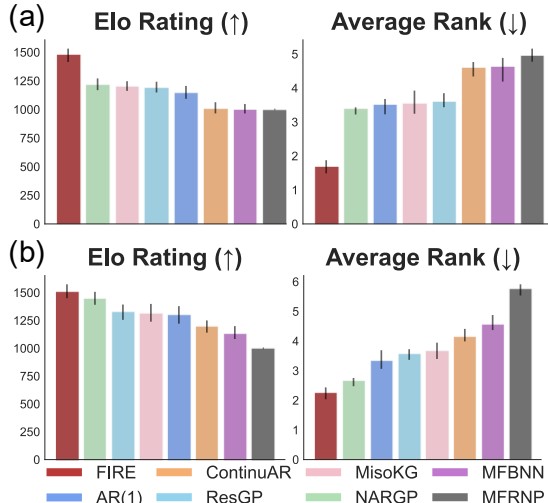

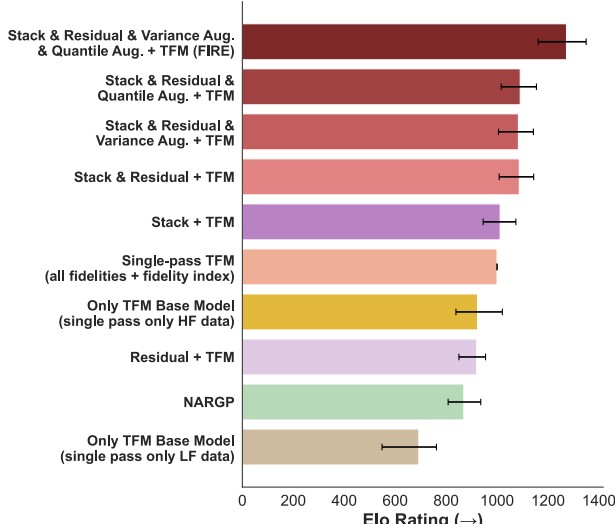

*Figure 2.* **Overall performance.** Elo rating and average rank for (a) accuracy performance (NRMSE) and (b) uncertainty calibration (NLL) across 31 problems. FIRE outperforms the other baselines in both accuracy and uncertainty.

*Figure 4.* **Ablation study of FIRE components.** We compare the Elo rating of various configurations against NARGP, the strongest GP baseline. The full FIRE framework (Top), which combines residual learning with statistical augmentation, significantly outperforms partial implementations.

conditioning strategy: as shown in our ablation study (Figure 4), removing the variance or quantile features from the residual model drops $\approx 250$ Elo rating. This gap highlights that standard mean-only residual couplings used by GP-based methods are insufficient for capturing the complex, input-dependent error structures in these tasks.

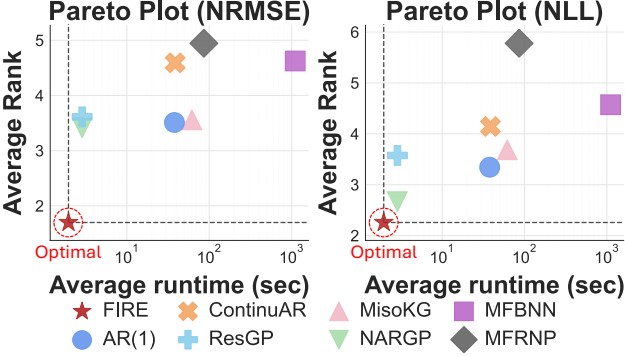

*Figure 3.* **Pareto frontier analysis.** Pareto plots of accuracy (Left) and uncertainty (Right) vs. runtime. FIRE dominates the Pareto frontier for both metrics, being the fastest and most accurate.

**Runtime Efficiency and Pareto Optimality.** We analyze the trade-off between predictive quality and computational cost in Figure 3, which plots the average rank of each method against its wall-clock runtime. FIRE is the fastest and dominates the Pareto frontier for both accuracy (NRMSE) and uncertainty quantification (NLL), residing in the optimal low-rank, low-latency region.

**Robustness under Extreme Data Imbalance.** We next analyze performance as a function of the HF to LFsample ratio ($N_{\text{HF}}/N_{\text{LF}}$), a defining challenge in MF regression. As

detailed in Figure 5, this stability holds across domains of problems. Figure 6 shows that FIRE maintains a clear advantage when HF data is extremely scarce (2–5%), where GP-based and deep learning baselines degrade. The performance gap is the largest on real-world physics simulations and HPO tasks, confirming FIRE's ability to effectively transfer information without overfitting.

## 6. Discussion

FIRE is faster and more accurate than current baselines, standing out when high-fidelity data is very scarce (2-5%). This suggests that large-scale pre-training on diverse synthetic priors provides a strong inductive bias that compensates for HF data scarcity. In contrast, GP models struggle with kernel settings here, and deep models need extensive retraining. Our benchmarks also reveal a generalization gap in existing literature. While methods like ContinuAR and MFRNP excel in modeling spatially correlated PDE fields, they struggle to generalize to the heterogeneous feature spaces of tabular engineering and HPO problems. To disentangle the FIRE decomposition from the TFM backbone, we also instantiate FIRE with GP surrogates and observe consistent gains over standard MF GPs (Appendix A.2). We further ablate the role of in-context learning by swapping the base/residual learners, showing a clear drop when ICL is removed. We also show that FIRE can be generalized to different TFMs (Appendix A.4), and using post-hoc ensembling could help improve FIRE's performance (Appendix A.7).

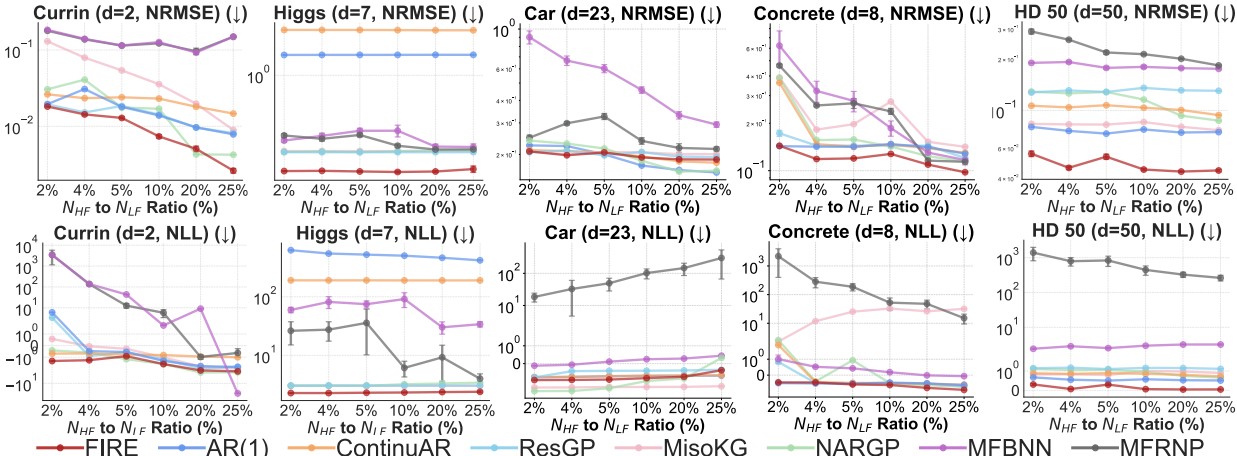

*Figure 5.* Performance (NRMSE and NLL) for representative synthetic (Currin, HD 50), physics-based (Car, Concrete), and HPO (Higgs) tasks. FIRE maintains top-ranked accuracy and uncertainty calibration in the extreme low-data regime (2–5% HF ratio), where baseline performance degrades.

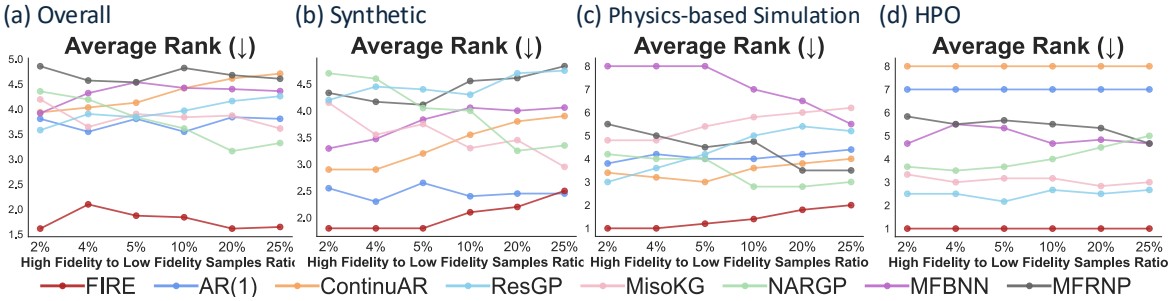

*Figure 6.* **Performance under varying data imbalance.** Aggregated results for problem domain as a function of the High-Fidelity to Low-Fidelity sample ratio ($N_{\text{HF}}/N_{\text{LF}}$ %): (a)Overall (all 31 problems), (b)Synthetic, (c)Engineering, (d)HPO. FIRE demonstrates superior robustness compared to other baselines when HF data is extremely scarce ($2\% - 5\%$), confirming the efficacy of TFM's zero-shot adaptation, especially for the real-world physics simulation and HPO tasks.

## 7. Theoretical Analysis

Section 5 and 6 empirically demonstrate FIRe's strong performance. We now motivate FIRE with theory, showing that TFMs act as valid zero-shot models for Bayesian inference and that conditioning HF residuals on LF base model's distributional summaries is theoretically principled.

### 7.1. TFMs as Bayesian Posterior Approximations

We model the low-fidelity predictor as an amortized approximation to Bayesian posterior inference. Specifically, we adopt an approximation-quality assumption from the theoretical and statistical foundations of TFMs (Gordon et al., 2019; Müller et al., 2022; Nagler, 2023), under which the predictive distribution produced by a TFM approximates the Bayesian posterior predictive distribution.

**Assumption 7.1** (TFM Posterior Approximation Quality). Let $\Pi$ denote a task prior over tabular regression functions. The TFM $f_\theta$ trained via in-context learning on synthetic tasks sampled from $\Pi$ produces a predictive distribution

$q_\theta(\cdot|x, D_{\text{LF}})$ that approximates the Bayesian posterior predictive distribution:

$$q_\theta(y|x, D_{\text{LF}}) \approx \pi(y|x, D_{\text{LF}}) = \int p(y|x)\, d\Pi(p|D_{\text{LF}})$$

in the sense of minimizing expected KL divergence $\mathbb{E}_\Pi[\text{KL}(\pi(\cdot|x, D)\|q_\theta(\cdot|x, D))]$ over tasks drawn from $\Pi$.

Importantly, we note that this holds even when the observed task (our multi-fidelity problem) lies outside the pre-training distribution, provided the task prior $\Pi$ has sufficient support (Müller et al., 2022; Nagler, 2023). Under Assumption 7.1, the outputs $\mu_\theta(x)$, $\sigma_\theta^2(x)$, and $\{q_\theta^{(\tau)}(x)\}_\tau$ can be treated as statistically meaningful summaries of epistemic uncertainty rather than heuristic features.

### 7.2. Justification for Distributional Conditioning

To theoretically justify incorporating variance and quantiles into the residual model, we analyze the Bayes risk under different conditioning sets. We compare a baseline "mean-only" strategy against FIRE's augmented strategy

**Theorem 7.2** (Bayes Risk Monotonicity). *Let* $r = y^{(T)} - \mu_\theta(x^{(T)})$ *be the residual. Define the conditioning sets* $\mathcal{Z}_{mean} = \{x, \mu_\theta(x)\}$ *and* $\mathcal{Z}_{aug} = \{x, \mu_\theta(x), \sigma_\theta^2(x), \{q_\theta^{(\tau)}(x)\}_{\tau \in \mathcal{Q}}\}$. *Under the squared error loss, the Bayes risk satisfies:*

$$\mathcal{R}(\mathcal{Z}_{aug}) \leq \mathcal{R}(\mathcal{Z}_{mean})$$

*with equality if and only if* $\mathbb{E}[r \mid \mathcal{Z}_{aug}] = \mathbb{E}[r \mid \mathcal{Z}_{mean}]$ *almost surely.*

*Proof Sketch.* The result follows from the law of total variance and the tower property of conditional expectation (Casella & Berger, 2024). Intuitively, $\mathcal{Z}_{\text{aug}}$ represents a strictly richer information set than $\mathcal{Z}_{\text{mean}}$. In Bayesian inference, conditioning on additional relevant information monotonically reduces (or maintains) the expected posterior variance, thereby lowering the irreducible error. A full formal derivation is provided in Appendix C.2 for full proof.

Theorem 7.2 confirms that adding variance and quantiles does not increase Bayes risk. In finite-sample settings with constrained model classes (including TFMs used in-context), adding features may increase empirical risk due to estimation error or overfitting. Thus, distributional conditioning is a motivation and not a guarantee.

### 7.3. Heteroscedastic Residuals and the Need for Variance Conditioning

We characterize regimes where variance conditioning is strictly necessary for accurate residual modeling. This follows the heteroscedastic GP framework (Kersting et al., 2007; Lázaro-Gredilla & Titsias, 2011)

**Assumption 7.3** (Heteroscedastic Discrepancy). The variance of the cross-fidelity residual $\delta(x) = y^{(T)}(x) - y^{(T-1)}(x)$ depends on the epistemic uncertainty of the low-fidelity model:

$$\text{Var}[\delta(x)|x, D_{\text{LF}}] = g(\sigma_\theta^2(x))$$

for some non-constant, monotone function $g : \mathbb{R}^+ \to \mathbb{R}^+$. That is, regions where the LF model is uncertain exhibit larger or more variable residuals.

In MF settings, LF models are often least reliable in regions where they are most uncertain (see Figure 7). Under Assumption 7.3, a residual model conditioned only on the mean $\mu_\theta(x)$ is misspecified, as it implicitly assumes residual variance is independent of model uncertainty. By conditioning on $\sigma_\theta^2(x)$, FIRE enables the residual model to adapt the magnitude of its correction to the local epistemic uncertainty of the base model. See Appendix C.5 for details.

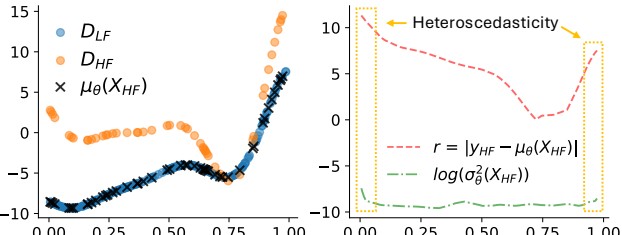

*Figure 7.* **Heteroscedastic cross-fidelity discrepancy on the Forrester**($d = 1$) **benchmark** (right). Regions of larger LF uncertainty $\sigma_\theta^2$ coincide with larger and more variable HF residuals $r$, motivating uncertainty-conditioned residual modeling (left).

### 7.4. Quantiles for Distributional Shape and Misspecification

While variance captures the magnitude of uncertainty (scale), it fails to capture higher-order distributional properties such as skewness, heavy tails, or multi-modality. To address this, FIRE conditions the residual model on a set of predictive quantiles $\mathcal{Q} = \{\tau_1, \ldots, \tau_K\}$, providing a non-parametric embedding of the base predictive distribution.

**Lemma 7.4** (Quantile Embedding). *The quantile function* $q_\theta(\tau; x)$ *uniquely characterizes the predictive distribution* $p(y|x)$ *(Gushchin & Borzykh, 2017). Furthermore, a finite set of quantiles* $\{q_\theta^{(\tau_k)}(x)\}_{k=1}^K$ *acts as a discrete approximation of this embedding, which converges uniformly to the true distribution as* $K \to \infty$ *(see Appendix A.6).*

Conditioning on $\{q_\theta^{(\tau)}(x)\}$ allows the residual model $\delta_\phi$ to exploit distributional mismatch. While Theorem 7.2 establishes that this augmentation is statistically "safe" (it cannot increase Bayes risk), we now characterize the regime where it provides strictly superior performance.

**Assumption 7.5** (Non-Gaussian Residuals). The cross-fidelity residual $r(x)$ exhibits distributional features not captured by its first two moments (e.g., skewness, kurtosis $\neq 3$, or or heavy-tailed).

**Proposition 7.6** (Quantile Conditioning Reduces Risk Under Misspecification). *Let* $\mathcal{Z}_{mv} = \{x, \mu_\theta, \sigma_\theta^2\}$ *and* $\mathcal{Z}_{quant} = \mathcal{Z}_{mv} \cup \{q_\theta^{(\tau)}\}_{\tau \in \mathcal{Q}}$. *If the cross-fidelity residual* $r$ *is non-Gaussian given* $\mathcal{Z}_{mv}$ *(e.g., skewed), then:*

$$R(\mathcal{Z}_{quant}) < R(\mathcal{Z}_{mv}),$$

*where* $R(\cdot)$ *denotes the Bayes risk under squared error loss. (See Appendix A.7 for proof.)*

This inequality captures the key insight: when residuals are non-Gaussian, a common scenario in MF modeling where LF physics approximate HF reality (Perdikaris et al., 2017; Colombo et al., 2025), quantiles encode shape information that variance cannot represent, thereby strictly reducing the irreducible error of the residual model.

# 8. Conclusion, Limitations, and Future work

We presented FIRE, a training-free multi-fidelity regression framework that integrates tabular foundation models with residual and distribution-based uncertainty conditioning to tackle problems with very scarce high-fidelity data. Across 31 benchmark problems, including scalable synthetic functions, hyperparameter optimization data, and engineering simulations, FIRE consistently achieves SOTA performance while maintaining a favorable runtime profile, often outperforming extensive GP or deep learning model retraining from the existing methods.

While FIRE eliminates the need for problem-specific training, it is limited by the TFM's context window, model size, and pre-training quality. Despite eliminating training overhead, FIRE's inference latency is bounded by the quadratic complexity of transformer attention (Qu et al., 2025) too. Consequently, while significantly faster than deep learning baselines, its speed advantage over lightweight GP surrogates is more modest on larger contexts. Looking forward, we plan to expand FIRE to multi-fidelity optimization setting with its outstanding performance in predictive accuracy and uncertainty calibration.

## Acknowledgements

The authors thank Mohamed Elrefaie for providing the car design images of DrivAerNet datasets for Figure 1.

## Impact Statement

This work aims to accelerate scientific discovery and engineering design by improving the efficiency of multi-fidelity regression. By reducing the reliance on expensive high-fidelity simulations (e.g., in computational fluid dynamics or hyperparameter optimization), our method with faster runtime has the potential to lower the computational energy associated with data-intensive scientific modeling. We do not foresee any immediate negative ethical or societal consequences beyond the environmental impact of running inference on foundation models and standard considerations in automating engineering workflows.

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

## Table of Contents for Appendices

# A. Additional Experiment Results and Ablations

### A.1. Why Only Two Surrogates in FIRE? — Ablation on the Number of Surrogates Used in FIRE

A natural alternative to our bi-level strategy is the classical recursive formulation that explicitly models each step in the fidelity hierarchy, requiring $T - 1$ distinct TFMs for a $T$-fidelity problem that commonly used by recursive GP co-kriging (Le Gratiet & Garnier, 2014; Cutajar et al., 2019). We compare this variant against the standard FIRE implementation (2 surrogates) in Figure 8. We observe that adding surrogates provides no distinct advantage in predictive accuracy (Elo), likely because the fidelity token sufficiently conditions the TFM on the source variance. However, the recursive strategy drastically degrades computational efficiency, increasing runtime by over $10\times$. Therefore, we adopt the bi-level approach to maximize computational throughput without compromising the fidelity of the posterior approximation.

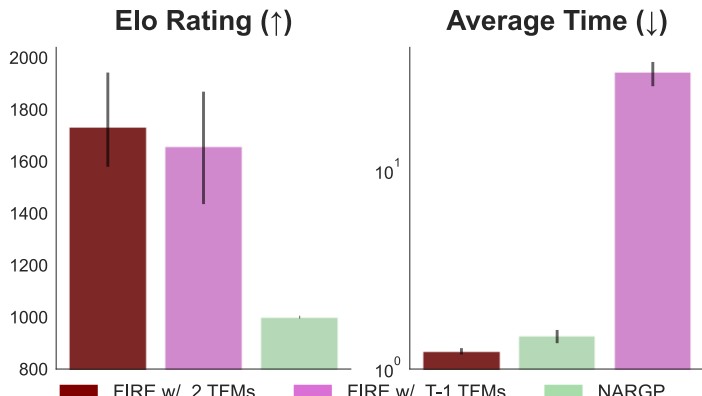

*Figure 8.* **Ablation on the number of surrogates.** We compare the standard bi-level FIRE (using 2 TFMs) against a fully recursive variant modeling every fidelity step (using $T - 1$ TFMs) and the NARGP baseline. The bi-level approach achieves superior trade-offs between accuracy and efficiency, avoiding the prohibitive runtime costs of the recursive strategy.

### A.2. Can FIRE be Applied to GPs? — Result of Applying FIRE Framework to GPs

A key question is whether FIRE's performance stems solely from the Tabular Foundation Model (TFM) or from our distribution-conditioned residual framework. To test this, we instantiate FIRE(GP), replacing the TFM with a standard Gaussian Process to generate the $\mu$ and $\sigma^2$ statistics for the augmented context. As shown in Figure 9, FIRE(GP) significantly outperforms traditional GP-based multi-fidelity methods (e.g., AR(1), NARGP, ResGP). This validates our theoretical analysis in Section 7: conditioning residuals on uncertainty estimates improves performance regardless of the surrogate. However, FIRE(TFM) still outperforms FIRE(GP), confirming that while the framework is effective, the TFM's pre-trained prior offers a distinct advantage over standard kernels in the low-data regime.

### A.3. Why Not Use Existing PFNs for Bayesian Optimization for Multi-fidelity Regression? — Benchmarking against LC-PFN and FT-PFN

We do not use FT-PFN (Rakotoarison et al., 2024) or LC-PFN (Lee et al., 2025) in the main benchmark because they restrict input dimensions to fewer than 10. We validate this decision by comparing their performance on the compatible LCBench dataset in Figure 10. We also include **Single-pass TabPFNv2.5 (all fidelities + fidelity index)** as a baseline to quantify the gain from solely the underlying model. This single-pass approach has all fidelity data as the input, and treats the fidelity index as an ordinary integer categorical feature within the joint dataset.

The results indicate that Single-pass TabPFNv2.5 (all fidelities + fidelity index) is already superior to both FT-PFN and LC-PFN. The specialized models struggle to match the generalization capabilities of the larger, general-purpose foundation model. Single-pass TabPFNv2.5 (all fidelities + fidelity index) achieves better accuracy and ranking without any task-specific adjustments. FIRE further extends this lead by incorporating distribution-conditioned residual learning. This demonstrates that a robust generalist model is preferable to restricted specialist models for these tasks.

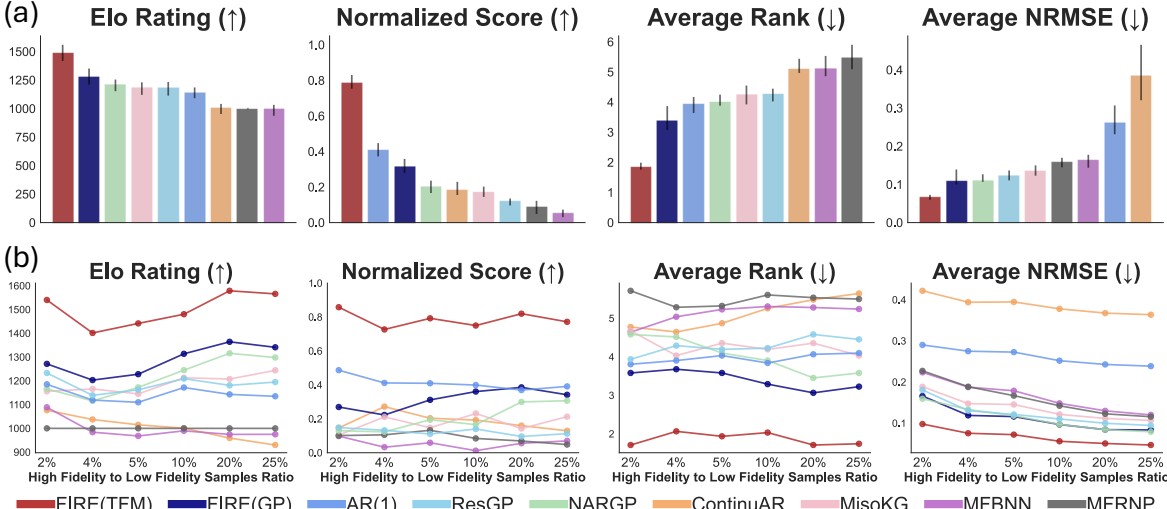

*Figure 9.* **Impact of surrogate choice on performance.** Replacing the TFM with a standard GP (FIRE(GP)) still yields state-of-the-art results compared to existing GP baselines, confirming that our residual learning framework drives significant gains. However, the performance gap between FIRE(TFM) and FIRE(GP) demonstrates the additional value provided by the TFM's in-context prior.

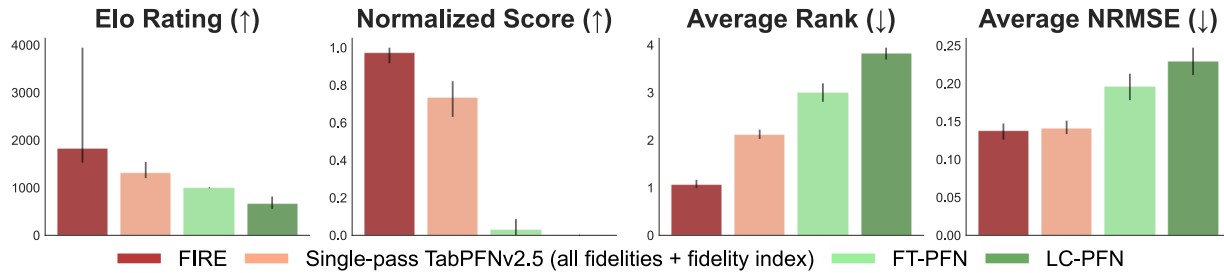

*Figure 10.* **Comparison against specialized HPO TFM surrogates.** We benchmark FIRE against state-of-the-art PFNs explicitly designed for learning curve extrapolation and Bayesian optimization (FT-PFN, LC-PFN) on the LCBench dataset. Despite FIRE using a general-purpose tabular foundation model, it significantly outperforms these specialized architectures in NRMSE and ranks among the top on the 6 HPO datasets in this study.

### A.4. What About Using Other TFMs for FIRE? — Benchmarking FIRE with different TFMs

To assess whether FIRE's performance is driven by the specific choice of foundation model and the algorithm generalization, we instantiate the framework with four different TFMs: RealTabPFN (Garg et al., 2025), TabPFNv2.5, TabPFNv2 (Hollmann et al., 2025), and Mitra (Zhang et al., 2025). As shown in Figure 11, all TFM-based FIRE variants significantly outperform the strongest GP baseline, NARGP, confirming the robustness of the distribution-conditioned residual framework. Among the TFMs, TabPFNv2.5 and RealTabPFN achieve the highest Elo ratings and lowest NRMSE, with RealTabPFN showing a slight edge likely due to its fine-tuning on real-world data distributions. While Mitra outperforms the GP baseline, it lags behind the TabPFN family. We attribute this to the fact that Mitra's current AutoGluon interface lacks native support for direct quantile and variance outputs, necessitating an approximation via a split conformal wrapper that may degrade the quality of the distributional features used for residual conditioning.

### A.5. Does FIRE Maintain SOTA on Nested Data? — Performance Comparison on Nested vs. Non-nested Inputs

All benchmarks in the main text use disjoint (non-nested) input sets to mimic realistic, asynchronous data collection. We investigate whether this protocol unfairly penalizes autoregressive baselines that typically favor nested designs. We compare the performance of FIRE against NARGP, AR(1), and ResGP on both nested ($X_{HF} \subseteq X_{LF}$) and disjoint ($X_{HF} \nsubseteq X_{LF}$) inputs of our synthetic benchmarks, since the real-world simulation and HPO datasets come disjointly. Figure 12 shows the results across all benchmarks.

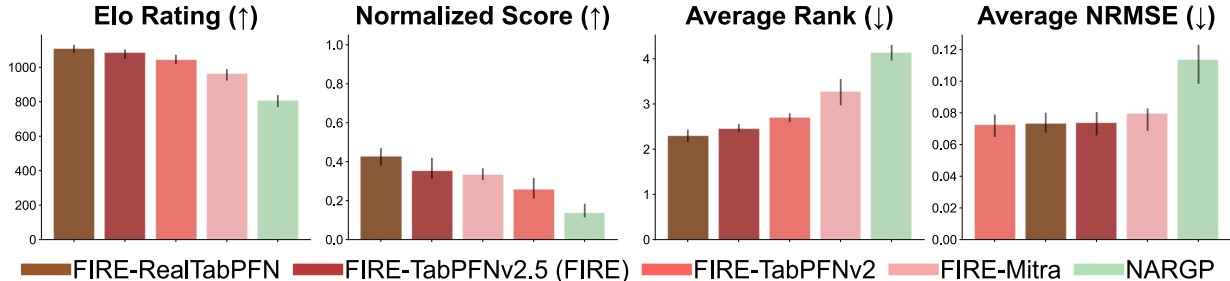

*Figure 11.* **Benchmark of FIRE instantiated with different TFM.** We compare FIRE using RealTabPFN, TabPFNv2.5, TabPFNv2, and Mitra against the strongest GP-baseline (NARGP). While all TFM-based variants outperform the GP baseline, TabPFN-based backbones yield the strongest results.

We observe two key trends. First, GP-based autoregressive methods (NARGP, AR(1), ResGP) consistently perform better in the nested setting, confirming their sensitivity to data alignment. However, even under these ideal conditions, they still lag behind FIRE in both Elo rating and average rank. Second, the performance gap widens significantly in the disjoint regime. While the baselines degrade when inputs are unpaired, FIRE maintains superior performance and stability. This demonstrates that FIRE remains SOTA in the nested setting while offering unique robustness to the irregular, non-nested data structures common in real-world applications.

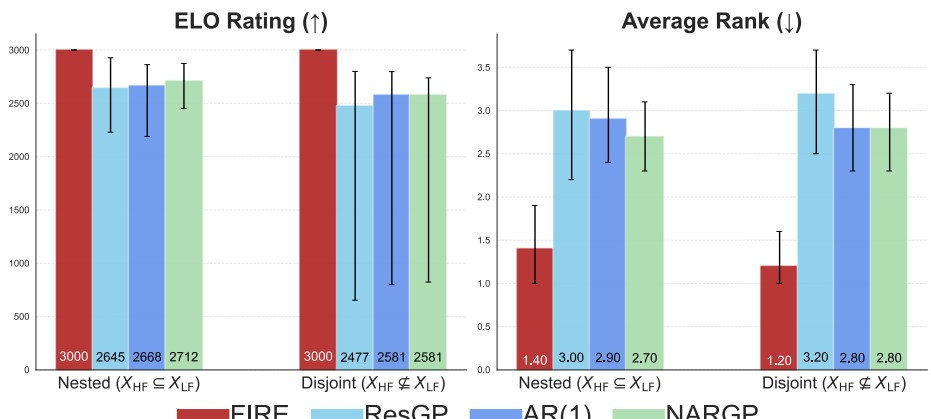

*Figure 12.* **Robustness to non-nested data.** We compare FIRE against autoregressive GP baselines (NARGP, AR(1), ResGP) on nested ($X_{HF} \subseteq X_{LF}$) versus disjoint ($X_{HF} \not\subseteq X_{LF}$) input designs. Elo ratings (left) are calibrated to FIRE=3000. While baselines benefit from nesting, FIRE achieves the highest Elo and lowest rank in both settings, demonstrating superior generalization to disjoint data.

## A.6. Is In-Context Learning (ICL) Necessary? — Ablation of Base and Residual Learner Models

We investigate if FIRE's performance comes from the distributional features or the TFM architecture. We test four combinations of surrogates. We use either a GP or a TFM for the low-fidelity (base) model and the high-fidelity (residual) model. Figure 13 shows the results. We see two trends: First, configurations using a TFM for the residual model consistently outperform those using a GP. This suggests that the TFM's flexible prior handles the complex, non-Gaussian structure of residuals better than a GP, even when both receive the same distributional summaries. Second, using a TFM for the base model (hatched bars) further improves accuracy compared to a GP base model.

## A.7. Can FIRE Be Improved Further? – Enhancing Residual Modeling via Post-Hoc Ensembling

We further investigate if improving the residual model directly translates to better multi-fidelity performance. We apply post-hoc ensembling (PHE) to the residual TFM, a technique introduced in Hollmann et al. (2025) to boost the residual model accuracy. Figure 14 shows that FIRE with PHE outperforms the standard FIRE implementation. This confirms that the quality of the residual prediction is a bottleneck for multi-fidelity regression. However, the improvement comes with a computational cost. The inference time increases by an order of magnitude.

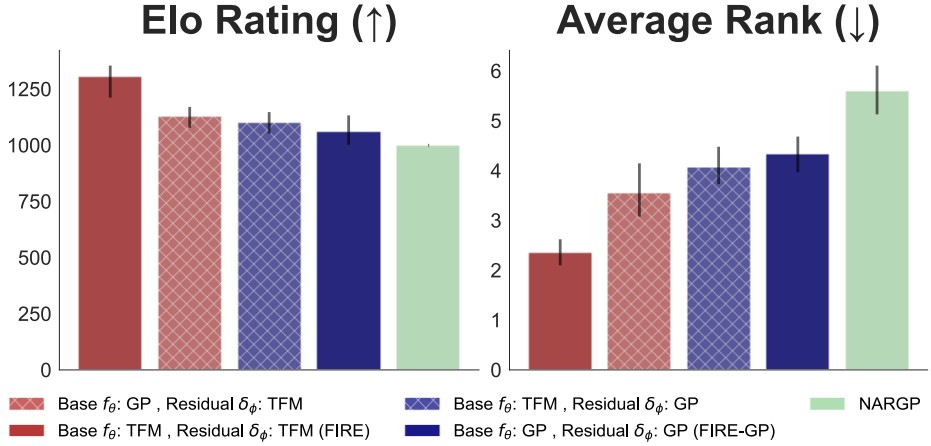

*Figure 13.* **Impact of surrogate choice for base $f_\theta$ and residual $\delta_\phi$ modeling.** We compare combinations of GP and TFM backbones on their accuracy (NRMSE). Red bars indicate a TFM residual model, while navy bars indicate a GP. Hatching indicates a mixture of models in the FIRE framework. The TFM's ability to model residuals via ICL provides a distinct advantage over standard kernels.

Finally, we tested whether training the residual from scratch, rather than relying on the TFM's generic prior, yields a better accuracy perofrmance. Holding the TabPFNv2.5 base fixed, we swap the residual learner for scratch-trained GP, Random Forest, XGBoost, and LightGBM models. Figure 15 shows the TFM residual wins, outperforming GP (1000 vs. 774 Elo), RF (738), and XGBoost (605). This indicates the pre-trained TFM prior, not task-specific fitting, drives FIRE's gains, consistent with TabPFNv2.5 leading tree and gradient boosting methods on TabArena (Erickson et al., 2025).

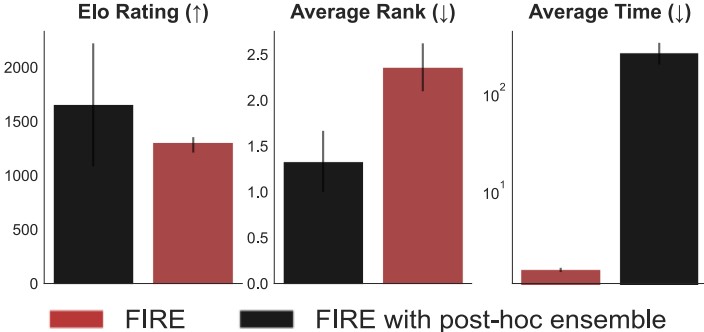

*Figure 14.* **Impact of post-hoc ensembling on FIRE.** Ensembling the residual model improves predictive accuracy (Elo and rank of NRMSE) but increases inference time significantly.

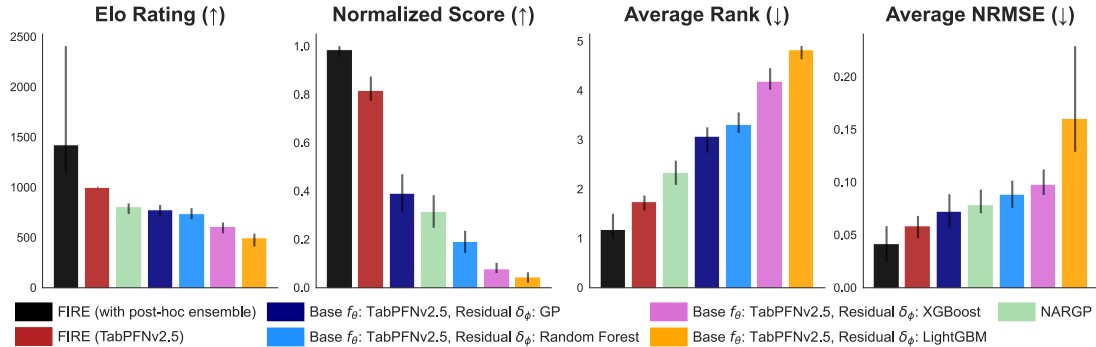

*Figure 15.* **Residual model choice for $\delta_\phi$.** With the TabPFNv2.5 base fixed, we replace the TFM residual with scratch-trained GP, RF, XGBoost, and LightGBM models (Elo calibrated to FIRE=1000). The TFM residual achieves the best accuracy, confirming that training the residual from scratch does not improve FIRE's Pareto frontier.

## A.8. Is the uncertainty truly calibrated? – Uncertainty Quantification (UQ) Analysis with Reliability Diagram and Interval Scores

To verify that FIRE's uncertainty estimates are genuinely calibrated rather than merely useful as features, we evaluate two complementary diagnostics across all benchmarks. The reliability diagram compares nominal coverage to empirical coverage: for each quantile level $p$, we form the predicted Gaussian quantile $\hat{q}_p = \mu + \sigma \, \Phi^{-1}(p)$ from each model's predictive mean and variance, and measure the empirical frequency of test points falling below it. The perfect calibration lies on the diagonal. The Interval Score (IS), computed with the same formulation as Yousefpour et al. (2024), evaluates the central prediction interval by rewarding narrow intervals and penalizing observations that fall outside, with lower scores indicating better-calibrated intervals. As shown in Figure 16, both FIRE variants track the calibration diagonal closely and rank among the top methods on IS. FIRE-GP results in the strongest pure calibration, while FIRE-TabPFNv2.5 remains competitive but slightly less calibrated, consistent with the limitations of current TFM predictive distributions (Nagler, 2023; van Leeuwen, 2025). This supports recommending FIRE-GP for calibration-critical applications and FIRE-TabPFNv2.5 for accuracy- and runtime-driven ones.

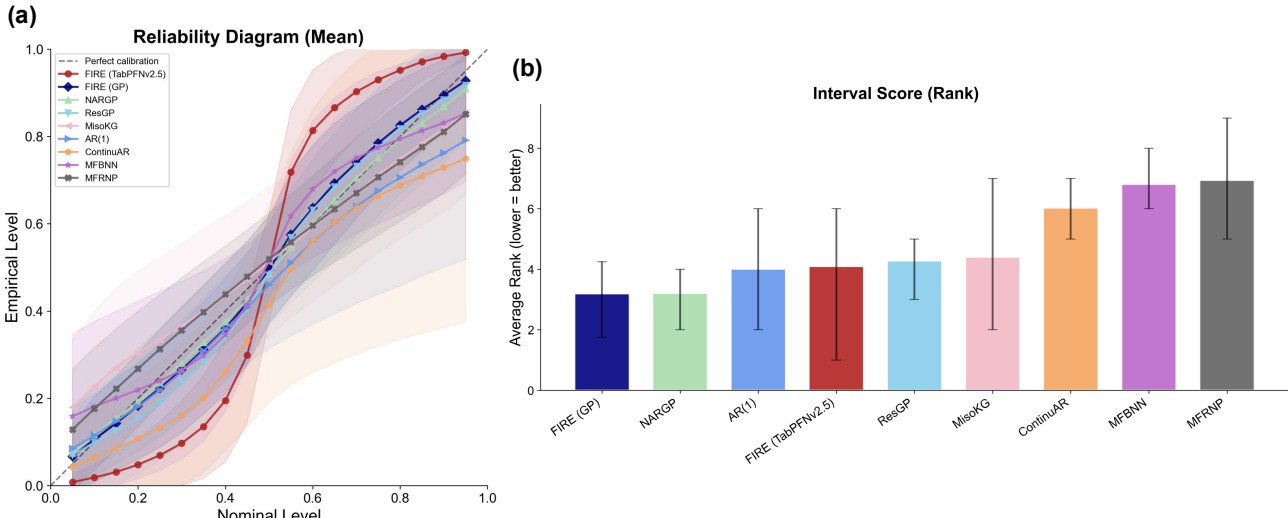

*Figure 16.* **Uncertainty quantification analysis.** (a) Reliability diagram (empirical vs. nominal coverage; the dashed diagonal denotes perfect calibration) and (b) average rank of Interval Score (lower is better) across all benchmarks. Both FIRE variants are well-calibrated and top-ranked on IS, with FIRE-GP achieving the best calibration and FIRE-TabPFNv2.5 remaining competitive.

# B. Full Statistical Analysis and Extended Results

## B.1. Evaluation Metrics and Aggregation

We expand upon the experimental protocol in Section 4.3 by including $R^2$ assessment and detailing our aggregation hierarchy. The individual metric formulations are:

$$\text{NRMSE} = \frac{\sqrt{\frac{1}{N} \sum_{i=1}^{N}(y_i - \hat{y}_i)^2}}{y_{\max} - y_{\min}}$$

$$\text{NLL} = \frac{1}{2N} \sum_{i=1}^{N} \left( \ln(2\pi\sigma_i^2) + \frac{(y_i - \hat{y}_i)^2}{\sigma_i^2} \right)$$

$$R^2 = 1 - \frac{\sum_{i=1}^{N}(y_i - \hat{y}_i)^2}{\sum_{i=1}^{N}(y_i - \bar{y})^2}$$

To ensure fair comparison across diverse problem scales, we utilize four aggregation metrics following Erickson et al. (2025):

1. **Elo Rating**, a rating system for pairwise comparisons, in which a model's score reflects its predicted probability of winning against other models. A 400-point Elo difference corresponds to a 10-to-1 (91%) expected win rate. We calibrate MFRNP to 1000 as the baseline.

2. **Normalized Score**, where the best and median models are scaled to 1.0 and 0.0, respectively, neutralizing scale differences between datasets.

3. **Average Rank**, providing a robust, scale-invariant ordering.

4. **Statistical Ranking**, visualized via **Critical Difference (CD) diagrams** based on the Friedman test and Nemenyi post-hoc analysis ($\alpha = 0.05$).

## B.2. Overall Performance Analysis

**Global Statistical Ranking.** Figure 17 reports the combined performance over 31 benchmark problems under the four aggregation protocols in B.1. FIRE reaches the top Elo rating and Normalized Score on all three error metrics (NRMSE, NLL, and $R^2$). The performance advantage of FIRE is consistent across aggregation schemes and not metric-dependent. Notably, in terms of Average Rank, FIRE maintains a significant lead ($\approx 1.8$) over the strongest GP baseline (NARGP, rank $\approx 3.5$), validating the effectiveness of the TFM prior in diverse tabular settings.

**Robustness to Data Imbalance.** Figure 19 decomposes performance by the high-fidelity to low-fidelity sample ratio ($N_{HF}/N_{LF}$). While strong baselines like NARGP and MFKG degrade rapidly as $N_{HF}$ drops below 5%, FIRE maintains superior stability in extremely data-scarce regimes ($2 - 5\%$). This confirms that the frozen prior of the TFM acts as a robust regularizer, preventing the overfitting common in kernel-based methods when high-fidelity data is sparse.

**Statistical Significance and Win Rates.** To rigorously verify these rankings, Figure 18 presents Critical Difference (CD) diagrams based on the Nemenyi post-hoc test ($\alpha = 0.05$). FIRE is statistically distinguishable from all GP-based baselines in NRMSE, forming a distinct top-tier significance group. This dominance is further corroborated by the pairwise win-rate matrix in Figure 21, where FIRE achieves a positive win rate ($> 60\%$) against every competing baseline, including deep learning methods like MFRNP and MFBNN.

**Runtime Analysis.** Efficiency and Scaling. Figure 11 analyzes algorithmic scalability on the synthetic HD suite ($d = 10$ to 50). Results highlight distinct efficiency tiers: neural methods (MFBNN, MFRNP) are computationally inefficient due to training convergence requirements, while standard GPs form a middle tier bounded by kernel fitting costs. FIRE demonstrates superior efficiency, consistently achieving the lowest runtime due to its zero-shot nature. Although limited by the quadratic attention mechanism of the underlying transformer (Qu et al., 2025), FIRE's zero-shot paradigm still offers a decisive speed advantage for high-dimensional regression tasks.

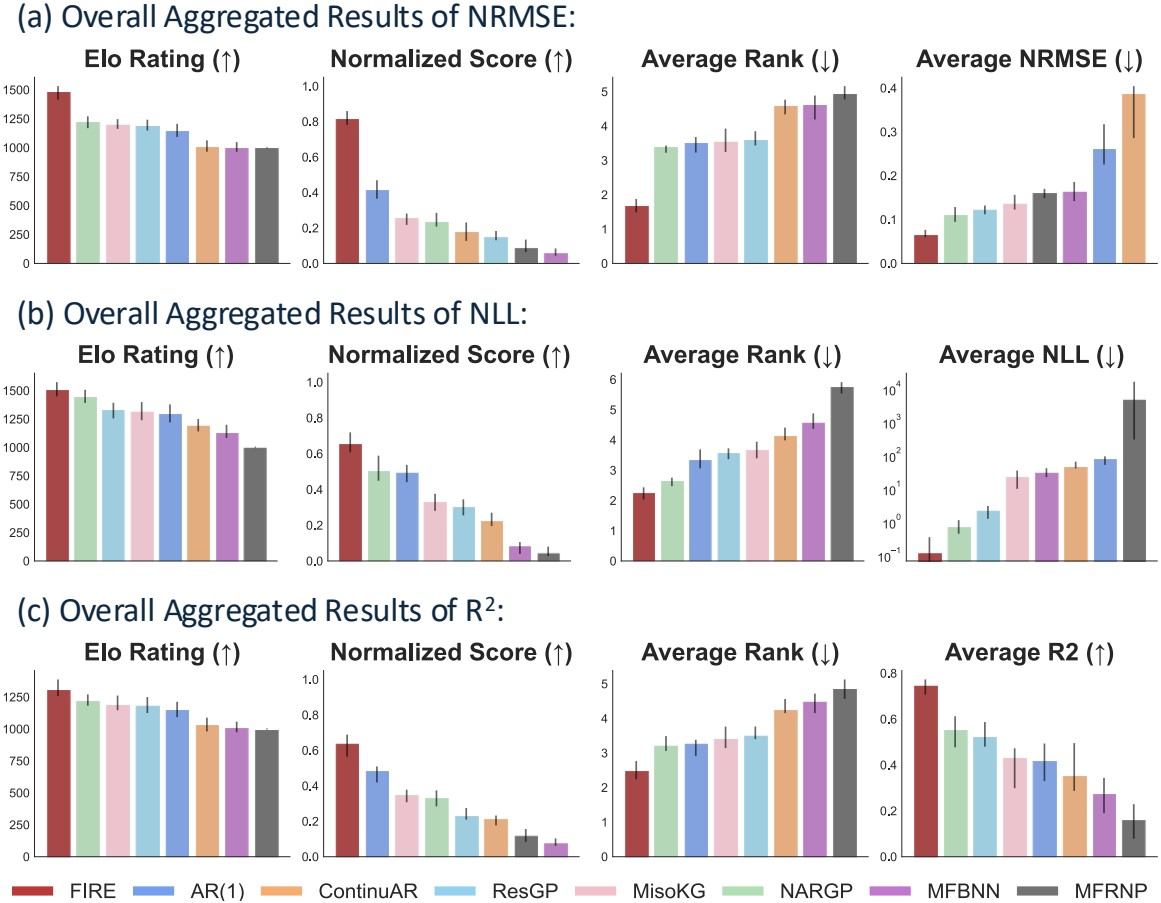

*Figure 17.* **Overall aggregate performance across 31 problems.** Performance is aggregated via Elo, Normalized Score, Average Rank, and raw averages for NRMSE, NLL, and $R^2$. FIRE dominates all aggregation schemes, confirming its status as the state-of-the-art method for accuracy and uncertainty quantification.

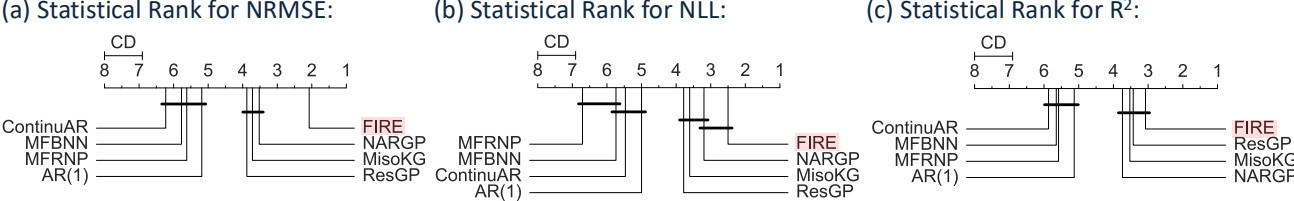

*Figure 18.* **Critical Difference (CD) diagrams** based on the Nemenyi post-hoc test ($\alpha = 0.05$) for (a) NRMSE, (b) NLL, and (c) $R^2$. Methods connected by a horizontal bar are not statistically significantly different. FIRE ranks first statistically across all three metrics, specifically forming a distinct significance group from the GP baselines in accuracy (NRMSE).

## B.3. Performance by Problem Category and Data Imbalance

Figure 12-14 decompose performance across Synthetic, Physics-based, and HPO domains under varying high-fidelity data budgets ($2 - 25\%$):

- **Synthetic Functions:** FIRE exhibits superior robustness in the low-data regime ($< 5\%$), outperforming deep learning baselines in NLL and $R^2$ while remaining competitive with GP methods (AR(1), NARGP, MFKG) on smooth surfaces.

- **Physics-based Simulations:** FIRE dominates engineering benchmarks, achieving the lowest NRMSE and NLL across all ratios and outperforming strong baselines such as ContinuAR, which is designed for solving engineering multi-fidelity problems (e.g., PDE simulations). This confirms that FIRE's distributional conditioning effectively

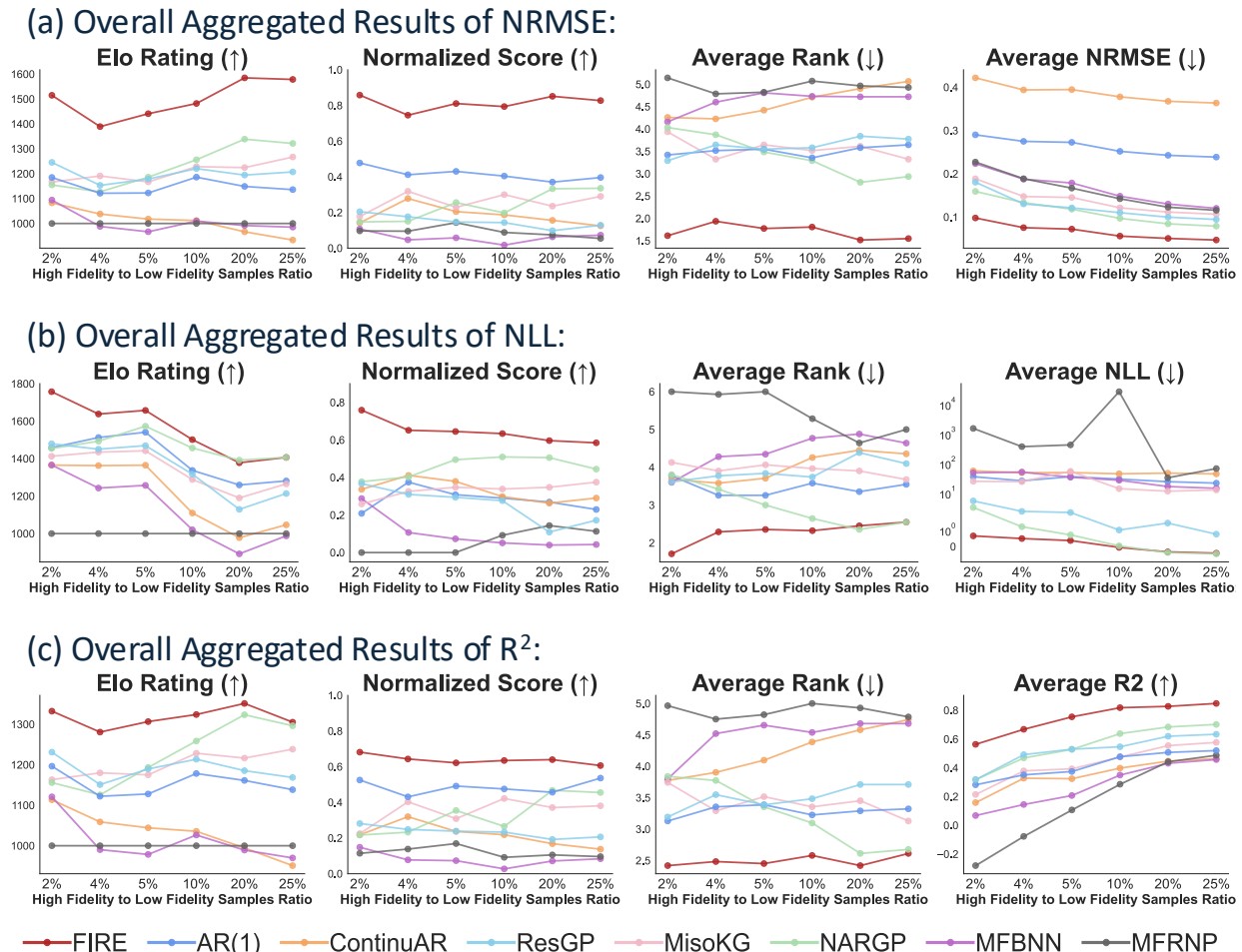

*Figure 19.* **Performance vs. Data Imbalance.** Aggregate metrics (Elo, Normalized Score, Rank, Raw) plotted against the HF/LF sample ratio. FIRE exhibits high stability in low-data regimes (2-5%), significantly outperforming baselines which require larger HF budgets to converge.

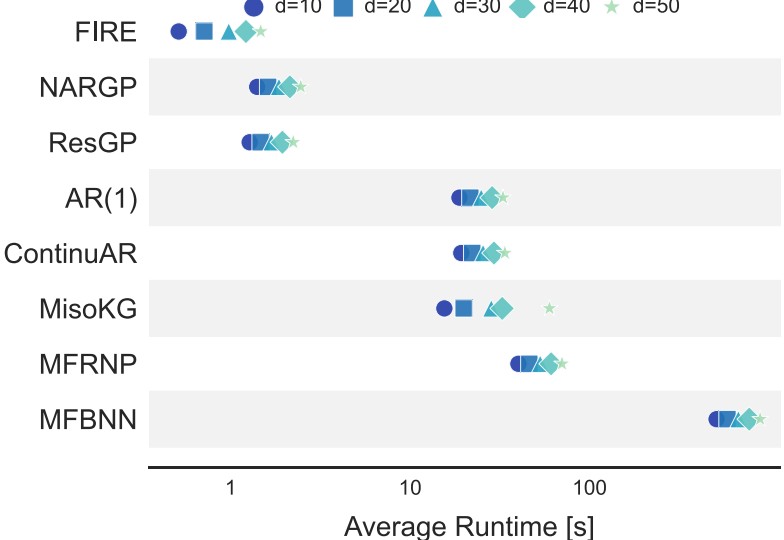

*Figure 20.* **Runtime scaling with input dimensionality.** This figure reports wall-clock runtime on the HD benchmark suite as the input dimension $d$ increases from 10 to 50. FIRE has the lowest runtime at all dimensions. GP and deep learning baselines take longer because they rely on kernel fitting or extensive training.

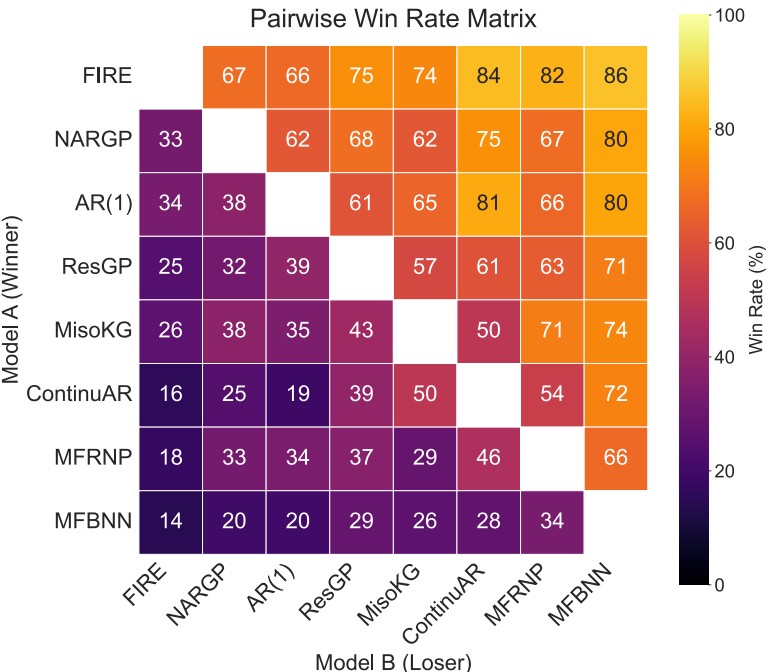

*Figure 21.* **Pairwise win-rate comparison for all problems.** The heatmap displays the percentage of tasks where the algorithm on the Y-axis outperforms the algorithm on the X-axis. FIRE consistently beats all baselines across the majority of tasks (higher numbers are better).

captures the heteroscedastic residuals inherent in complex physical solvers (e.g., CFD), where standard kernels can still suffer from overfitting.

- **HPO Tasks:** On LCBench problems, FIRE consistently ranks the best in all metrics, validating its dominant performance as a surrogate for irregular hyperparameter landscapes. Its stability in the extreme sparsity regime ($2\%$ HF data) highlights its potential for efficient, few-shot multi-fidelity optimization.

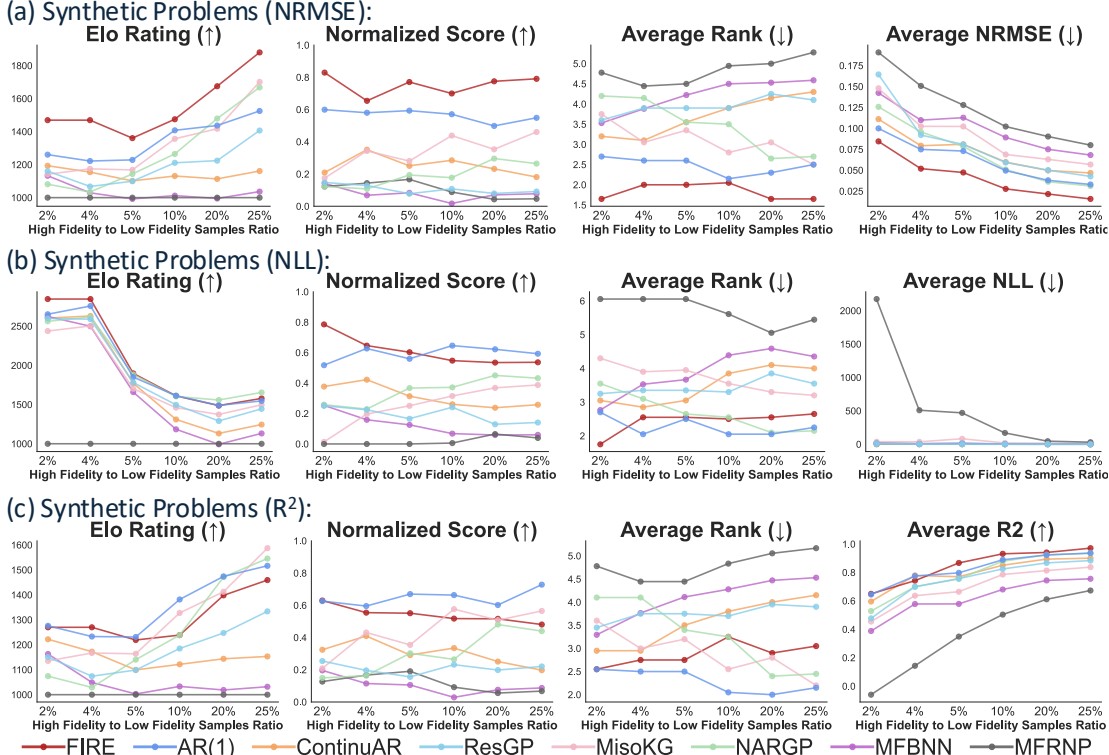

*Figure 22.* **Synthetic Benchmarks: Performance vs. data imbalance.** Aggregated metrics vs. HF/LF ratio for 18 synthetic tasks. While GP-based methods like AR(1) remain competitive, FIRE maintains top-tier performance, particularly in the low-data regime $(2-5\%)$ for NLL and $R^2$.

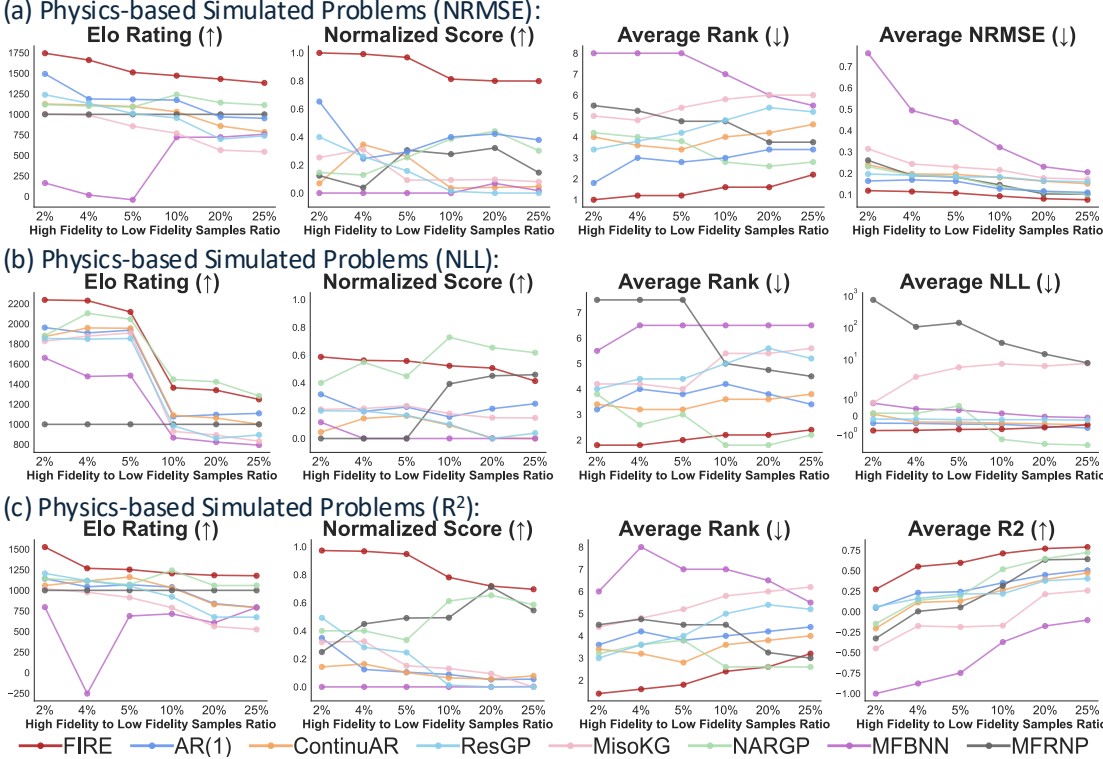

*Figure 23.* **Physics-based Simulations Benchmarks: Performance vs. data imbalance.** Aggregated results for the 7 engineering and physics problems. FIRE demonstrates a clear advantage here, significantly outperforming all baselines in NRMSE and NLL.

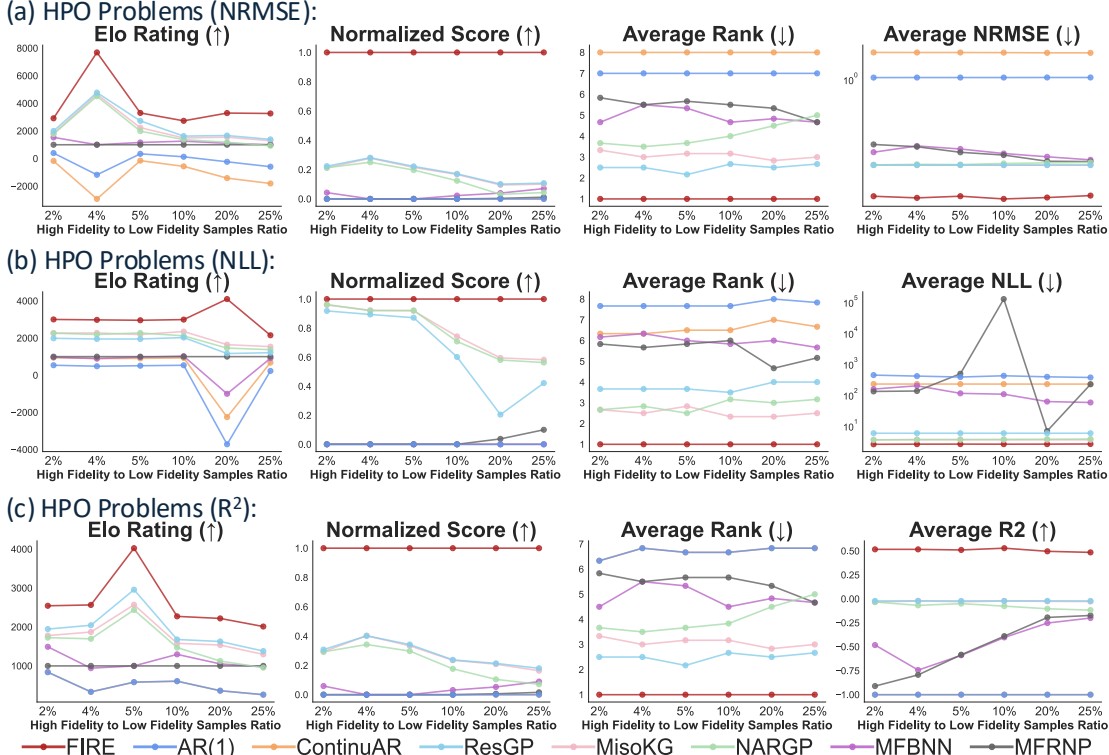

*Figure 24.* **Hyperparameter Optimization (HPO) tasks: Performance vs. data imbalance.** Aggregated results for the 6 LCBench learning curve datasets. FIRE outperforms all baselines, validating its effectiveness in modeling the irregular response surfaces typical of hyperparameter optimization landscapes.

## C. Theoretical Analysis

In this section, we establish the theoretical foundations for FIRE. Our analysis builds upon the statistical framework of Prior-Data Fitted Networks (PFNs) (Müller et al., 2022; Nagler, 2023), while accounting for the unique properties of TabPFN as a surrogate model and the principles of generalized Bayesian stacking (Yao et al., 2018).

### C.1. Formal Statement of Assumption 7.1 (TFM Posterior Approximation)

We ground Assumption 7.1 in the theoretical framework established for Prior-Data Fitted Networks (PFNs) Müller et al. (2022) and Nagler (2023).

**Theorem C.1** (PFN Posterior Approximation; Müller et al. (2022)). *Let $\Pi$ be a task prior over regression functions $f : \mathcal{X} \rightarrow \mathbb{R}$. Let $q_\theta(y|x, D)$ be a parametric model (e.g., a Transformer) trained to minimize the prior-data negative log-likelihood:*

$$\theta^* = \arg\min_\theta \mathbb{E}_{(f,D)\sim\Pi} \left[ -\sum_{(x,y)\in D_{val}} \log q_\theta(y|x, D_{train}) \right]$$

*Then, the learned predictive distribution $q_{\theta^*}$ approximates the true Bayesian posterior predictive distribution $\pi(y|x, D) = \int p(y|x, f)d\Pi(f|D)$ in the sense of minimizing the expected KL divergence:*

$$\mathbb{E}_{D,x} \left[ KL(\pi(\cdot|x, D) \,||\, q_{\theta^*}(\cdot|x, D)) \right]$$

*If the model class $q_\theta$ has sufficient capacity to represent $\pi$, then $q_{\theta^*} = \pi$ almost everywhere (Müller et al., 2022).*

**Regularity Conditions (Nagler, 2023):** While $q_\theta \approx \pi$ is guaranteed by the training objective, the utility of this approximation depends on $\pi$ itself, concentrating around the true data-generating function $p_0$. Nagler (2023) establishes that $\pi$ consistently converges to $p_0$ as $N \rightarrow \infty$, provided:

1. Compactness: The input domain $\mathcal{X}$ is compact and functions $f$ are uniformly bounded.

2. Identifiability: The mapping from functions to data distributions is injective.

3. Prior Support: The prior $\Pi$ assigns positive mass to KL neighborhoods of the true function $p_0$. This ensures that even if the specific physics of a multi-fidelity problem were not seen during pre-training, the model can adapt provided the true function lies within the support of the implicit prior.

**Application to FIRE:**  In the context of FIRE, we define $\Pi$ as the meta-prior induced by the TFM's pre-training scheme (e.g., structural causal models in TabPFN). By virtue of the KL-optimality established above, the TFM outputs conditioned on the low-fidelity data $\mathcal{D}_{LF}$ are not heuristic scores, but statistical estimators of the true posterior moments:

$$\mu_\theta(x) \approx \mathbb{E}_\pi[y|x, \mathcal{D}_{LF}], \quad \sigma_\theta^2(x) \approx \mathrm{Var}_\pi[y|x, \mathcal{D}_{LF}].$$

Consequently, the quantiles $\{q_\theta^{(\tau)}\}$ provide a non-parametric embedding of the posterior shape, validating their use as features for distribution-conditioned residual learning.

### C.2. Proof of Theorem 7.2 (Bayes Risk Monotonicity)

*Proof.* Let $\mathcal{F}_{\mathrm{mean}} = \sigma(\mathcal{Z}_{\mathrm{mean}})$ and $\mathcal{F}_{\mathrm{aug}} = \sigma(\mathcal{Z}_{\mathrm{aug}})$ denote the $\sigma$-algebras generated by the respective conditioning sets. Since $\mathcal{Z}_{\mathrm{mean}} \subset \mathcal{Z}_{\mathrm{aug}}$, we have the filtration $\mathcal{F}_{\mathrm{mean}} \subseteq \mathcal{F}_{\mathrm{aug}}$. The Bayes risk for the squared error loss with respect to a conditioning set $\mathcal{Z}$ is given by the expected conditional variance:

$$\mathcal{R}(\mathcal{Z}) = \mathbb{E}\left[(r - \mathbb{E}[r \mid \mathcal{Z}])^2\right] = \mathbb{E}\left[\mathrm{Var}(r \mid \mathcal{Z})\right].$$

We apply the Law of Total Variance (Casella & Berger, 2024) conditioning on the coarser $\sigma$-algebra $\mathcal{F}_{\mathrm{mean}}$:

$$\mathrm{Var}(r \mid \mathcal{F}_{\mathrm{mean}}) = \mathbb{E}\left[\mathrm{Var}(r \mid \mathcal{F}_{\mathrm{aug}}) \mid \mathcal{F}_{\mathrm{mean}}\right] + \mathrm{Var}\left(\mathbb{E}[r \mid \mathcal{F}_{\mathrm{aug}}] \mid \mathcal{F}_{\mathrm{mean}}\right).$$

Taking the expectation of both sides with respect to the joint distribution $p(x, r)$:

$$\mathbb{E}\left[\mathrm{Var}(r \mid \mathcal{F}_{\mathrm{mean}})\right] = \mathbb{E}\left[\mathrm{Var}(r \mid \mathcal{F}_{\mathrm{aug}})\right] + \mathbb{E}\left[\mathrm{Var}\left(\mathbb{E}[r \mid \mathcal{F}_{\mathrm{aug}}] \mid \mathcal{F}_{\mathrm{mean}}\right)\right].$$

Substituting the definition of Bayes risk $\mathcal{R}$:

$$\mathcal{R}(\mathcal{Z}_{\mathrm{mean}}) = \mathcal{R}(\mathcal{Z}_{\mathrm{aug}}) + \mathbb{E}\left[\mathrm{Var}\left(\mathbb{E}[r \mid \mathcal{Z}_{\mathrm{aug}}] \mid \mathcal{Z}_{\mathrm{mean}}\right)\right].$$

Since variance is non-negative, the second term on the RHS is non-negative. Therefore:

$$\mathcal{R}(\mathcal{Z}_{\mathrm{mean}}) \geq \mathcal{R}(\mathcal{Z}_{\mathrm{aug}}).$$

**Equality Condition:**  Equality holds if and only if $\mathbb{E}\left[\mathrm{Var}\left(\mathbb{E}[r \mid \mathcal{Z}_{\mathrm{aug}}] \mid \mathcal{Z}_{\mathrm{mean}}\right)\right] = 0$. Since variance is non-negative, this requires $\mathrm{Var}\left(\mathbb{E}[r \mid \mathcal{Z}_{\mathrm{aug}}] \mid \mathcal{Z}_{\mathrm{mean}}\right) = 0$ almost surely. This implies that the random variable $\mathbb{E}[r \mid \mathcal{Z}_{\mathrm{aug}}]$ is constant with respect to the finer information given the coarser information, i.e.,

$$\mathbb{E}[r \mid \mathcal{Z}_{\mathrm{aug}}] = \mathbb{E}[\mathbb{E}[r \mid \mathcal{Z}_{\mathrm{aug}}] \mid \mathcal{Z}_{\mathrm{mean}}] = \mathbb{E}[r \mid \mathcal{Z}_{\mathrm{mean}}] \quad \text{a.s.},$$

where the second equality follows from the Tower Property. Thus, augmentation strictly reduces risk unless the residual is conditionally independent of the distributional statistics given the mean. $\square$

### C.3. Variance Decomposition for Two-Stage Uncertainty Quantification

We provide the justification for the additive variance formula $\hat{\sigma}_*^2 = \sigma_\theta^2(x_*) + \sigma_\phi^2(z_{aug,*})$ used in Section 3.4. This formulation draws inspiration from the additive independence structure employed from the ResGP (Xing et al., 2021). We use this independence as a practical approximation, and we acknowledge that correlations may exist.

**Proposition C.2** (Additive Variance). *Let the high-fidelity response be modeled as the sum of a low-fidelity latent variable and a residual latent variable:*

$$y^{(T)}(x) = Y_{LF}(x) + R(x)$$

*where $Y_{LF}(x) \sim q_\theta(\cdot|x, \mathcal{D}_{LF})$ is the predictive distribution of the base TFM, and $R(x) \sim q_\phi(\cdot|z_{aug}, \mathcal{D}_{aug})$ is the predictive distribution of the residual TFM.*

**Assumption C.3** (Conditional Independence). We assume that given the input $x$ and the augmented features $z_{aug}$, the prediction error of the low-fidelity model is uncorrelated with the prediction error of the residual model. That is, $Cov(Y_{LF}, R|x, z_{aug}) = 0$.

*Proof.* The variance of the sum of two uncorrelated random variables is the sum of their variances. By definition of the predictive statistics in Section 3.2 and 3.4:

$$Var[Y_{LF}|x] = \sigma_\theta^2(x), \quad Var[R|z_{aug}] = \sigma_\phi^2(z_{aug})$$

Therefore, the total predictive variance for the high-fidelity query is:

$$\begin{aligned}
Var[y^{(T)}|x, z_{aug}] &= Var[Y_{LF} + R|x, z_{aug}] \\
&= Var[Y_{LF}|x] + Var[R|z_{aug}] + 2Cov(Y_{LF}, R|x, z_{aug}) \\
&= \sigma_\theta^2(x) + \sigma_\phi^2(z_{aug})
\end{aligned}$$

$\square$

This validates the additive uncertainty quantification strategy in FIRE, ensuring that the final uncertainty estimate accounts for both the uncertainty in the low-fidelity approximation and the uncertainty in the residual correction.

### C.4. Information-Theoretic Bottleneck of Mean-Only Conditioning

We formalize the claim that mean-only conditioning discards information relevant to the residual.

**Proposition C.4** (Data Processing Inequality for Residuals). *Let $r$ be the high-fidelity residual, $F_{LF}$ the full predictive distribution of the low-fidelity model, and $\mu_{LF}$ its mean. Since $\mu_{LF}$ is a deterministic function of $F_{LF}$, the Data Processing Inequality implies:*

$$I(r; F_{LF}) \geq I(r; \mu_{LF})$$

*Equality holds if and only if $r \perp F_{LF} \mid \mu_{LF}$.*

The inequality above provides an information-theoretic justification for why mean-only surrogates are insufficient in multi-fidelity learning. This result aligns with findings in Bayesian model averaging, specifically the work of Yao et al. (2018), who demonstrate that in "M-open" settings (where the true model is not in the candidate set), combining full predictive distributions yields asymptotically superior performance compared to stacking point estimates (means).

In our context, the residual process $r$ often exhibits heteroscedasticity or non-Gaussianity. The variance and shape information required to model these features is contained in $F_{LF}$ but lost in $\mu_{LF}$. Consequently, $I(r; F_{LF} \mid \mu_{LF}) > 0$, implying that mean-only conditioning creates an information bottleneck that limits the residual model's ability to correct the low-fidelity error. FIRE bypasses this bottleneck by explicitly conditioning on the variance and quantiles, effectively reconstructing a sufficient statistic for $F_{LF}$.

### C.5. Variance-Residual Coupling and Mean-Only Misspecification

We formalize the dependency between low-fidelity uncertainty and high-fidelity residuals, drawing on the framework of heteroscedastic Gaussian processes (Lázaro-Gredilla & Titsias, 2011).

**Proposition C.5** (Variance-Residual Coupling). *Let the cross-fidelity residual be $\delta(x) = y^{(T)}(x) - \mu_\theta(x)$. We posit that the residual process is heteroscedastic, with a variance structure dependent on the epistemic uncertainty of the low-fidelity model:*

$$Var[\delta(x) \mid x] = g(\sigma_\theta^2(x))$$

*for some monotone increasing function $g : \mathbb{R}^+ \to \mathbb{R}^+$.*

**Corollary C.6** (Mean-Only Misspecification). *Under the assumption above, a residual model $\delta_\phi$ conditioned solely on the mean $\mu_\theta(x)$ is misspecified.*

*Proof.* Let the true residual distribution be $p(\delta|x)$. We analyze the entropy of the predictive distribution under the two conditioning sets: $\mathcal{Z}_{mean} = \{x, \mu_\theta\}$ and $\mathcal{Z}_{aug} = \{x, \mu_\theta, \sigma_\theta^2\}$. By the chain rule for conditional mutual information, we can decompose the entropy of the coarser conditioning set as:

$$H(\delta \mid \mathcal{Z}_{mean}) = H(\delta \mid \mathcal{Z}_{mean}, \sigma_\theta^2) + I(\delta; \sigma_\theta^2 \mid \mathcal{Z}_{mean})$$

Since $\mathcal{Z}_{aug} = \mathcal{Z}_{mean} \cup \{\sigma_\theta^2\}$, the first term is exactly $H(\delta \mid \mathcal{Z}_{aug})$. Rearranging terms yields:

$$H(\delta \mid \mathcal{Z}_{mean}) - H(\delta \mid \mathcal{Z}_{aug}) = I(\delta; \sigma_\theta^2 \mid \mathcal{Z}_{mean})$$

We now show that this mutual information term is strictly positive. Under Proposition A.9, the conditional variance is given by $Var[\delta \mid \mathcal{Z}_{aug}] = g(\sigma_\theta^2)$. In contrast, $Var[\delta \mid \mathcal{Z}_{mean}]$ integrates out $\sigma_\theta^2$ and thus cannot depend on it. Since $g$ is a non-constant function, we have:

$$Var[\delta \mid \mathcal{Z}_{aug}] \neq Var[\delta \mid \mathcal{Z}_{mean}]$$

This inequality in conditional moments implies that the random variable $\delta$ is not conditionally independent of $\sigma_\theta^2$ given $\mathcal{Z}_{mean}$ (i.e., $\delta \not\perp \sigma_\theta^2 \mid \mathcal{Z}_{mean}$). A fundamental property of mutual information is that $I(X; Y \mid Z) = 0$ if and only if $X \perp Y \mid Z$. Therefore:

$$I(\delta; \sigma_\theta^2 \mid \mathcal{Z}_{mean}) > 0$$

Consequently:

$$H(\delta \mid \mathcal{Z}_{mean}) > H(\delta \mid \mathcal{Z}_{aug})$$

$\square$

The mean-only model has strictly higher conditional entropy (uncertainty) regarding the residual than the augmented model. This increase in entropy represents an increase in the irreducible error of the probabilistic model, constituting misspecification.

This result parallels findings in heteroscedastic GP regression, where modeling input-dependent noise variance significantly improves predictive log-likelihood. In our setting, the TFM's predictive variance $\sigma_\theta^2(x)$ serves as a learned proxy for this local noise regime

### C.6. Quantile Embedding Consistency Theorem

We provide the rigorous foundation for Lemma 7.4, establishing that the quantile function acts as a valid distributional embedding. We rely on the duality between integrated distribution functions and integrated quantile functions established by Gushchin & Borzykh (2017).

**Theorem C.7** (Quantile Function as Distributional Representation). *Let $F : \mathbb{R} \to [0,1]$ be a continuous CDF and $Q : [0,1] \to \mathbb{R}$ be its associated quantile function defined as $Q(\tau) = \inf\{y \in \mathbb{R} : F(y) \geq \tau\}$.*

- *Uniqueness (Gushchin & Borzykh (2017), Theorem 3): The integrated quantile function $K_X(u) = \int_0^u Q(s)ds$ is the Fenchel transform of the integrated distribution function $J_X(x) = \int_0^x F(t)dt$. Consequently, $Q(\tau)$ uniquely determines the underlying probability measure.*

- *Finite Approximation: Let $Q_K = \{Q(\tau_1), \ldots, Q(\tau_K)\}$ be a set of quantiles on a grid $\tau_k = k/(K+1)$. As $K \to \infty$, the empirical CDF $\hat{F}_K$ constructed from $Q_K$ converges uniformly to $F$:*

$$\sup_{y \in \mathbb{R}} |\hat{F}_K(y) - F(y)| \to 0.$$

**Application to FIRE:** In FIRE, the TFM outputs quantiles at levels $\mathcal{Q} = \{0.1, \ldots, 0.9\}$ ($K = 9$). By Theorem C.7, these quantiles provide a coarse but faithful representation of the full predictive CDF $F_\theta(\cdot|x, \mathcal{D}_{LF})$. This embedding captures distributional features (skewness, kurtosis) that are orthogonal to the mean and variance, providing the residual model with the necessary context to correct higher-order discrepancies.

## C.7. Proof of Proposition 4.2 (Quantile Risk Reduction)

We show that conditioning on quantiles strictly reduces the Bayes risk (Mean Squared Error) when residuals exhibit non-Gaussian structure.

*Proof.* Let $r = y^{(T)} - \mu_\theta(x)$ be the residual. For any conditioning set $\mathcal{Z}$, the Bayes-optimal predictor under squared error loss is the conditional expectation $\hat{r}_{\mathcal{Z}} = \mathbb{E}[r|\mathcal{Z}]$, and the associated Bayes risk is the expected conditional variance:

$$R(\mathcal{Z}) = \mathbb{E}[(r - \mathbb{E}[r|\mathcal{Z}])^2] = \mathbb{E}[\text{Var}(r|\mathcal{Z})].$$

We compare the risk under $\mathcal{Z}_{mv} = \{x, \mu_\theta, \sigma_\theta^2\}$ and $\mathcal{Z}_{quant} = \mathcal{Z}_{mv} \cup \{q_\theta^{(\tau)}\}_{\tau \in \mathcal{Q}}$. Since $\mathcal{Z}_{mv} \subset \mathcal{Z}_{quant}$, the Law of Total Variance applied to $r$ yields:

$$\text{Var}(r|\mathcal{Z}_{mv}) = \mathbb{E}[\text{Var}(r|\mathcal{Z}_{quant}) \mid \mathcal{Z}_{mv}] + \text{Var}(\mathbb{E}[r|\mathcal{Z}_{quant}] \mid \mathcal{Z}_{mv}).$$

Taking expectations over $\mathcal{Z}_{mv}$ gives the risk decomposition:

$$R(\mathcal{Z}_{mv}) = R(\mathcal{Z}_{quant}) + \mathbb{E}[\text{Var}(\mathbb{E}[r|\mathcal{Z}_{quant}] \mid \mathcal{Z}_{mv})].$$

Since variance is non-negative, $R(\mathcal{Z}_{mv}) \geq R(\mathcal{Z}_{quant})$. Strict inequality holds if and only if $\text{Var}(\mathbb{E}[r|\mathcal{Z}_{quant}] \mid \mathcal{Z}_{mv}) > 0$ with positive probability.

Under Assumption 7.5 (Non-Gaussianity), the residual distribution is not fully characterized by its first two moments. Consequently, the conditional expectation of $r$ given the quantiles, $\mathbb{E}[r|\mathcal{Z}_{quant}]$, is not almost surely constant with respect to $\mathbb{E}[r|\mathcal{Z}_{mv}]$. For example, if the residual distribution is skewed, the median $q^{(0.5)}$ (contained in $\mathcal{Z}_{quant}$) will diverge from the mean (contained in $\mathcal{Z}_{mv}$), implying that the quantiles contain independent information about the residual's central tendency. Thus, the variance term is strictly positive, and $R(\mathcal{Z}_{quant}) < R(\mathcal{Z}_{mv})$. $\square$

## C.8. Efficiency of Quantile Conditioning

The choice of the quantile set $\mathcal{Q}$ in FIRE is motivated by efficiency results from Composite Quantile Regression (CQR). Zou & Yuan (2008) prove that estimating a regression function using a grid of quantiles achieves an Asymptotic Relative Efficiency (ARE) of at least 70% compared to Ordinary Least Squares (OLS), regardless of the error distribution. Importantly, for heavy-tailed distributions (e.g., Double Exponential or $t$-distributions), CQR significantly outperforms OLS (ARE > 100%).

In the context of FIRE, this theory suggests that augmenting the residual model with quantiles $\{q_\theta^{(\tau_k)}\}_{k=1}^K$ provides robustness against misspecification. Even if the low-fidelity model captures the mean trend correctly (where OLS/Mean-only conditioning would suffice), the quantiles allow the residual model to efficiently correct for heavy-tailed or asymmetric errors that characterize the "small-data" multi-fidelity regime. Zou & Yuan (2008) shows the relative efficiency approaches its limit for $K \geq 9$, motivating our use of deciles ($K = 9$), though they recommend $K = 19$ in their regression setting.

## C.9. Excess Risk of Mean-Only Conditioning

We quantify the theoretical penalty incurred by ignoring distributional information (variance and quantiles) in the residual model. We model this as the irreducible approximation error arising from the information bottleneck.

*Setup.* Let $z_{full} = [x, \mu_\theta(x), \sigma_\theta^2(x), q_\theta(x)]$ be the full augmented feature vector used by FIRE, and let $z_{mean} = [x, \mu_\theta(x)]$ be the restricted feature vector used by standard baselines. Let the optimal residual correction functions be $\delta^*(z_{full}) = \mathbb{E}[r|z_{full}]$ and $\hat{\delta}_{mean}(z_{mean}) = \mathbb{E}[r|z_{mean}]$.

**Theorem C.8** (Excess Risk Decomposition). *Let $\mathcal{R}(\hat{f}) = \mathbb{E}[(r - \hat{f}(z))^2]$ be the risk (mean squared error) under the $L_2$ loss. The excess risk incurred by the mean-only model compared to the augmented model is exactly the expected variance of the residual explained by the distributional features:*

$$\mathcal{R}(\hat{\delta}_{mean}) - \mathcal{R}(\delta^*) = \mathbb{E}_x \left[ \text{Var}\left(\delta^*(z_{full}) \mid z_{mean}\right) \right]$$

*Proof.* By the Law of Total Expectation, $\hat{\delta}_{mean}$ is the orthogonal projection of $\delta^*$ onto the coarser $\sigma$-algebra generated by $z_{mean}$. The Pythagorean theorem of statistics (variance decomposition) (Casella & Berger, 2024) gives:

$$\mathbb{E}[(r - \hat{\delta}_{mean})^2] = \mathbb{E}[(r - \delta^*)^2] + \mathbb{E}[(\delta^* - \hat{\delta}_{mean})^2]$$

The first term is $\mathcal{R}(\delta^*)$. The second term is the irreducible approximation error:

$$\mathbb{E}[(\delta^*(z_{full}) - \mathbb{E}[\delta^*(z_{full})|z_{mean}])^2] = \mathbb{E}[\text{Var}(\delta^*|z_{mean})]$$

This term is strictly positive whenever $\delta^*$ depends on $\sigma_\theta^2$ or $q_\theta$ (i.e., when the residual distribution is heteroscedastic or non-Gaussian). $\qquad\square$

**Corollary C.9** (Heterogeneity Penalty). *Under the assumption that $\delta^*$ is L-Lipschitz with respect to the quantile vector $q_\theta$ (i.e., the residual correction is sensitive to distributional shape), deviations in $q_\theta$ that are not explained by the mean $\mu_\theta$ induce proportional squared error in the mean-only baseline.*

1. *Heteroscedasticity: In regions where $\sigma_\theta^2$ varies while $\mu_\theta$ is constant, $\hat{\delta}_{mean}$ is constant while $\delta^*$ varies, forcing $Var(\delta^*|z_{mean}) > 0$.*

2. *Skewness: If the distribution is skewed, the quantiles contain shape information orthogonal to the mean, further increasing the excess risk.*

*Remark* C.10 (Comparison to ResGP (Xing et al., 2021)). This result provides a theoretical explanation for why FIRE outperforms ResGP in heteroscedastic settings (e.g., Table 2 ). ResGP conditions corrections solely on the low-fidelity mean $\mu_\theta(x)$. Consequently, it incurs the excess risk quantified in Theorem A.6 whenever the residual error is input-dependent (Assumption 4.3). FIRE's inclusion of $z_{aug}$ minimizes this approximation gap.

## C.10. Connection to Probabilistic Boosting and Stacking

While standard Bayesian stacking (Yao et al., 2018) combines predictive distributions linearly via a mixture $p(y|x) = \sum_k w_k p_k(y|x)$ to maximize a proper scoring rule (typically Log-Score), FIRE adopts a functional, additive approach akin to probabilistic boosting.

**FIRE as Distributional Convolution.** Unlike standard stacking which averages models, FIRE constructs the final predictive distribution via convolution. Let $p_{LF}(y|x) = q_\theta(y|x, \mathcal{D}_{LF})$ be the low-fidelity predictive density and $p_{res}(r|z_{aug}) = q_\phi(r|z_{aug}, \mathcal{D}_{aug})$ be the residual predictive density conditioned on the augmented statistics of the base model. Under the additive independence structure (Proposition A.2), where we assume the base prediction error and residual error are uncorrelated given $z_{aug}$, the final predictive density is the convolution:

$$p_{\text{FIRE}}(y|x) = (p_{LF}(\cdot|x) * p_{res}(\cdot|z_{aug}))(y) = \int p_{LF}(y - r|x)\, p_{res}(r|z_{aug})\, dr$$

**Theoretical Alignment with Stacking.** Yao et al. (2018) demonstrate that in $\mathcal{M}$-open settings (where the true data-generating process is not in the candidate model set), combining full predictive distributions via proper scoring rules yields better predictive distributions than combining point estimates (stacking of means).

FIRE extends this insight to the autoregressive setting. While GP-based multi-fidelity methods implicitly handle uncertainty via kernels, their autoregressive coupling is typically parameterized by a scaling of the lower-fidelity function value $f_{t-1}$ (Kennedy & O'Hagan, 2000), which functions analogously to "stacking of means." By conditioning the residual model $p_{res}$ on the full set of distributional summaries $z_{aug} = \{\mu, \sigma^2, q^{(\tau)}\}$, FIRE allows the residual model to optimize the Log-Score of the convolved distribution $p_{\text{FIRE}}(y|x)$. Thus, FIRE can be viewed as what we term a distribution-conditioned boosting scheme. While Yao et al. (2018) prove asymptotic KL-optimality for linear mixtures of predictive distributions, FIRE's convolutional combination represents a different functional form. However, both approaches share the principle of propagating full distributional information rather than collapsing to point estimates, which Yao et al. (2018) show is suboptimal in $\mathcal{M}$-open settings.

## C.11. Empirical Validations of Distributional Conditioning on Classic Heteroscedastic Problems

To isolate the impact of FIRE's distributional conditioning, we benchmark against three classic heteroscedastic regression problems:

1. Goldberg et al. (1997): Input data $x \in [0, 1]$; Base function $\mu(x) = 2\sin(2\pi x_i)$; Noise $\varepsilon(x) = \mathcal{N}(0, \sigma(x))$ (Gaussian) with $\sigma(x) = 0.5 + x$ that increases linearly.

2. Yuan & Wahba (2004): Input data $x \in [0, 1]$; Base function $\mu(x) = 2\left[\exp\left(-30(x - 0.25)^2\right) + \sin(\pi x^2)\right] - 2$; Noise $\varepsilon(x) = \mathcal{N}(0, \sigma(x))$ (Gaussian) with $\sigma(x) = e^{\sin(2\pi x_i)}$.

3. Williams (1996): Input data $x \in [0, 1]$; Base function $\mu(x) = \sin(2.5x) \cdot \sin(1.5x)$; Noise $\varepsilon(x) = \mathcal{N}(0, \sigma(x))$ (Gaussian) with $\sigma(x) = 0.01 + 0.25(1 - \sin(2.5x))^2$.

We adapted these single-fidelity benchmarks into a multi-fidelity (shown in Figure 25) setting by generating LF proxies with the base function, while HF observations retained the complex, input-dependent noise structures defined in the original literature (Kersting et al., 2007). We perform a controlled ablation of the stacking strategy. We fix the model backbone, using either all GPs (FIRE-GP) or all TFMs (FIRE), and vary only the input to the residual learner. We compare standard mean-only stacking (without distribution info $\sigma^2, \{q^{(\tau)}\}$), where the residual model conditions only on the LF mean, against FIRE's distributional stacking, where the residual model conditions on the LF mean, variance, and quantiles.

Table 1 and 2 report the accuracy (NRMSE) and negative log-likelihood (NLL). Methods without distribution info $\sigma^2, \{q^{(\tau)}\}$ (mean-conditioned) capture the global trend and achieve comparable NRMSE. However, they perform poorly on NLL. For example, on the Williams function, the mean-only TFM yields an NLL of 2.34, while the distribution-conditioned TFM achieves 1.50. The mean-only approach remains overconfident in high-noise regions. Conditioning on variance and quantiles allows the residual model to detect heteroscedasticity and widen confidence intervals locally.

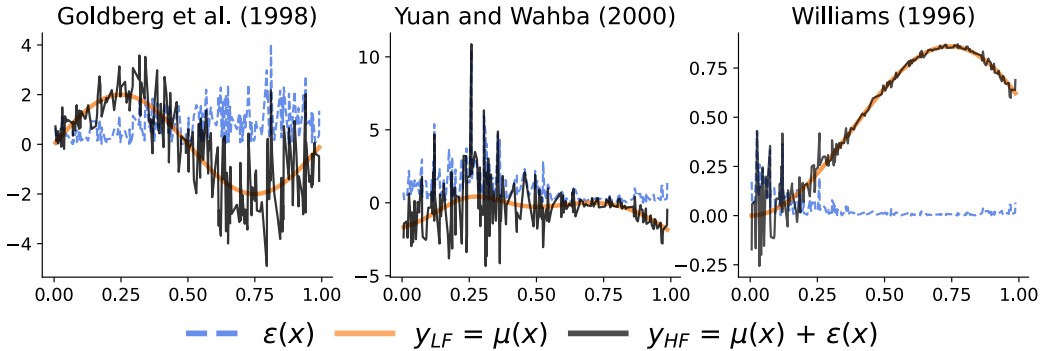

*Figure 25.* Visualizations of classic heteroscedastic regression benchmarks.

*Table 1.* Predictive Accuracy (NRMSE) on Heteroscedastic Benchmarks (lower the better, **bold** is the best).

|  | Goldberg et al. (1997) | Yuan & Wahba (2004) | Williams (1996) |
| --- | --- | --- | --- |
| FIRE-GP **without** distribution info $\sigma^2, \{q^{(\tau)}\}$ | 0.1933±0.0289 | 0.1844±0.0328 | 0.1110±0.0328 |
| FIRE-GP | 0.1867±0.0341 | 0.1834±0.0328 | 0.1082±0.0323 |
| FIRE **without** distribution info $\sigma^2, \{q^{(\tau)}\}$ | 0.1710±0.0317 | 0.1707±0.0117 | 0.1076±0.0320 |
| FIRE | **0.1699±0.0322** | **0.1696±0.0105** | **0.1073±0.0321** |

*Table 2.* Uncertainty Calibration (NLL) on Heteroscedastic Benchmarks (lower the better, **bold** is the best).

|  | Goldberg et al. (1997) | Yuan & Wahba (2004) | Williams (1996) |
| --- | --- | --- | --- |
| FIRE-GP **without** distribution info $\sigma^2, \{q^{(\tau)}\}$ | 5.4922±2.9140 | 9.2186±5.435 | 21.797±18.610 |
| FIRE-GP | 3.2145±0.7244 | 6.9235±3.456 | 17.366±12.476 |
| FIRE **without** distribution info $\sigma^2, \{q^{(\tau)}\}$ | 1.5759±0.1027 | 1.6433±0.1705 | 2.3486±3.3993 |
| FIRE | **1.5631±0.1031** | **1.6359±0.1884** | **1.5039±2.3797** |

## D. Benchmark Dataset Details

This section describes the benchmark datasets used in our experiments, organized by problem category. For each dataset, we detail the problem dimensionality, fidelity structure, and data generation procedure. A quantitative summary of input dimension, number of fidelities, and sample counts is provided in Table 4.

### D.1. Synthetic Datasets:

The implementations for the total synthetic problems are taken from

1. **MF2** library (van Rijn & Schmitt, 2020) (11 problems): A Python library that contains a variety of synthetic problems for benchmarking multi-fidelity kriging and regression algorithms. We select the most commonly used benchmark problems for GP-based MFR: `bohachevsky`, `booth`, `borehole`, `branin`, `currin`, `forrester`, `hartmann6`, `himmelblau`, `park91a`, `park91b`, `six_hump_camelback`. (link: `https://mf2.readthedocs.io/en/latest/`, license: GPL-3.0 license, last accessed: January 22nd, 2026)

2. **Emukit** library (Cutajar et al., 2019) (2 problems): A Python toolkit with code for multi-fidelity emulation, Bayesian optimization, and uncertainty-based decision making. We take the problem formulation of our `branin3f` and `hartmann3f` from the test functions module in this library. (link: `https://github.com/EmuKit/emukit/tree/main/emukit/test_functions`, license: Apache-2.0 license, last accessed: January 22nd, 2026)

3. Equation 3 shows the two-fidelity high-dimensional problems we selected from the existing MFR literature for benchmarking deep-learning-model-based MFR algorithms (Yi et al., 2024; Meng & Karniadakis, 2020; Shan & Wang, 2010). In this research, we tested on 5 variants: $d \in \{10, 20, 30, 40, 50\}$.

$$f^{HF}(\mathbf{x}) = \sum_{i=2}^{d}(2x_i^2 - x_{i-1})^2 + (x_1 - 1)^2$$

$$f^{LF}(\mathbf{x}) = 0.8f^{HF}(\mathbf{x}) + \sum_{i=2}^{d}0.4x_{i-1}x_i - 50$$

(3)

### D.2. Hyperparameter Optimization (HPO) Datasets:

We select 6 HPO MF datasets (`adult`, `Fashion-MNIST`, `higgs`, `jasmine`, `vehicle`, `volkert`) from LCBench (Zimmer et al., 2021) (link: `https://github.com/automl/LCBench`, license: Apache-2.0 license, last accessed: January 22nd, 2026), which is commonly used by MF Bayesian optimization research (Rakotoarison et al., 2024; Lee et al., 2025). The raw datasets have $d = 7$ and contain 2000 learning curves with 51 epochs (fidelity level, $T = 51$). The detail of each feature in $x$ is in Table 3.

*Table 3.* The hyperparameters for LCBench datasets.

| Name | Type | Vaules |
|------|------|--------|
| batch_size | integer | $[2^4, 2^9]$ (log) |
| learning_rate | continuous | $[10^{-4}, 10^{-1}]$ (log) |
| max_dropout | continuous | $[0, 1]$ |
| max_units | integer | $[2^6, 2^{10}]$ (log) |
| momentum | continuous | $[0.1, 0.99]$ |
| max_layers | integer | $[1, 5]$ |
| weight_decay | continuous | $[10^{-5}, 10^{-1}]$ |

We apply two processing steps to construct the multi-fidelity benchmark tasks:

1. Fidelity selection: We define the fidelity levels using the top 5 epochs (46, 47, 48, 49, 50). Using the full 51 epochs is computationally infeasible for autoregressive GP baselines like NARGP, which train separate surrogates for each fidelity level. We reached the compute time limit when attempting to train on all 51 levels. We restrict the problem to 5 fidelities to ensure all algorithms converge within the benchmark time budget.

2. Data imbalance: We enforce a hierarchical sampling structure ($N_T \ll \cdots \ll N_1$) to mimic extreme data scarcity. We fix the low-fidelity sample sizes at $N_1 = 1000$, $N_2 = 750$, $N_3 = 500$, and $N_4 = 250$. We vary the high-fidelity target set $N_5$ in $\{1000, 750, 500, 250\}$ to evaluate robustness.

### D.3. Physics-based Simulation Problems and Datasets:

#### D.3.1. EXISTING ANALYTICAL PARAMETRIC DATASETS:

We selected the two two-fidelity analytical problems (wing and beam) and the three-fidelity, mixed-integer material design problem (HOIP) from the code repository of GP+ (Yousefpour et al., 2024), a Python library for generalized GP modeling (link: `https://github.com/Bostanabad-Research-Group/GP-Plus`, license: MIT license, last accessed: January 22nd, 2026).

#### D.3.2. EXISTING SIMULATED PARAMETRIC DATASETS: BLENDEDNET MULTI-FIDELITY EXTENSION

We use the BlendedNet Multi-Fidelity Extension dataset (Sung et al., 2026), which pairs the high-fidelity FUN3D SA-RANS lift/drag coefficients from BlendedNet (Sung et al., 2025) with matched low-fidelity OpenVSP/VSPAERO vortex-lattice predictions evaluated on the same BWB geometries and flight conditions (link: `https://dataverse.harvard.edu/dataset.xhtml?persistentId=doi:10.7910/DVN/VJT9EP`, license: CC0 1.0, last accessed: January 22nd, 2026).

#### D.3.3. CONCRETE DATASET

The **Concrete** problem ($d$=8) is built from the experimental concrete compressive strength dataset of Yeh (1998) (link:`https://archive.ics.uci.edu/dataset/165/concrete+compressive+strength`, license: CC BY 4.0, last accessed: January 22nd, 2026), which reports 1030 mix designs with eight quantitative inputs (cement, slag, fly ash, water, superplasticizer, coarse aggregate, fine aggregate, and curing age) and measured compressive strength in MPa. We treat the experimental results as our high-fidelity data, and modeling an LF dataset with a physics-inspired multiplicative power-law model in the spirit of Abrams' law on log-transformed inputs (water–cement ratio, cement, slag, fly ash, superplasticizer, age) (Abrams, 1919; Yeh, 2006), similar to modern extensions of Abrams-type relationships for fly-ash concretes and reformed water–cement ratio laws (Mondal & Bhanja, 2022).

**High-Fidelity Data:**     We use the Concrete Compressive Strength dataset of Yeh (1998) from the UC Irvine Machine Learning Repository (link: `https://archive.ics.uci.edu/dataset/165/concrete+compressive+strength`, license: CC BY 4.0, last accessed: January 22nd, 2026). The dataset reports $N_{HF}$=1030 distinct concrete mix designs. Each row contains eight quantitative inputs: cement, blast furnace slag, fly ash, water, superplasticizer, coarse aggregate, fine aggregate, and curing age (in days), and a measured compressive strength target in MPa. This dataset is obtained experimentally and known to exhibit strong nonlinear interactions between mixture proportions and time.

**Low-Fidelity Modeling:**     We derive a physics-inspired empirical LF formulation by fitting a multiplicative power-law model in the spirit of Abrams' law. Classical Abrams formulations relate compressive strength $f_c$ to the water–cement ratio ($w/c$) via a power law (Abrams, 1919). We generalize this idea and fit a log–log regression, consistent with modern extensions of Abrams-type relationships that incorporate age and supplementary cementitious materials (Yeh, 2006; Mondal & Bhanja, 2022), of the form

$$\log f_c^{LF}(\mathbf{x}) = \beta_0 + \beta_1 \log(w/c) + \beta_2 \log(\text{cement}) + \beta_3 \log(\text{slag}) + \beta_4 \log(\text{fly ash}) + \beta_5 \log(\text{superplasticizer}) + \beta_6 \log(t),$$

where $t$ is the curing age (days). This corresponds to the multiplicative model

$$f_c^{LF}(\mathbf{x}) = \exp(\beta_0)\,(w/c)^{\beta_1}(\text{cement})^{\beta_2}(\text{slag})^{\beta_3}(\text{fly ash})^{\beta_4}(\text{superplasticizer})^{\beta_5}t^{\beta_6},$$

with coefficients $\{\beta_i\}$ estimated by ordinary least squares on the log-strength. For the Concrete surrogate, the fitted values (rounded to three significant figures) are

$$
\begin{aligned}
\beta_0 &= 2.56, \\
\beta_1 &= -0.815 \quad (\log(w/c)), \\
\beta_2 &= -0.0380 \quad (\log \text{cement}), \\
\beta_3 &= 0.0161 \quad (\log \text{slag}), \\
\beta_4 &= 0.00231 \quad (\log \text{fly ash}), \\
\beta_5 &= 0.0148 \quad (\log \text{superplasticizer}), \\
\beta_6 &= 0.292 \quad (\log \text{age}).
\end{aligned}
$$

We then define $y^{LF}(\mathbf{x}) = f_c^{LF}(\mathbf{x})$ and $y^{HF}(\mathbf{x})$ as the measured compressive strength.

### D.3.4. CAR DATASET

We present the **Car** dataset ($d$=23), a multi-fidelity automotive aerodynamics dataset that we construct from the DrivAerNet dataset (Elrefaie et al., 2025). (link: `https://github.com/Mohamedelrefaie/DrivAerNet`, license: Attribution-NonCommercial 4.0 International, last accessed: January 22, 2026)

**High-Fidelity Data:** The high-fidelity labels $y^{HF}$ are the DrivAerNet steady 3D RANS drag coefficients. Each input $x$ consists solely of parametric shape controls such as ramp and diffusor angles, trunklid angle, side-mirror rotation and placement, rear-window and windscreen inclination, overall car length and width, roof height, and green-house angle. The external flow condition is held fixed across all cases (constant freestream speed, Reynolds number, and boundary setup).

**Low-Fidelity Simulation:** Using the same design parameter as the high-fidelity data, our low-fidelity labels $y^{LF}$ are obtained from a quasi-2D OpenFOAM pipeline: we remove wheel geometry from the 3D mesh, form a 2D silhouette projection, generate a refined 2D mesh with Gmsh, extrude it into a thin 3D slab, and run incompressible RANS (`simpleFoam`) to estimate drag (Figure 26). This new paired HF–LF CAR dataset is one of the main contributions of our benchmark suite and is designed to mimic a realistic but geometry-only car drag prediction problem.

We present the details of the quasi-2D LF CFD pipeline used to generate $C_D^{LF}$ for the CAR dataset described in the main text. For each parametric geometry, we start from the DrivAerNet STL, construct a 2D silhouette, generate a quasi-2D mesh, and run a steady incompressible RANS simulation in OpenFOAM. The overall workflow is illustrated in Fig. 26. Note that the mesh panel shows a zoomed-in region near the car, while the actual far-field domain extends several vehicle lengths upstream and downstream.

1. **Flow conditions and fluid properties.** All LF simulations are run at the same freestream speed and fluid properties as the HF DrivAerNet cases. We prescribe a uniform inlet velocity $U_\infty = 30 \text{ m/s}$ for air with constant density and kinematic viscosity $\rho_\infty = 1.184 \text{ kg/m}^3$, $\nu = 1.56 \times 10^{-5} \text{ m}^2/\text{s}$.

   This yields Reynolds numbers comparable to the HF setup for a typical passenger-car length scale. The reference area used in the LF drag normalization is defined from the projected frontal height of the car, $h_{\text{front}}$, multiplied by the extrusion thickness, $A_{\text{ref}}^{LF} = h_{\text{front}} H_{\text{extrude}}$, $H_{\text{extrude}} = 10^{-3}$ m. We keep HF and LF drag coefficients in their respective native normalizations and let the surrogate models learn any scale differences between fidelities.

2. **Geometry processing and 2D silhouette.** Starting from the 3D STL surface of the car body, we first remove all wheel geometry before further processing. This avoids artificial lobes and unphysical wheel wakes in the 2D approximation, and focuses the LF model on the main body shape. The remaining surface is sampled densely: we collect all mesh vertices and an additional $3 \times 10^5$ random surface points, projected onto the $x$–$z$ plane. From the resulting point cloud we construct a watertight 2D silhouette by computing an $\alpha$-shape with $\alpha = 0.02 \times \big(\text{diagonal of the 3D bounding box}\big)$, which scales automatically with the geometry size. The polygon boundary is then cleaned by removing near-duplicate points and very short edges, yielding a smooth, single closed curve that captures the outer hull of the vehicle. This step corresponds to the top-right panel of Fig. 26.

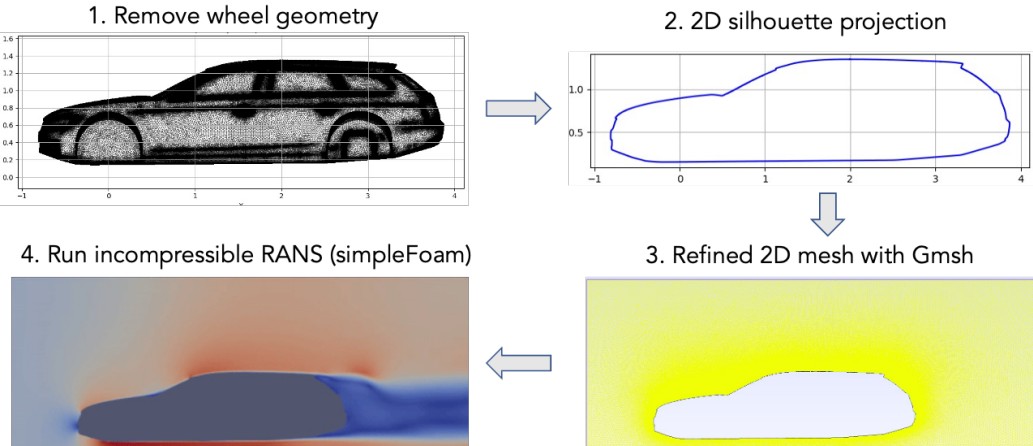

*Figure 26.* Quasi-2D low-fidelity pipeline for the CAR dataset. Top-left: projected mid-plane view of the 3D DrivAer-style geometry after removing wheels. Top-right: cleaned 2D silhouette obtained from the projected points. Bottom-right: Gmsh-generated quasi-2D mesh (shown as a zoom-in around the car; the full domain extends several car lengths upstream/downstream and vertically) with refinement concentrated near the body. Bottom-left: sample incompressible RANS solution from the `simpleFoam` run used to compute the LF drag coefficient.

3. **Computational domain in 2D.** Let $L$ denote the car length and $x_{\text{car,min}}$, $x_{\text{car,max}}$ the minimum and maximum $x$-coordinates of the silhouette. We embed the car in a rectangular 2D far-field domain defined by

$$x_{\min} = x_{\text{car,min}} - 3L,$$
$$x_{\max} = x_{\text{car,max}} + 10L,$$
$$z_{\min} = 0, \qquad z_{\max} = z_{\text{car,max}} + 5 \text{ m}.$$

For a nominal passenger car with $L \approx 4.5$–$4.6$ m this corresponds to roughly 13–14 m of upstream length, 45 m downstream, and a top boundary several meters above the roof. The mesh close-up in Fig. 26 shows only a zoomed-in subregion near the body; the full computational box is significantly larger in all directions.

4. **Gmsh meshing and 2.5D extrusion.** The 2D silhouette and far-field box are meshed using a fully automated Gmsh pipeline. We employ a distance-based size field with two nominal element sizes: $h_{\text{fine}} = 0.01$ m, $h_{\text{coarse}} = 0.1$ m. Cells near the car boundary are refined towards $h_{\text{fine}}$, while elements in the far field gradually transition towards $h_{\text{coarse}}$ with additional upstream/downstream growth limits. The resulting 2D meshes contain on the order of $\mathcal{O}(10^5)$ triangular/quadrilateral cells per case, providing a reasonably resolved near-body region for trend-accurate drag without explicit $y^+$ targeting.

To run OpenFOAM, the 2D region is extruded into 3D by a small thickness, $H_{\text{extrude}} = 10^{-3}$ m, with a single prism layer, yielding a quasi-2D ("2.5D") volume mesh. The extrusion exists solely to satisfy OpenFOAM's 3D requirements; we explicitly enforce flow invariance across the thin spanwise direction by using empty boundary conditions on the extruded faces. During meshing, we automatically label the boundary patches as `carWall`, `inlet`, `outlet`, `ground`, `freestream` (top), and `front`/`back` (extruded faces).

5. **Mesh conversion and boundary conditions.** The Gmsh mesh is converted to OpenFOAM format using `gmshToFoam`, and quality is checked with `checkMesh` (non-orthogonality, skewness, and boundary consistency). Custom scripts adjust the patch names to match the simulation setup. Boundary conditions are as follows:

- **Inlet (`inlet`):** fixed velocity $U = (30, 0, 0)$ m/s, prescribed turbulence fields $k$ and $\omega$ consistent with the inlet turbulence intensity.
- **Outlet (`outlet`):** zeroGradient for $U$ and turbulence fields, fixedValue $p = 0$ for pressure.
- **Ground (`ground`):** no-slip condition for $U$, standard wall treatment for $k$ and $\omega$.
- **Car body (`carWall`):** no-slip condition on the silhouette, with wall functions for turbulence as in a standard $k$–$\omega$ SST setup.

- **Top boundary (`freestream`):** slip condition for velocity (zero normal velocity, zero shear), zeroGradient for pressure.
- **Front/back (`front, back`):** `empty` boundaries, enforcing 2D behavior across the thin extrusion direction.

This combination encodes the modeling assumption that the flow is invariant in the spanwise direction and can be approximated as 2D, while still using a 3D finite-volume solver.

6. **Numerical schemes and linear solvers.** All simulations use the steady-state `simpleFoam` solver with an incompressible RANS formulation and a $k$–$\omega$ SST turbulence model. The discretization schemes in `fvSchemes` are:

- **Time derivative:** `ddtSchemes{ default steadyState;}`.
- **Gradients:** Gauss linear with cell-limited gradients for $\nabla U$, $\nabla k$, and $\nabla \omega$.
- **Convective terms:** first-order Gauss upwind for `div(phi,U)`, `div(phi,k)`, and `div(phi,omega)` for robustness across many geometries.
- **Diffusive/viscous terms:** Gauss linear with corrected surface-normal gradients (e.g., for $\nabla \cdot (\nu_{\text{eff}} \nabla U)$ and Laplacians).

The linear solvers and SIMPLE controls in `fvSolution` are:

- Pressure $p$: `GAMG` with Gauss–Seidel smoother, absolute tolerance $10^{-6}$ and relative tolerance 0.1 for intermediate iterations; a final solve with `relTol = 0` ensures full convergence.
- Velocity and turbulence fields $(U, k, \omega)$: `smoothSolver` with symmetric Gauss–Seidel smoothing, absolute tolerance $10^{-6}$ and relative tolerance 0.1 for intermediate steps, followed by final solves with zero relative tolerance.
- SIMPLE algorithm: `nNonOrthogonalCorrectors = 2` non-orthogonal corrections per iteration, with under-relaxation factors of 0.3 for $p$ and 0.5 for $U$, $k$, and $\omega$.

In practice, we iterate until residuals have dropped by several orders of magnitude and the forceCoeffs-reported drag has reached a stable plateau.

7. **Drag extraction and computational cost.** Drag and lift coefficients are obtained using OpenFOAM's `forceCoeffs` function object applied to the `carWall` patch:

$$C_D^{LF} = \frac{1}{\frac{1}{2}\rho_\infty U_\infty^2 A_{\text{ref}}^{LF}} \left(\text{streamwise force on } \texttt{carWall}\right),$$

The function object is evaluated every time step, and the final drag coefficient is taken from the plateau region of the force history.

Each LF simulation typically requires on the order of 10 minutes of wall-clock time on a node with $\sim 16$ CPU cores, reflecting the modest mesh size and relatively simple numerics. In contrast, the HF DrivAerNet simulations rely on significantly larger 3D meshes and more expensive numerics, requiring on the order of hundreds of CPU-hours per case. The resulting CAR dataset therefore provides a realistic multi-fidelity pairing: a costly, high-accuracy 3D RANS drag coefficient $C_D^{HF}$ and a much cheaper, trend-accurate quasi-2D estimate $C_D^{LF}$ for the same parametric car geometries.

## D.4. Data Sampling Imbalance Protocol.

As mentioned in 4.2, we systematically evaluate performance under data imbalance regimes by fixing the LF sample size and $N_{\text{LF}}$ and varying the HF budget $N_{\text{HF}}$ across six imbalance ratios: $\{2\%, 4\%, 5\%, 10\%, 20\%, 25\%\}$. For synthetic functions, we sampled the data using Latin hypercube sampling and set $N_{\text{LF}} = 200$ for problems with dimension $d < 10$, and $N_{\text{LF}} = 2000$ for $d \geq 10$ based on the curse of dimensionality exponential scaling laws. For LCBench HPO problems, we fixed the $\{N_1, N_2, N_3, N_4\} = \{1000, 750, 500, 250\}$ following how the exising literature partitioned their data imbalance for problems with more than 2 fidelity (Yousefpour et al., 2024). For engineering problems, we set the $N_{\text{LF}}$ based on the existing dataset size. The $N_{\text{test}}$ for all problems is set to be $\frac{N_{\text{LF}}}{2}$.

*Table 4.* List of datasets

| Dataset | Data Source | Dimension (Features, $d$) | Fidelity Level ($T$) | $\{N_{t=1}, N_{t=2}, \cdots, N_{t=T-1}\}$ | $N_T$ |
|---|---|---|---|---|---|
| Bohachevsky | Synthetic (MF2) | 2 | 2 | {200} | {4, 8, 10, 20, 40, 50} |
| Booth | Synthetic (MF2) | 2 | 2 | {200} | {4, 8, 10, 20, 40, 50} |
| Borehole | Synthetic (MF2) | 8 | 2 | {200} | {4, 8, 10, 20, 40, 50} |
| Branin | Synthetic (MF2) | 2 | 2 | {200} | {4, 8, 10, 20, 40, 50} |
| Currin | Synthetic (MF2) | 2 | 2 | {200} | {4, 8, 10, 20, 40, 50} |
| Forrester | Synthetic (MF2) | 1 | 2 | {200} | {4, 8, 10, 20, 40, 50} |
| Hartmann | Synthetic (MF2) | 6 | 2 | {200} | {4, 8, 10, 20, 40, 50} |
| Himmelblau | Synthetic (MF2) | 2 | 2 | {200} | {4, 8, 10, 20, 40, 50} |
| Park91a | Synthetic (MF2) | 4 | 2 | {200} | {4, 8, 10, 20, 40, 50} |
| Park91b | Synthetic (MF2) | 4 | 2 | {200} | {4, 8, 10, 20, 40, 50} |
| Six-Hump Camelback | Synthetic (MF2) | 2 | 2 | {200} | {4, 8, 10, 20, 40, 50} |
| HD 10 | Synthetic (BNN) | 10 | 2 | {2000} | {40, 80, 100, 200, 400, 500} |
| HD 20 | Synthetic (BNN) | 20 | 2 | {2000} | {40, 80, 100, 200, 400, 500} |
| HD 30 | Synthetic (BNN) | 30 | 2 | {2000} | {40, 80, 100, 200, 400, 500} |
| HD 40 | Synthetic (BNN) | 40 | 2 | {2000} | {40, 80, 100, 200, 400, 500} |
| HD 50 | Synthetic (BNN) | 50 | 2 | {2000} | {40, 80, 100, 200, 400, 500} |
| branin3f | Synthetic (GP+) | 2 | 3 | {200, 50} | {40, 80, 100, 200, 400, 500} |
| hartmann3f | Synthetic (GP+) | 3 | 3 | {200, 50} | {40, 80, 100, 200, 400, 500} |
| adult | LCBench | 7 | 5 | {1000, 750, 500, 250} | {20, 40, 50, 100, 200, 250} |
| Fashion-MNIST | LCBench | 7 | 5 | {1000, 750, 500, 250} | {20, 40, 50, 100, 200, 250} |
| Higgs | LCBench | 7 | 5 | {1000, 750, 500, 250} | {20, 40, 50, 100, 200, 250} |
| Jasmine | LCBench | 7 | 5 | {1000, 750, 500, 250} | {20, 40, 50, 100, 200, 250} |
| Vehicle | LCBench | 7 | 5 | {1000, 750, 500, 250} | {20, 40, 50, 100, 200, 250} |
| Volkert | LCBench | 7 | 5 | {1000, 750, 500, 250} | {20, 40, 50, 100, 200, 250} |
| Beam | Engineering | 5 | 2 | {200} | {4, 8, 10, 20, 40, 50} |
| Wing | Engineering | 10 | 2 | {2000} | {40, 80, 100, 200, 400, 500} |
| HOIP | Engineering | 3 | 3 | {100, 25} | {2, 4, 5, 10, 20, 25} |
| Concrete | Engineering (Our) | 8 | 2 | {200} | {4, 8, 10, 20, 40, 50} |
| BWB-CD | Engineering (Our) | 14 | 2 | {1000} | {20, 40, 50, 100, 200, 250} |
| BWB-CL | Engineering (Our) | 14 | 2 | {1000} | {20, 40, 50, 100, 200, 250} |
| Car | Engineering (Our) | 23 | 2 | {500} | {10, 20, 25, 50, 100, 125} |

# E. Algorithms Selection and Implementation Details

## E.1. Baseline Algorithms

We benchmark FIRE against eight state-of-the-art multi-fidelity regression methods, using GPU-accelerated PyTorch implementations whenever available to ensure a fair and consistent comparison with TFMs run on GPU.

1. AR(1) (Kennedy & O'Hagan, 2000): Since the original paper does not come with a public code repository, we follow the latest PyTorch implementation from the ContinuAR paper (Xing et al., 2023)'s supplementary material for our AR(1).

2. NARGP (Perdikaris et al., 2017): We implemented the NARGP since the original code is not in PyTorch (link to original code: `https://github.com/Xiao-dong-Wang/Multifidelity-GP`, license: MIT license, last accessed: January 22nd, 2026). We used the latest implementation of NARGP, following both the original paper (Perdikaris et al., 2017) and Cutajar et al. (2019), which expand NARGP to more than two fidelities by sequentially modeling each fidelity.

3. ResGP (Xing et al., 2021): Since the original paper does not come with a public code repository, we follow the latest PyTorch implementation from the ContinuAR paper (Xing et al., 2023)'s supplementary material for our ResGP.

4. MisoKG (Poloczek et al., 2017): We take the open-source MisoKG code of its surrogate from BoTorch's GitHub repository (link: `https://botorch.org/docs/tutorials/discrete_multi_fidelity_bo/`, license: MIT license, last accessed: January 22nd, 2026).

5. ContinuAR (Xing et al., 2023): We follow the implementation from the original paper's supplementary material as our ContinuAR code.

6. MFBNN (Yi et al., 2024): We take the code from the paper's open-source GitHub repository. (link: `https://github.com/bessagroup/mfbml`, license: BSD 3-Clause License, last accessed: January 22nd, 2026).

7. MFRNP (Niu et al., 2024): We take the code from the paper's open-source GitHub repository. (link: `https://github.com/Rose-STL-Lab/MFRNP`, license: MIT license, last accessed: January 22nd, 2026).

8. FIRE-GP: We implemented FIRE-GP using BoTorch's `SingleTaskGP` since the model support the API for accessing mean, variance, and quantiles through posterior estimation (link: `https://botorch.org/docs/getting_started`, license: MIT license, last accessed: January 22nd, 2026). This API analytically evaluates the inverse cumulative distribution function (ICDF) of the marginal Gaussian posterior at each candidate point for computing quantiles.

### E.2. Tabular Foundation Models

We use all tabular foundation models in a zero-shot setting, without any task-specific fine-tuning or hyperparameter tuning, to isolate their out-of-the-box generalization performance in multi-fidelity regression.

1. TabPFNv2.5 (Grinsztajn et al., 2025): We use the default pretrained checkpoint of TabPFNv2.5, `tabpfn-v2.5-regressor-v2.5_default.ckpt`, released by the authors (Prior Labs) (link: `https://huggingface.co/Prior-Labs/tabpfn_2_5`, license: Prior Labs License, last accessed: January 22nd, 2026).

2. TabPFNv2 (Hollmann et al., 2025): We use the default pretrained checkpoint of TabPFNv2, `tabpfn-v2-regressor.ckpt`, released by the authors (Prior Labs) (link: `https://huggingface.co/Prior-Labs/TabPFN-v2-reg`, license: Prior Labs License, last accessed: January 22nd, 2026).

3. RealTabPFN (Garg et al., 2025): We use the default pretrained checkpoint of RealTabPFN, `tabpfn-v2.5-regressor-v2.5_real.ckpt`, which is trained on a mixture of synthetic and real-world tabular datasets (link: `https://huggingface.co/Prior-Labs/tabpfn_2_5`, license: Prior Labs License, last accessed: January 22nd, 2026).

4. Mitra (Zhang et al., 2025): We use the pretrained `mitra-regressor` model released by AutoGluon without fine-tuning (link: `https://huggingface.co/autogluon/mitra-regressor`, license: Prior Labs License, last accessed: January 22nd, 2026).

**Accessing statistical information from TFMs:** All TabPFN models suport the API for accesing mean, variance, and quantile through their `regressor.py`. TabPFN approximates the posterior predictive distribution $p(y|x, \mathcal{D})$ using a piecewise-constant Riemann distribution defined over fixed intervals (buckets), where the model outputs logits representing the probability mass of each bucket (Müller et al., 2022; Hollmann et al., 2023; 2025). Quantiles are analytically derived by computing the discrete cumulative distribution function (CDF) from these logits and performing linear interpolation between bucket borders to solve for the inverse CDF.

Mitra does not support the API for accessing quantile. To extract uncertainty estimates from the Mitra model, we implement a split conformal prediction wrapper that calibrates the model's outputs using a held-out portion of the training data. Specifically, we calculate the empirical percentiles of the prediction residuals ($y_{true} - \hat{y}$) on this calibration set to determine constant additive offsets for each desired quantile. At inference time, these offsets are added to the model's mean prediction to generate the final quantile estimates.

## F. Performance Results Per Datasets

This section provides the complete breakdown of predictive performance for all 31 datasets included in our benchmark suite. We visualize the impact of data scarcity by plotting Normalized Root Mean Squared Error (NRMSE), Negative Log Likelihood (NLL), and Coefficient of Determination ($R^2$) against varying high-fidelity to low-fidelity sample ratios. The figures below are organized by domain to facilitate detailed comparison: Figures 27-28 cover Physics-based simulations,

Figure 29 details Hyperparameter Optimization (HPO) tasks, and Figures 30-32 display results for Synthetic functions. Across these domains, FIRE demonstrates robust generalization, maintaining stable performance rankings even as the high-fidelity budget is severely constrained.

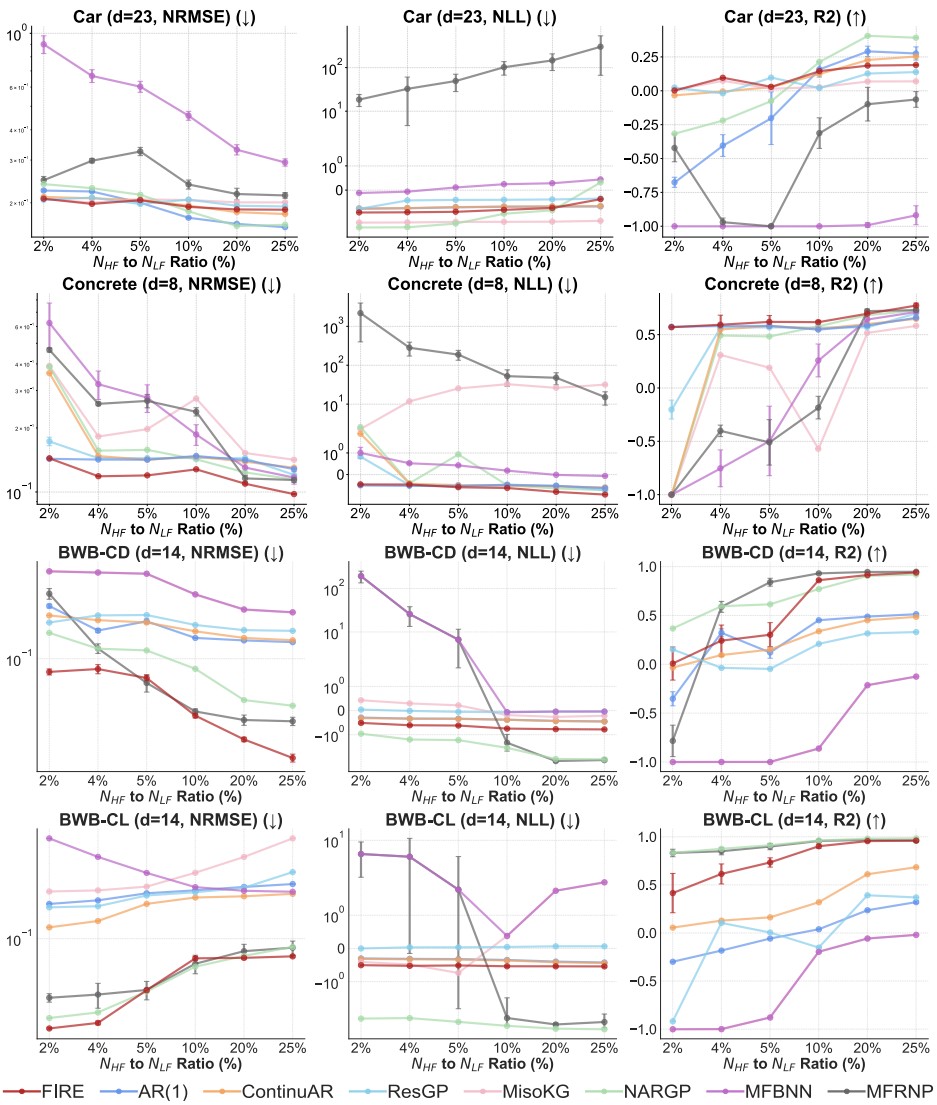

*Figure 27.* Result of Physics-based simulation problems (Car, Concrete, BWB).

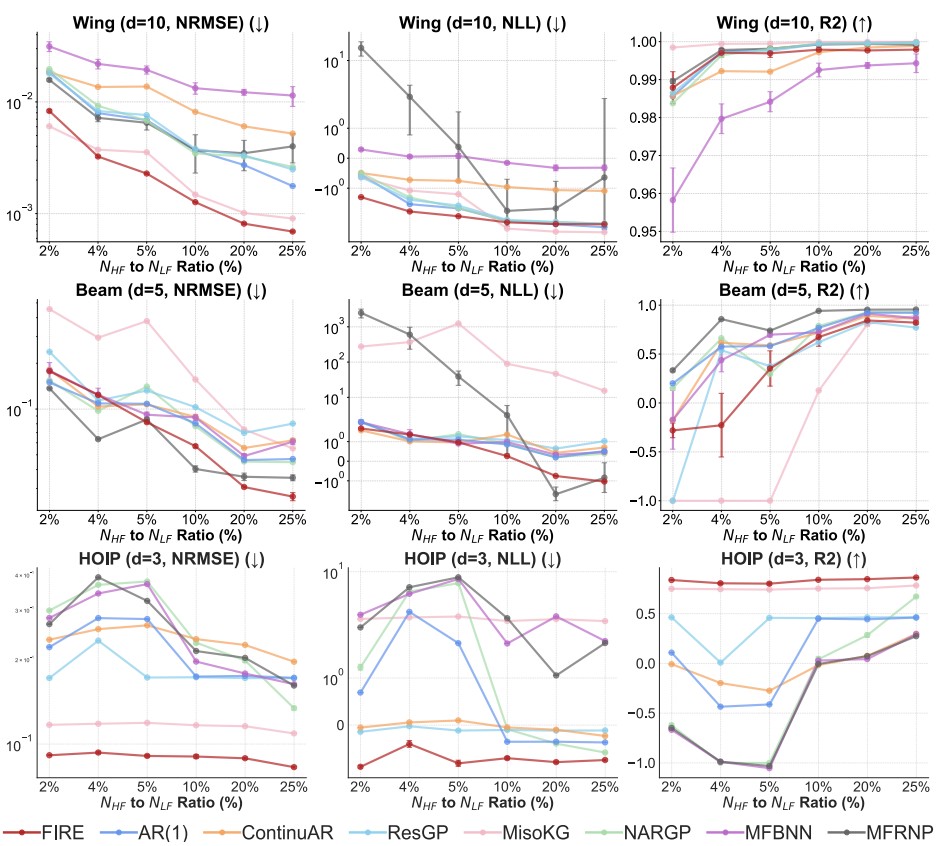

*Figure 28.* Result of Physics-based parametric problems (Wing, Beam, HOIP).

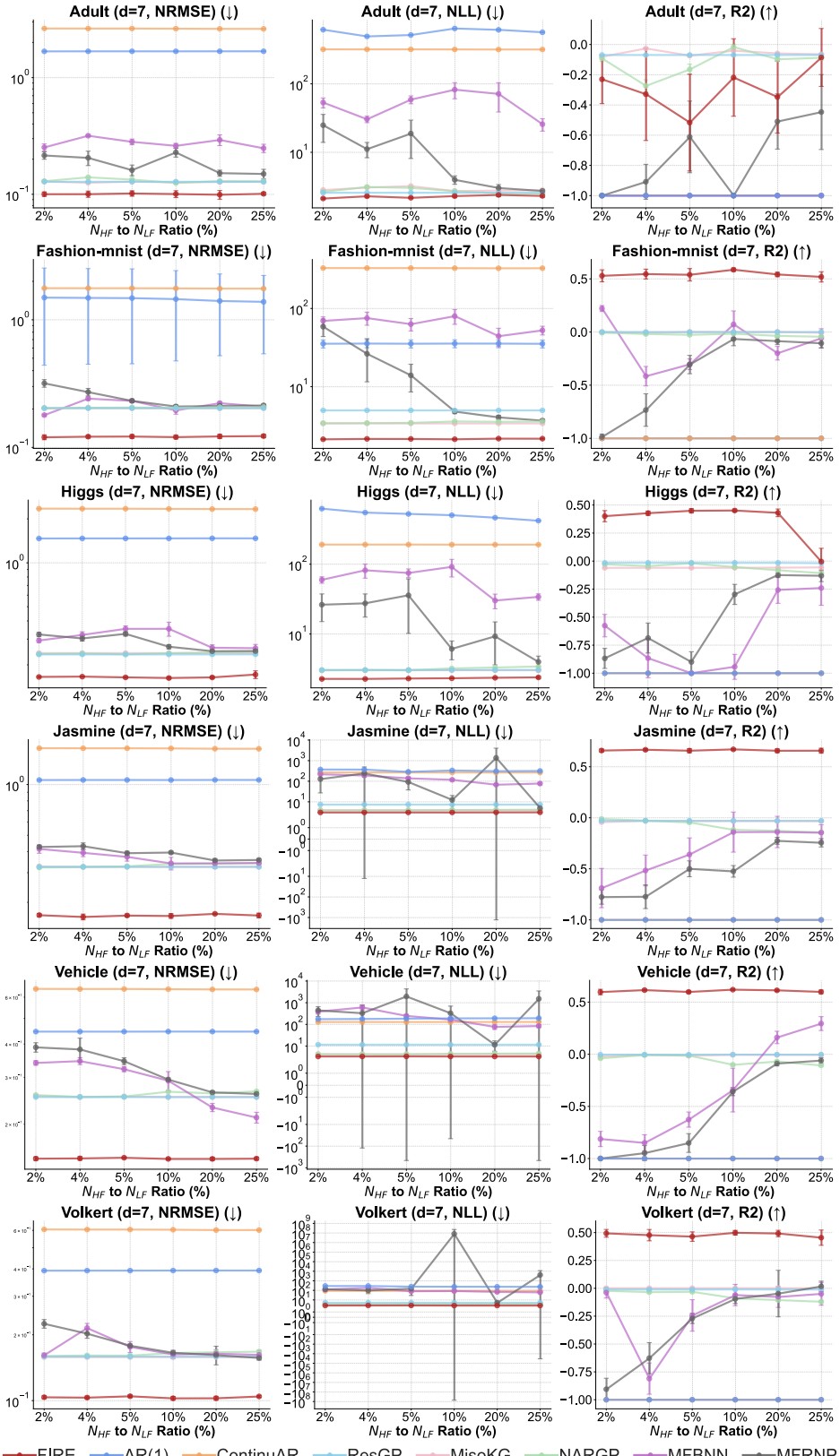

*Figure 29.* Result of Hyperparameter Optimization (HPO) problems from LCBench.

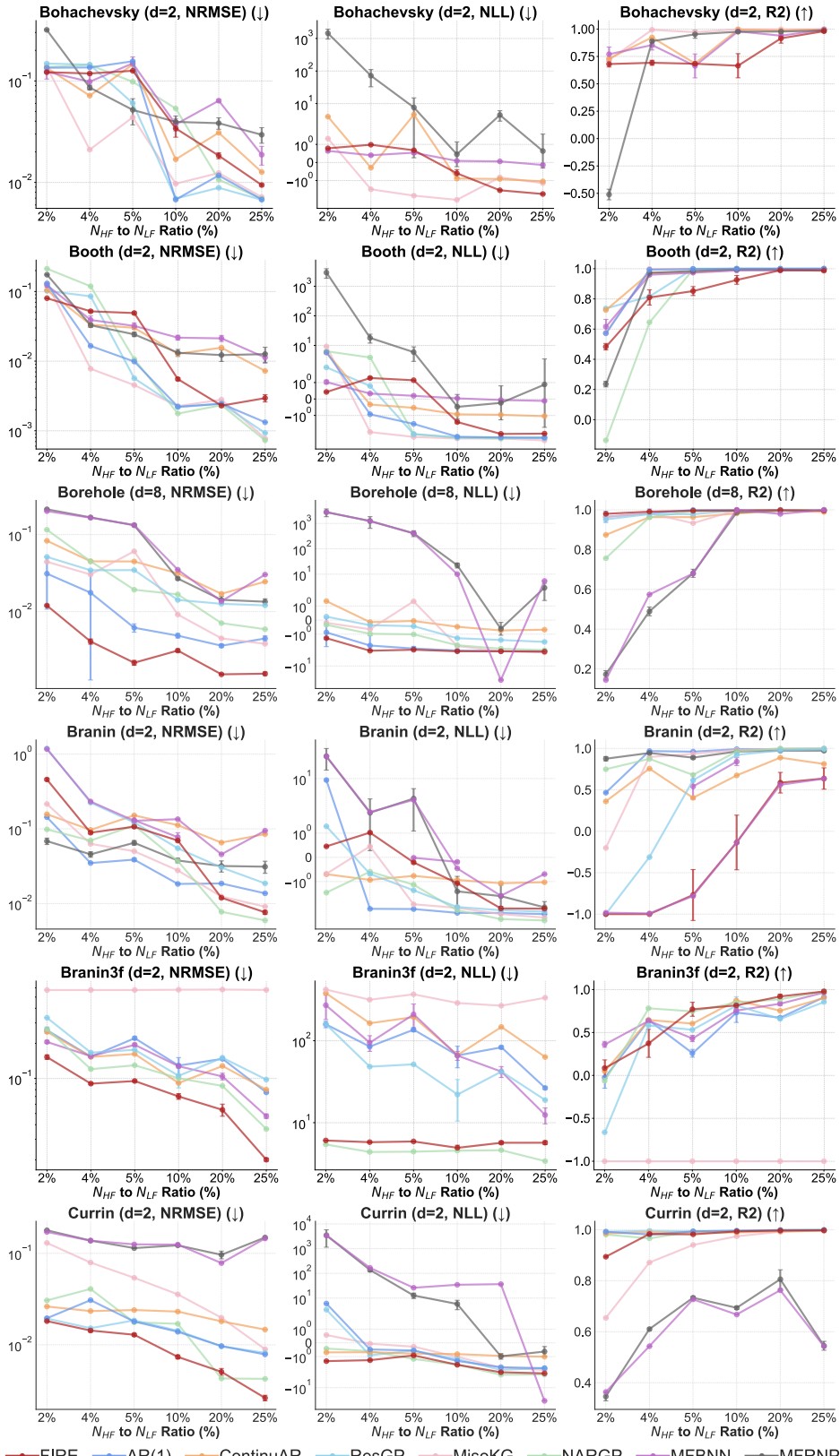

*Figure 30.* Result of synthetic problems (Bohachevsky, Booth, Borehole, Branin, Branin3f, Currrin).

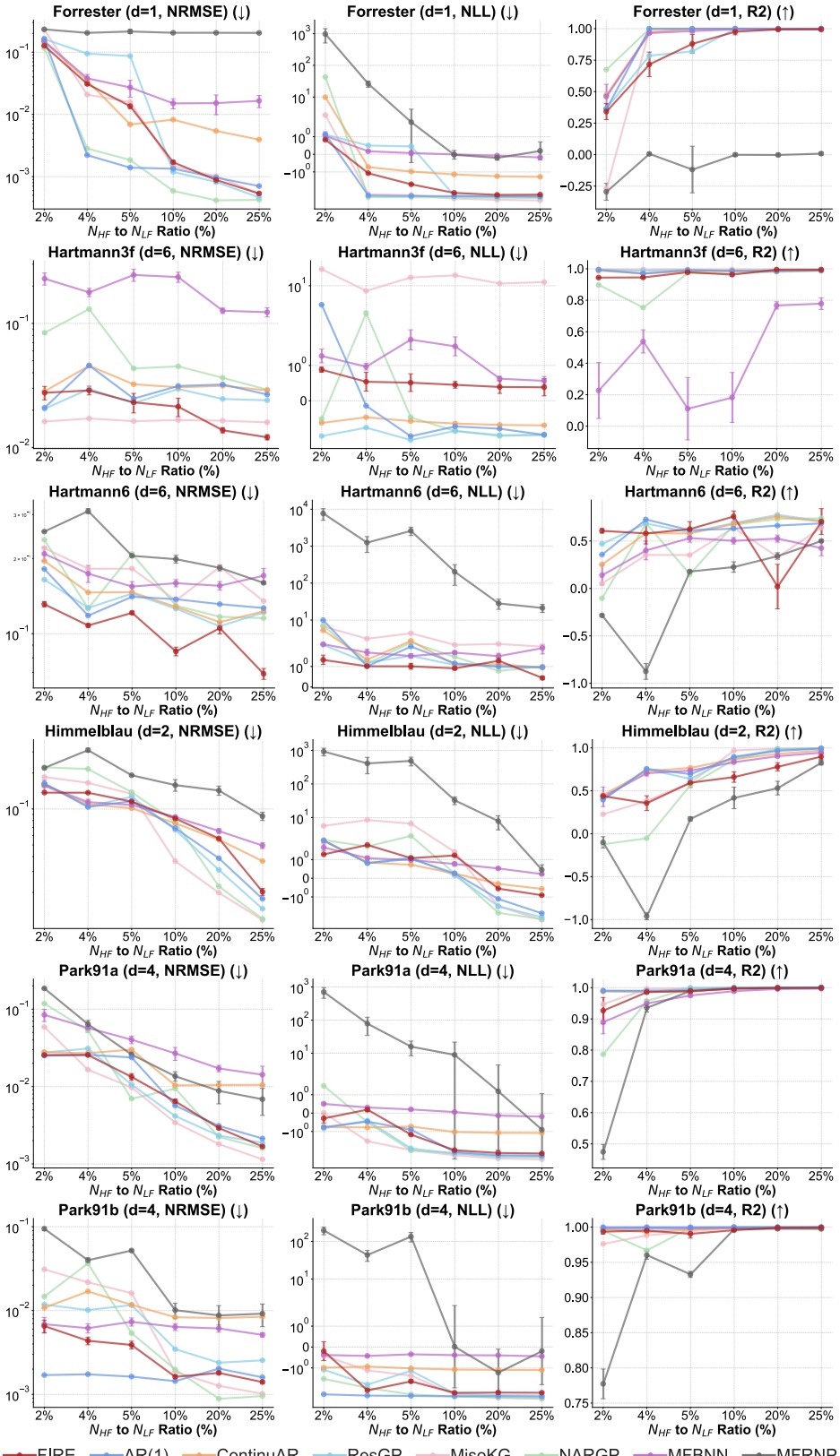

*Figure 31.* Result of synthetic problems (Forrester, Hartmann3f, Hartmann6, Himmelblau, Park91a, Park91b).

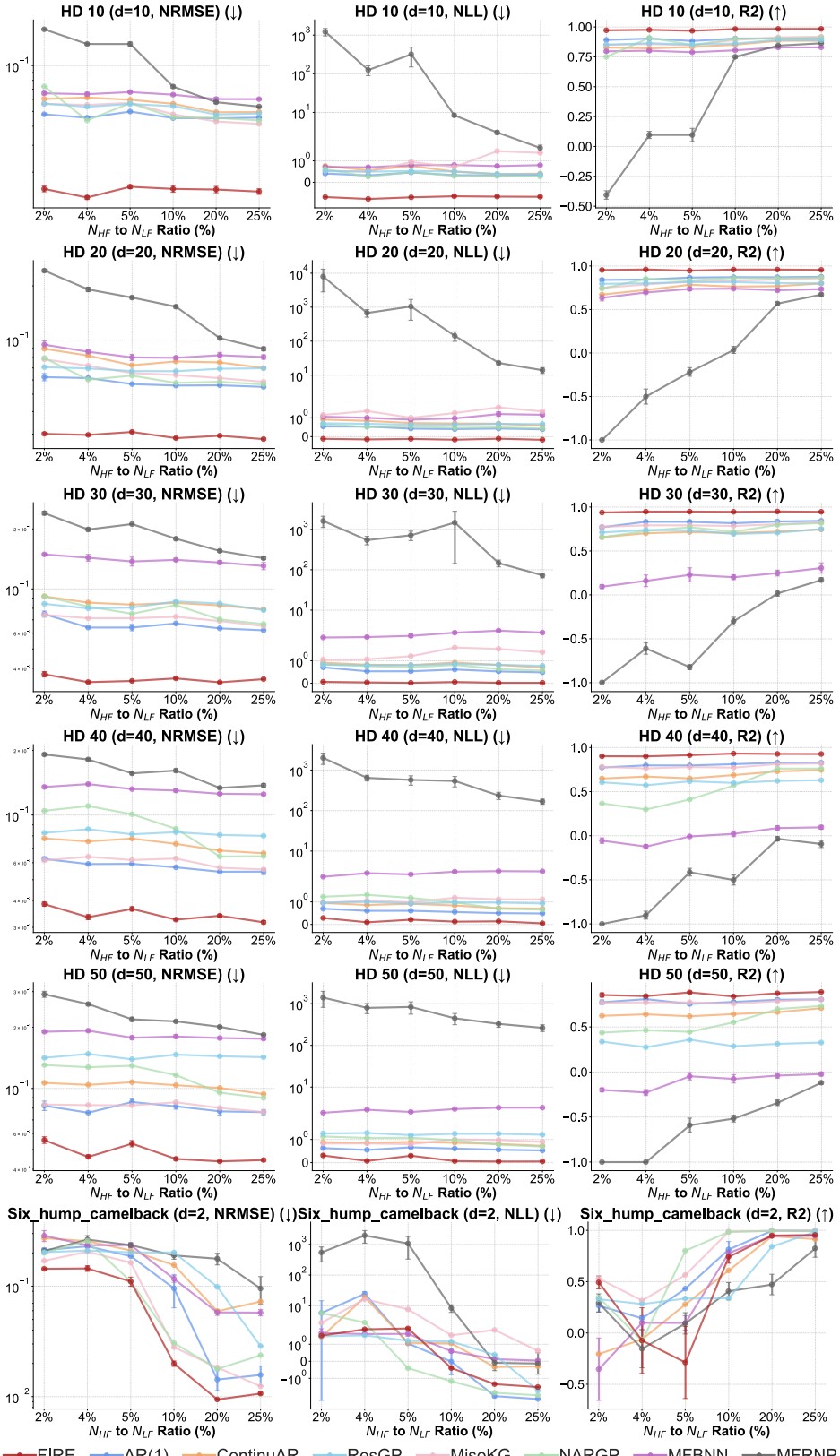

*Figure 32.* Result of synthetic problems (HD 10∼HD 50, sixhump-camelback).

