# OpenReview forum: "FIRE: Multi-Fidelity Regression with Distribution-Conditioned In-Context Learning Using Tabular Foundation Models"
_ICML.cc/2026/Conference — ICML 2026 spotlight_

### Official Review · Reviewer_u38Z · 2026-03-04

**Soundness:** 4
**Presentation:** 4
**Significance:** 4
**Originality:** 3
**Overall Recommendation:** 5
**Confidence:** 3

**Summary:**

This paper proposes FIRE, a training-free multi-fidelity (MF) regression framework that builds on tabular foundation models (TFMs), specifically TabPFN, to perform zero-shot in-context Bayesian inference for high-fidelity (HF) prediction. The key idea is to (i) use a TFM as a low-fidelity (LF) surrogate, (ii) model the HF residuals conditioned on LF predictive distributions (mean, variance, quantiles) to capture heteroscedastic cross-fidelity discrepancies, and (iii) propagate uncertainty additively. The method is evaluated on 31 MF benchmarks (including engineering CFD-style simulations and HPO tasks), where it reportedly outperforms seven GP-based and deep learning MF baselines in terms of predictive accuracy, uncertainty calibration, and runtime, while requiring no task-specific training of the TFM.

**Compliance With Llm Reviewing Policy:**

Affirmed.

**Final Justification:**

The authors have addressed all my concerns during the rebuttal. Therefore, I retain my score and recommend acceptance.

**Key Questions For Authors:**

Please refer to Questions 4 and 5 in weaknesses.

**Limitations:**

yes

**Strengths And Weaknesses:**

## Strengths

- **High-significance problem and practical impact**
  A critical issue assessed by this manuscript is how to leverage cheap but biased LF simulations together with a very small amount of expensive HF data to obtain accurate predictions with uncertainty quantification. MF regression is central in scientific and engineering workflows where HF simulations/experiments are extremely expensive and LF models are cheap but biased. Tackling extreme HF scarcity and non-nested designs is highly relevant for real-world use cases like CFD benchmarks and HPO.

- **Clear problem formulation and well-justified design choices**
  The paper gives a crisp formalization of MF regression and the limitations of GP-based approaches (cubic scaling, overfitting in HF-scarce regimes, nested-design assumptions). The distributional conditioning of the HF correction on LF posterior summaries (mean/variance/quantiles) is particularly well-motivated, with a solid discussion of heteroscedasticity and predictive stacking (e.g., Sections 7.2 and 7.3). I especially appreciated the theoretical and conceptual arguments for why conditioning on the full LF predictive distribution (rather than just the mean) should better capture input-dependent discrepancies. The variance-conditioned selection and discussion of how LF uncertainty guides HF correction are intuitive and well-argued, helping justify the design beyond heuristic choices.

- **Extensive and convincing empirical validation**
  The experiments span 31 benchmarks covering synthetic MF settings, engineering-like CFD simulations (e.g., DrivAerNet), and HPO problems. FIRE consistently shows superior accuracy–uncertainty–runtime trade-offs, with strong performance both in nested and non-nested regimes and in extremely low HF data fractions (2–5%), where classical GP-based methods are known to struggle. The runtime advantage and the ability to operate training-free—using an off-the-shelf TFM—are clear practical benefits.

- **Simplicity and deployability**
  The framework leverages pre-trained TabPFN as a drop-in TFM without any additional task-specific training, which makes it highly attractive for practitioners. The overall pipeline, especially as summarized around Algorithm 1, is conceptually straightforward and easy to implement with a pre-trained TFM API.


## Weaknesses

- **1. Clarity and referencing around Algorithm 1**
  Algorithm 1 is central to understanding how the three-stage procedure (LF inference, residual modeling, uncertainty propagation) operates in practice. In my reading, the text does not clearly call out or refer back to Algorithm 1 at all the natural points where it should (e.g., when describing the full FIRE pipeline). Please make sure Algorithm 1 is referenced explicitly in the exposition and that the mapping between narrative and pseudocode is easy to follow.

- **2. Lack of explicit probabilistic formulation**
  While the narrative is Bayesian and the method is described in terms of predictive distributions, the paper could benefit from a more explicit probabilistic model:
  - For example, a clear definition of $p(y^\* \mid x^\*, D_{\mathrm{HF}}, D_{\mathrm{LF}})$ induced by the three-step procedure, and how the LF TFM posterior and residual TFM posterior combine.
  - A simple graphical model/plate diagram connecting LF latent function, HF residuals, and observations would help ground the method and distinguish what is approximated by the TFM vs. what is assumed structurally.
  This is not a correctness issue, but I think such a probabilistic formulation would considerably strengthen the conceptual clarity and perceived rigor of the approach.

- **3. Relationship to transfer learning and in-context learning**
  Conceptually, FIRE sits at the intersection of MF regression, transfer learning, and in-context learning with TFMs. It would be valuable to more explicitly articulate:
  - How FIRE relates to standard views of transfer learning.
  - How its use of a TFM for MF regression compares to in-context learning paradigms in NLP and tabular domains (e.g., what is being “transferred” across tasks and fidelities, beyond the standard TabPFN usage).
  A more explicit positioning along these axes would make the conceptual novelty clearer to readers coming from those communities.

- **4. Deep learning baselines and retraining details**
  The empirical section (Fig 2 - Fig 5) includes deep learning–based MF baselines, which underperform not just FIRE but also some GP-based baselines. It would help to clarify:
  - Whether these deep models were retrained from scratch on each MF dataset.
  - Why, in the authors’ view, such retrained deep baselines perform worse than GP-based methods that require no training and are sensitive to kernel design.

- **5. Baseline choice: stronger TabPFN variants**
  Given that FIRE is built on TabPFN, it would be very informative to include TabPFN v2.5 (or the strongest current vanilla TabPFN model) as a baseline. In particular, the “vanilla TFM” (Fig. 4) seems to correspond to a strong baseline;  I suspect that TabPFN v2.5 would also be a competitive baseline in previous sections (Fig 2 - Fig 6). Including such a baseline would help isolate how much gain comes from the distribution-conditioned MF design vs. improvements in the underlying TFM.

---

> ### Author Rebuttal · Authors · 2026-03-31
>
> > I especially appreciated the theoretical and conceptual arguments for why conditioning on the full LF predictive distribution (rather than just the mean) should better capture input-dependent discrepancies.
>
> > FIRE consistently shows superior accuracy–uncertainty–runtime trade-offs, with strong performance both in nested and non-nested regimes and in extremely low HF data fractions (2–5%), where classical GP-based methods are known to struggle.
>
> **Authors:** We thank Reviewer u38Z for recommending the paper for acceptance and pointing out that our method FIRE has superior accuracy–uncertainty–runtime trade-offs that preserve practical benefits. We also appreciate the recognition of our extensive empirical validation and our well-justified distributional conditioning design choices.
>
> ---
>
> > 1. Clarity and referencing around Algorithm 1 & 2. Lack of explicit probabilistic formulation
>
> **Authors:** Thank you for the suggestions. We agree that Algorithm 1 should be explicitly referenced at each corresponding point in the narrative (Sections 3.1–3.4) to improve readability. We also agree that a formal probabilistic formulation, including a graphical model distinguishing structural assumptions from TFM approximations, would strengthen conceptual clarity. We will include these edits in the revision.
>
> ---
>
> > 3. Relationship to transfer learning and in-context learning
>
> **Authors:** Thank you for this suggestion. The in-context learning and transfer mechanisms in FIRE are properties of the underlying TFM (TabPFN), which learns to predict by conditioning on in-context examples—paralleling few-shot learning in NLP and tabular domains [1]. This is not a FIRE-specific design choice but an inherited capability of TabPFN (Hollmann, 2025). FIRE's distinct contribution is formulating multi-fidelity regression as an in-context learning problem, enabling transfer across fidelity levels, a perspective absent in standard TFM or tabular in-context learning settings.  We will add this statement in our revision.
>
> [1] Gardner, Josh, Juan C. Perdomo, and Ludwig Schmidt. "Large scale transfer learning for tabular data via language modeling." Advances in Neural Information Processing Systems 37 (2024).
>
> ---
>
> > 4. Deep learning baselines and retraining details
>
> **Authors:** We noted in Section 4.3 that all deep learning baselines were retrained from scratch for 1000 epochs on each dataset following established protocols in Xing et al. (2023). Their underperformance in the sparse HF regime (2–5%) is consistent with prior findings: both Xing et al. (2023) and Wang et al. (2022) independently report that MFBNN exhibits instability and lower accuracy than GP methods. Additionally, MFRNP assumes nested input domains, which might work less well on our non-nested benchmarks.
>
> ---
>
> > `Reviewer 9GNh`: Is **"vanilla TFM"** TabPFN on just the high fidelity dataset? I also think that it would be useful to try a single in-context dataset comprising the data from all fidelity levels rather than a two-stage approach. How well would this perform?
>
> > `Reviewer 5Zaw`: The contributions of the base model and the residual model are not fully analyzed. It would be helpful to provide further analysis or **ablations to understand how much each component contributes** to the final predictive performance.
>
> > `Reviewer u38Z `: 5. Baseline choice: stronger TabPFN variants: Given that FIRE is built on TabPFN, it would be very informative to include TabPFN v2.5 (or the **strongest current vanilla TabPFN model**) as a baseline.
>
> **Authors:** We collectively answer all these similar questions together here. We want to clarify that throughout the paper, FIRE already uses TabPFNv2.5 as its TFM backbone (Section 4.1). In Figure 4, we systematically analyzed each component of FIRE (stacking, residual learning, variance augmentation, quantile augmentation). The "Vanilla TFM" is a method using only a base TFM model. It’s a method that uses a single TabPFNv2.5, with fidelity as a categorical token. To avoid ambiguity, we will rename “Vanilla TFM” to **Single-pass TabPFNv2.5 (all fidelities + fidelity token)** throughout the text and figures.
>
> We present two more ablation baselines as asked by `Reviewer 5Zaw`:
> 1. "Only TFM Base Model (single pass only HF data)": a single pass in-context learning (ICL) conditioned only on the HF data
> 2. "Only TFM Base Model (single pass only LF data)": a single pass in-context learning (ICL) conditioned only on the LF data.
>
> The result is shown here: https://anonymous.4open.science/r/Anonymous_Figures-10E4/Figure4_new.png
>
> The results clearly show that the base model alone is insufficient (Vanilla TFM,Only TFM Base Models), and each added component — stacking & residual learning (+80 Elo), variance & quantile augmentation (+270 Elo) — contributes meaningfully.

---

> > ### Author Rebuttal · Reviewer_u38Z · 2026-04-01
> >
> > The rebuttal has effectively addressed my concerns. I especially appreciate the author's efforts in  Q5 for ablation studies. I am therefore maintaining my score and continuing to recommend acceptance.

---

> > > ### Author Response · Authors · 2026-04-01
> > >
> > > Thank you for your timely response. We are glad our rebuttal and additional ablation studies addressed your concerns.
> > >
> > > We sincerely appreciate your recommendation for acceptance.

---

### Official Review · Reviewer_9GNh · 2026-03-08

**Soundness:** 4
**Presentation:** 4
**Significance:** 3
**Originality:** 3
**Overall Recommendation:** 5
**Confidence:** 4

**Summary:**

This manuscript focuses on multi-fidelity regression where a model must effectively combine abundant low-fidelity data with scarce high-fidelity observations. Standard practice using traditional Gaussian-process–based surrogates often struggles with scalability and overfitting.

The paper proposes FIRE, a sensible training-free framework that leverages tabular foundation models (TFMs) to perform zero-shot in-context Bayesian inference, decomposing the task into low-fidelity prediction followed by a residual correction model conditioned on distributional summaries (mean, variance, and quantiles) of the low-fidelity posterior.

The paper carries out a thorough set of experiments to assess the new approaches and compare them to tranditional approaches. It also had a really nice ablation experiment to tease out the benefits of each part of the system.

Empirical evaluation on 31 multi-fidelity regression benchmarks spanning synthetic functions, hyperparameter optimization tasks, and engineering simulations shows that FIRE achieves superior accuracy and uncertainty calibration while offering favorable runtime compared to several Gaussian-process and neural multi-fidelity baselines.

The paper ends with some simple theory that supports some of the design choices.

**Compliance With Llm Reviewing Policy:**

Affirmed.

**Final Justification:**

I like the paper, recommend acceptance, and maintain my score.

**Key Questions For Authors:**

The idea of taking tabular foundation models like TabPFN and TabICL and applying them to mutifidelity problems is really sensible. It was clear that the authors had a very good understanding of multi-fidelity problems and how this should be handled when modelling (support for non-nested evals, non-stationary reliability of the low fidelity information, treating fidelity as an input rather than as an extra output etc.)

I thought figure 1 looks excellent. I found it hard to understand though — the pseudo-code and description in the text were very clear. I’d think about splitting up figure 1 into stages in the same way e.g. first show the fitting of the lower fidelity model, then using this to augment the high-fidelity dataset inputs and diff the outputs, then fitting the augmented high fidelity dataset.

I really like the ablation study. This really helped visualise the various sources of improvement. Is "vanilla TFM" TabPFN on just the high fidelity dataset? This is a useful comparison to have  to show that the multi fidelity approach improves TabPFN. It was interesting that the target differencing made things worse than just the vanilla TFM approach. It was also quite surprising that the variance input augmentation had such a strong effect. I wonder whether you considered not only differencing the target with the LF mean, but also then dividing by the LF predicted standard deviation? In this way you’d perform input dependent normalisation (this has helped in other contexts e.g. [1] but does require the LF model to be quite strong).

[1] Warm Starts Accelerate Conditional Diffusion

I also think that it would be useful to try a single in context dataset comprising the data from all fidelity levels rather than a two stage approach. How well would this perform? Is this too out of distribution from the TabPFN pertaining data? Did the authors consider finetuning TabPFN for multi-fidelity problems e.g. using synthetic MF data? Ideally this type of problem would be added into the pre-training distribution used by TabFFN. This would obviously be a more involved project and perhaps could be discussed as future work.

I also liked the Pareto frontier analysis — the model clearly inherits all the excellent properties of TabPFN in terms of good performance in blindingly fast run times.

In the theory section "7.1. TFMs as Bayesian Posterior Approximations” when citing the Muller et al., 2022, it would be fairer to also cite [1] as Muller essentially reproduced the results in this paper without clear attribution and has since captured the citations. In general the theory here is ok, but theorem 7.2 (that conditioning on extra information can only help or leave the performance the same) is straightforward. I felt similarly about Proposition 7.6 (that adding the quantile information helps), but applaud the efforts to place the method design into a rigorous foundation.

[2] Jonathan Gordon, John Bronskill, Matthias Bauer, Sebastian Nowozin, and Richard Turner. Meta-learning probabilistic inference for prediction. In International Conference on Learning Representations, 2019.

Minor comments

line 75 righthand column after equation 2 — comma in strange place

line 106 "TabPFN estimate test data x*’s posterior distribution p(y|x∗) ≈qθ(y|x∗,Dcontext)”
This should really say something like TabPFN estimates the posterior predictive distribution of the target y* at the input location x* …

**Limitations:**

The theory is a little underwhelming, but I don't see this as a major issue.

Some other model configurations would be good to compare to e.g. the single in context dataset approach.

**Strengths And Weaknesses:**

I liked this paper. I thought it was well written and presented. The justification for rationale behind the method design choices made sense and was well-explained. Figures were clear throughout and the results look strong.

It is true that the approach is fairly straightforward. The fact that it does not involve any training and can be built on top of any TabPFN / TabICL paper is definitely a strong point as it will continue to be relevant. But the simplicity means that the bar is raised for the experimental work. I actually thought that there was sufficient experimental work and evidence for strong performance to warrant acceptance, but I'm not a multi-fidelity modelling expert.

---

> ### Author Rebuttal · Authors · 2026-03-31
>
> > FIRE achieves superior accuracy and uncertainty calibration while offering favorable runtime compared to several Gaussian-process and neural multi-fidelity baselines. I liked this paper. I thought it was well written and presented. The justification for the rationale behind the method design choices made sense and was well-explained. Figures were clear throughout and the results look strong.
>
> **Authors:** We sincerely thank Reviewer 9GNh for liking our paper and recommending it for acceptance. We appreciate your recognition of FIRE’s superior accuracy and uncertainty calibration performance, as well as our extensive evaluations.
>
> ---
>
> > Is "vanilla TFM" TabPFN on just the high fidelity dataset? I also think that it would be useful to try a single in-context dataset comprising the data from all fidelity levels rather than a two-stage approach. How well would this perform?
>
> **Authors:** We kindly ask the reviewer to look at our collective response to this question at the bottom of `Reviewer u38Z`'s rebuttal.
>
> ---
>
> > I wonder whether you considered not only differencing the target with the LF mean, but also then dividing by the LF predicted standard deviation?
>
> **Authors:** We ran a new experiment with the suggested standardized mean. We show the result in the table below:
> | Method | Avg Rank | Avg NRMSE |
> |--------|----------|-----------|
> | Stacking w/ Mean | 1.32 | 0.051 |
> | Stacking w/ Standardized Mean | 1.68 | 0.054 |
>
> We see that in the results, *Stacking w/ Mean* and *Stacking w/ Standardized Mean* yield statistically indistinguishable Average NRMSE values. We hypothesize this is because TabPFN API’s internal data preprocessing (Hollmann, 2025; Section "Data preparation") already performs adaptive standardization on context inputs, effectively absorbing the benefit when running the model.
>
> ---
>
> > `Reviewer 9GNh`: Is this too **out of distribution** from the TabPFN pertaining data? Did the authors consider finetuning TabPFN?
>
> > `Reviewer 5Zaw`: How sensitive is FIRE to **out-of-distribution** tabular data that structurally deviates from the TFM's synthetic pre-training distribution? Could the TFM backbone benefit from **additional pretraining on related datasets** to improve its in-context learning performance?
>
> > `Reviewer d51u`: More careful analysis of the **TFM failure cases** is needed. FIRE relies on TFMs to make predictions. However, TFMs are known to have their own failure cases ...... Further, they tend to generalise poorly for datasets with domain / time shifts and feature noise (e.g. TabRED of Rubachev 2025). It is important to analyse how these drawbacks affect the multi-fidelity regression setup.
>
>
> **Authors:** Thank you for bringing this up. We agree that it will be part of future work discussions. We believe that fine-tuning for MF tasks is a valuable research direction for TFM. However, given that our HF dataset is very small, fine-tuning could easily lead to overfitting or catastrophic forgetting. We consider this topic outside the scope of the current paper, which focuses on the training-free MF regression goal.
>
> As for the out-of-distribution questions, unfortunately, the pre-training data of TabPFNv2.5 is not publicly released, so we cannot directly determine which datasets fall outside the pre-training distribution. Instead, we conduct a failure mode analysis. We identify five problems among the 31 datasets where FIRE ranks relatively worse (2nd–3rd in accuracy). The uncertainty quantification (UQ) results of NLL and Interval Score (IS) of 90% prediction interval (Yousefpour et al., 2024) are shown below (lower is better for both):
>
> | Problem | Avg NLL | Avg IS |
> |---------|---------|---------|
> | **FIRE’s average on all problems** | **-0.28** | **1.79** |
> | Branin | 0.05 | 4.91 |
> | Bohachevsky | 0.45 | 3.32 |
> | Beam | 3.41 | 5.59 |
> | Booth | -0.08 | 2.93 |
> | Himmelblau | 3.45 | 8.43 |
>
> We see that the co-occurrence of worse accuracy and worse calibration on these problems suggests a potential distribution mismatch. Importantly, FIRE's UQ remains informative even in these cases — they correctly signal reduced confidence when predictions are less reliable, a desirable property for downstream decision-making.
>
> ---
>
> **Authors' responses to other Reviewer 9GNh's comments and suggestions:**
> 1. We appreciate this constructive suggestion on Figure 1. We agree to restructure it into sequential stages (LF fitting → augmentation → residual fitting) to better align with the algorithmic description in the revision.
> 2. Thank you for pointing out these important references. We will properly cite Gordon et al. (2019) alongside Müller et al. (2022) in Section 7.1, and cite the Warm Starts work in our discussion of input-dependent normalization in the revision.
> 3. We will correct the misplaced comma on line 75 and revise line 106 to properly describe the posterior predictive distribution.
> 4. We will include all ablations done during the rebuttal in the Appendices in the revision.

---

> > ### Author Rebuttal · Reviewer_9GNh · 2026-04-01
> >
> > Thanks for the responses. I'm happy and maintain my "accept" recommendation.

---

> > > ### Author Response · Authors · 2026-04-01
> > >
> > > Thank you for maintaining the recommendation for acceptance. We are glad that our rebuttal addressed your concerns.

---

### Official Review · Reviewer_d51u · 2026-03-10

**Soundness:** 3
**Presentation:** 4
**Significance:** 3
**Originality:** 3
**Overall Recommendation:** 5
**Confidence:** 4

**Summary:**

The paper studies multi-fidelity regression in the difficult regime of non-nested low-/high-fidelity data and severe imbalance. The paper proposes **FIRE**, a training-free two-stage framework based on tabular foundation models (TFMs). In stage 1, FIRE predicts the low-fidelity (LF) target with a TFM. In stage 2, it predicts the HF residual using the LF predictive mean, variance, and quantiles. The final prediction is obtained as a sum of the two stages. The key idea is that distribution-level LF information may be a better transfer signal than the mean-only coupling used in many classical multi-fidelity methods. The experiments cover 31 datasets across, where FIRE demonstrates SOTA performance. The paper also includes an extensive ablation study and a theoretical argument for why distributional conditioning works.

**Compliance With Llm Reviewing Policy:**

Affirmed.

**Final Justification:**

The authors have addressed all my concerns during the rebuttal. Therefore, I retain my score and recommend acceptance.

**Key Questions For Authors:**

* **How sensitive is the method to LF uncertainty quality?** If the LF predictive distribution is poorly calibrated, does FIRE still help? Is the method robust to weak or biased LF signals?
* **How sensitive are results to the additive variance assumption?** The uncertainty propagation seems to assume the LF and residual errors are uncorrelated. How much does performance degrade if this assumption is violated?
* **How does FIRE behave when LF/HF correlation is weak?** It would be useful to include harder negative cases where LF information is only mildly informative.
* **How does FIRE scale with context size?** Since TFM inference can be expensive with growing context, can the authors clarify when FIRE remains preferable to lighter GP baselines?
* **Why does the performance degrade with more HF samples on Figure 6 for car and concrete datasets?** For all other datasets, the dependence seems intuitively declining as the data imbalance decreases. My suspicion is that the residual (HF correction) model is overfitting. Potentially, there is a distribution shift among the HF samples themselves.

**Strengths And Weaknesses:**

### Strengths

* **Clean and intuitive idea.** The method is simple: use a strong LF predictor, expose more of its predictive distribution, learn HF residuals from that richer signal. This is conceptually different from standard GP-style autoregressive multi-fidelity modeling, which relies on nested fidelity sets. The method is also training-free.
* **Strong empirical validation.** The authors evaluate on 31 datasets and offer extensive ablation results.
* **Interesting setup.** To be honest, this is the first time I come across multi-fidelity regression, despite coming from the statistics and engineering background. I find the setup intuitive and applicable. With respect to the previous literature on the topic, I find the proposed method more lightweight and also more practical as it extends to the non-nested and high LF/HF imbalance regimes.

### Weaknesses

* **Theory is more motivational than definitive.** The Bayes-risk argument is reasonable, but it mainly says: more conditioning information can help. That does not fully establish that the specific FIRE pipeline is theoretically well-justified or optimal in realistic finite-sample settings.
* **Uncertainty story is not fully convincing yet** The paper reports good NLL, which is important. But if uncertainty is one of the targets, then  the authors might want to consider including more direct evidence such as coverage, reliability diagrams, calibration by dataset or regime. Right now, it is not fully clear whether the uncertainty estimates are truly calibrated, or simply useful as features.
* **More careful analysis of the TFM failure cases is needed.** FIRE relies on TFMs to make predictions. However, TFMs are known to have their own failure cases. For example, their scaling cost is quadratic in sample size and number of features. Further, they tend to generalise poorly for datasets with domain / time shifts and feature noise (e.g. TabRED of Rubachev 2025). It is important to analyse how these drawbacks affect the multi-fidelity regression setup.

---

> ### Author Rebuttal · Authors · 2026-03-31
>
> > Clean and intuitive idea. The method is simple: use a strong LF predictor, expose more of its predictive distribution, learn HF residuals from that richer signal. Strong empirical validation. The authors evaluate on 31 datasets and offer extensive ablation results.
>
> **Authors:** Thank you for recommending the paper for acceptance with positive comments on FIRE's intuitive algorithm design. FIRE's distribution-conditioning and residual learning enable it to achieve top performance across 31 benchmarks under non-nested and severe-imbalance data settings.
>
> ---
>
> > Right now, it is not fully clear whether the uncertainty estimates are truly calibrated, or simply useful as features.
>
> **Authors:** Thank you for pointing this out. We conducted additional uncertainty quantification (UQ) analysis using a reliability diagram and Interval Scores (IS) (Yousefpour et al., 2024) across all benchmarks.
> - Reliability diagram figure: https://anonymous.4open.science/r/Anonymous_Figures-10E4/Uncertainty_Reliable.png
> - Overall average IS ranking figure: https://anonymous.4open.science/r/Anonymous_Figures-10E4/Uncertainty_IS.png
>
> The new reliability diagram and IS analyses show that FIRE-TabPFNv2.5 is well-calibrated and competitive on UQ, while FIRE-GP is strongest in pure calibration. We will revise the paper to state this trade-off explicitly: the TabPFN variant gives the best overall accuracy/runtime trade-off, while GP variants retain a calibration advantage.
>
> ---
>
> > More careful analysis of the TFM failure cases is needed. For example, their scaling cost is quadratic in sample size and number of features.
>
> **Authors**: We kindly ask the reviewer to look at our response in `Reviewer 5Zaw`'s rebuttal on our experiment of FIRE (TabICLv2) for the scaling issue and refer to `Reviewer 9GNh`'s rebuttal for failure mode analysis.
>
> ---
>
> > How sensitive is the method to LF uncertainty quality?
>
> **Authors:** To address this question, we corrupted the base model’s variance and quantile predictions with three different levels (low, medium, high) of Gaussian noise.  The result figure can be viewed here: https://anonymous.4open.science/r/Anonymous_Figures-10E4/Noise_to_Base.png
>
> FIRE remains above the strongest baseline under low-to-moderate corruption and loses the advantage only under extreme corruption, indicating robustness to imperfect LF posteriors but not invariance to arbitrarily poor LF signals.
>
> ---
>
> > How sensitive are results to the additive variance assumption?
>
> **Authors:** We acknowledge that the additive variance assumption (Assumption C.3) is a practical approximation, as noted in Appendix C.3. Empirically, FIRE achieves the best NLL across all 31 benchmarks (Figure 2b), suggesting this approximation is well-suited for the problem regimes studied. We believe additional specialized benchmarks will be required to study the regime where LF and residual errors are correlated, which is outside the current scope of our paper. We will add this as a direction for future work.
>
> ---
>
> > How does FIRE behave when LF/HF correlation is weak?
>
> **Authors:** We've shown a simple empirical validation in **Appendix C.11** of the paper using a classic heteroscedastic problem (Figure 23, Tables 1 and 2). FIRE performs best on these synthetic heteroscedastic cases with the LF-HF correlation worsened by adding input-dependent noise to HF samples. To further evaluate FIRE under weak LF-HF correlation on complex problems, we corrupted the LF data with 3 different levels (low, medium, high) of Gaussian noise. The result figure can be viewed here: https://anonymous.4open.science/r/Anonymous_Figures-10E4/Noise_to_LFdata.png
>
> Again, we observe that FIRE remains above the best method under low-to-moderate corruption and loses the advantage only under extreme corruption, indicating robustness to imperfect LF data.
>
> ---
>
> > `Reviewer 5Zaw`: TFMs strictly inherit **context window** constraints.
>
> > `Reviewer d51u`: How does FIRE scale with **context size**?
>
> **Authors:** Context length is a limitation of the current TFM backbone. However, in our 31-benchmark suite, no dataset exceeded the TabPFNv2.5 context limit (50k samples, 2k features), so the limitation did not affect the experiments reported here. For larger datasets, subset-selection or preprocessing strategies from prior TFM work are promising extensions [1], and we will explicitly discuss this in our future work.
>
> [1] Feuer, Benjamin, et al. "Tunetables: Context optimization for scalable prior-data fitted networks." Advances in Neural Information Processing Systems (2024).
>
> ---
>
> > Why does the performance degrade with more HF samples on Figure 6 for car and concrete datasets?
>
> **Authors:** We agree that overfitting is a plausible explanation for the Car and Concrete trends. Importantly, this behavior appears only on two small datasets (200 samples and 500 samples) and does not change the overall ranking. We will add this caveat and, if space permits, include further statistical analysis.

---

> > ### Author Rebuttal · Reviewer_d51u · 2026-04-01
> >
> > The rebuttal substantially addresses my main concerns, particularly through the additional analyses on uncertainty calibration and robustness to degraded LF uncertainty / LF data quality. I am therefore maintaining my score and continuing to recommend acceptance.

---

> > > ### Author Response · Authors · 2026-04-01
> > >
> > > Thank you for your timely response. We are glad our rebuttal and additional analyses on uncertainty calibration addressed your concerns.
> > >
> > > We sincerely appreciate your recommendation for acceptance.

---

### Official Review · Reviewer_5Zaw · 2026-03-12

**Soundness:** 2
**Presentation:** 3
**Significance:** 2
**Originality:** 3
**Overall Recommendation:** 4
**Confidence:** 3

**Summary:**

The paper proposes FIRE, a multi-fidelity regression framework based on distribution-conditioned in-context learning with tabular foundation models (TFMs). The key idea is to use predictive distribution statistics from a low-fidelity model as conditioning features for a residual model that predicts the discrepancy between low- and high-fidelity outputs. Both the base predictor and residual model are implemented using TFMs and operate via in-context learning without task-specific training. The paper provides theoretical motivation for conditioning on predictive distributions and evaluates the method on 31 benchmarks, demonstrating strong predictive performance and competitive efficiency.

**Compliance With Llm Reviewing Policy:**

Affirmed.

**Ethics Expertise Needed:**

["Other Expertise"]

**Final Justification:**

Concerns have been addressed.

**Key Questions For Authors:**

- Since pretrained TFMs are trained on generic tabular priors and may not encode specific priors for cross-fidelity discrepancy distributions, would training a residual model from scratch potentially yield a better Pareto frontier between accuracy and computational cost?

- Given that the framework relies on the pre-trained TFM's prior, how sensitive is FIRE to out-of-distribution tabular data that structurally deviates from the TFM's synthetic pre-training distribution?

- TFMs strictly inherit context window constraints. How does FIRE scale when the number of available low-fidelity observations significantly exceeds the TFM's maximum context length?

- Could the TFM backbone benefit from additional pretraining on related datasets to improve its in-context learning performance?

- How is uncertainty calibration evaluated in the experiments, and how does improved calibration translate into practical benefits?

**Limitations:**

### Limitations

- The approach relies on TFMs and therefore inherits context window limitations. More discussion or experiments on scalability with increasing context size would be helpful.

- Using two TFMs (base and residual) may increase inference cost due to transformer attention. Exploring lighter ICL mechanisms or alternative backbones could be interesting.

**Strengths And Weaknesses:**

#### Strengths

- The paper introduces a paradigm for multi-fidelity learning by leveraging distribution-conditioned in-context learning. Using predictive distribution statistics  as conditioning signals for residual correction is novel and effectively handles heteroscedastic discrepancies.

- The adoption of TFMs as both the base and residual models is well aligned with the proposed framework. The ICL mechanism enables approximate Bayesian inference without the burden of task-specific parameter training, avoiding overfitting in data-scarce regimes.

- The empirical evaluation is extensive. The experiments cover a wide range of datasets and demonstrate strong predictive accuracy and efficiency compared with existing methods.

#### Weaknesses

- The current design aggregates all low-fidelity data into a single LF model, which effectively reduces the setting to a bi-level structure. This may limit the ability of the method to fully exploit richer multi-fidelity hierarchies.

- The contributions of the base model and the residual model are not fully analyzed. It would be helpful to provide further analysis or ablations to understand how much each component contributes to the final predictive performance.

- It is unclear whether the performance gains mainly come from the proposed distribution-conditioned learning paradigm or from the strong capability of the TFM backbone.

---

> ### Author Rebuttal · Authors · 2026-03-31
>
> > Using predictive distribution statistics as conditioning signals for residual correction is novel and effectively handles heteroscedastic discrepancies. The experiments cover a wide range of datasets and demonstrate strong predictive accuracy and efficiency compared with existing methods.
>
> **Authors:** Thank you for recognizing FIRE’s novelty of using tabular foundation models (TFMs) with distribution-conditioned in-context learning, and highlighting our extensive empirical evaluations on a wide range of datasets.
>
> ---
>
> > The current design aggregates all low-fidelity data into a single LF model, which effectively reduces the setting to a bi-level structure.
>
> **Authors:** We have already addressed this question with empirical results in Appendix A.1 (Figure 8), which show that: FIRE’s bi-level structure is 10x faster and more accurate (+80 Elo rating). Moreover, FIRE’s bi-level structure does not discard fidelity information during aggregation: each data retains its fidelity index as an explicit feature column (fidelity token), enabling the TFM to learn inter-fidelity relationships within a single forward pass.
>
> ---
>
> > How much each component contributes to the final predictive performance? Performance gains mainly come from the proposed distribution-conditioned learning paradigm or from the strong capability of the TFM backbone?
>
> **Authors:** We kindly ask the reviewer to view our collective response and our updated Figure 4 component ablation in `Reviewer u38Z`'s rebuttal. We note that the TFM backbone alone is not enough: The single-pass TabPFNv2.5 scores 1000 Elo in Figure 4, while FIRE scores 1273. The paradigm alone is also not enough: FIRE-GP improves over MF-GP baselines but still trails FIRE-TabPFNv2.5 (Figure 9). So both the distribution-conditioned formulation and the TFM backbone contribute practically.
>
> ---
>
> > Would training a residual model from scratch potentially yield a better Pareto frontier between accuracy and computational cost?
>
> **Authors:** Our appendix already addresses this. The answer is no: training the residual model from scratch did not produce a better accuracy-cost frontier than FIRE (TabPFNv2.5). In Appendix A.6, TFM residuals consistently outperform GP residuals, and in Appendix A.7, further improving the TFM residual with post-hoc ensembling improves accuracy, confirming that residual model quality matters. We further tested scratch-trained residuals using XGBoost, Random Forest (RF), and GP models. The result is shown here: https://anonymous.4open.science/r/Anonymous_Figures-10E4/Res_models.png
> FIRE with a TFM residual achieved the best overall results. For example, TabPFN+FIRE outperforms TabPFN+GP (1000 vs. 774 Elo), TabPFN+RF (738), and TabPFN+XGBoost (605). This aligns with TabPFNv2.5 outperforming tree and gradient boosting methods on TabArena (Erickson, 2025).
>
> ---
>
> > How is uncertainty calibration evaluated in the experiments, and how does improved calibration translate into practical benefits?
>
> **Authors**: We evaluate uncertainty calibration using negative log-likelihood (NLL), which measures whether the predicted mean and variance match the observed outcomes. Lower NLL means the model is not only accurate, but also assigns uncertainty more appropriately. In practice, better NLL calibration means the model’s uncertainty estimates are more trustworthy. This is useful in engineering design because it supports better sample allocation, safer exploration, and more reliable risk-aware decisions, assisting downstream optimization and decision-making (Feldsteinet al., 2020; Ravi et al., 2024).
>
> ---
>
> > Exploring lighter ICL mechanisms or alternative backbones could be interesting.
>
> **Authors**: To directly address this question with `Reviewer d51u`'s TFM scaling concern, we implemented FIRE with TabICLv2, which uses a more efficient attention mechanism and achieves 10.6× faster inference on large contexts[1]. The result is shown here: https://anonymous.4open.science/r/Anonymous_Figures-10E4/TabICLv2.pdf
>
> FIRE with TabICLv2 is slightly less accurate (-100 Elo rating) than FIRE with TabPFNv2.5, but it still outperforms all seven baselines and remains on the NRMSE Pareto front with the fastest speed. This supports our main claim that FIRE is backbone-agnostic and offers a clear accuracy-speed tradeoff: TabICLv2 is preferable when runtime matters most, while TabPFNv2.5 remains the better choice when accuracy is the priority.
>
> [1] Qu, Jingang, et al. "TabICLv2: A better, faster, scalable, and open tabular foundation model." arXiv preprint arXiv:2602.11139 (2026).
>
> ---
>
> **Author's collective response in other rebuttal**:
> 1. Context window constraints: We kindly ask you to view our collective response in `Reviewer d51u`'s rebuttal.
> 2. Out of distribution performance & additional pretraining (finetuning): We kindly ask you to view our collective response in `Reviewer 9GNh`'s rebuttal.
>
> We hope our rebuttal and new experiments can convince the reviewer to raise their score.

---

### Decision · Program_Chairs · 2026-04-30

**Decision:**

Accept (spotlight)

**Comment:**

This paper introduces FIRE, a training-free multi-fidelity regression framework that utilizes tabular foundation models (TFMs) to perform zero-shot in-context learning. By conditioning a high-fidelity residual model on the predictive distributions (mean, variance, and quantiles) of a low-fidelity base model, the approach effectively captures heteroscedastic discrepancies across fidelities.

The reviewers unanimously agreed that the paper presents a clean, intuitive, and practical approach to a significant problem in scientific ML and hyperparameter optimization. The empirical validation was widely praised as extensive, convincing, and well-executed.

During the discussion phase, reviewers raised several constructive points, which the authors addressed thoroughly:
* **Uncertainty Quantification:** Reviewers requested more direct evidence of uncertainty calibration beyond Negative Log-Likelihood metrics. The authors provided Reliability Diagrams and Interval Scores, demonstrating that the framework produces well-calibrated uncertainty estimates.
* **Scalability and Robustness:** In response to concerns about TFM context window limits, the authors successfully implemented FIRE with a lighter TabICLv2 model, showing it maintains Pareto-optimal accuracy while running over 10x faster. They also added noise-corruption experiments to prove the method's robust performance even when faced with imperfect low-fidelity signals.
* **Ablations and Baselines:** The authors clarified the individual contributions of the underlying TFM versus the distribution-conditioned paradigm. The new ablations successfully demonstrated that the combination of both elements—stacking, residual learning, and statistical augmentation—is necessary to achieve the reported state-of-the-art gains.
* **Presentation and Clarity:** Reviewers identified minor structural gaps, such as orphaned algorithmic references in the text and the need for a more explicit probabilistic graphical model. The authors committed to incorporating these structural improvements and better contextualizing the work within the broader transfer learning literature.

Following the rebuttal, all reviewers recommended acceptance, confirming that their concerns were fully resolved. The paper is technically sound, well-written, and introduces a novel and practical application of foundation models to multi-fidelity tasks. The training-free nature of the framework and its robustness in data-scarce regimes make it highly useful to the ICML community. I confidently recommend this paper for acceptance.